# A Fractional Graph Laplacian Approach
# to Oversmoothing

**Sohir Maskey**[*]
Department of Mathematics,
LMU Munich
maskey@math.lmu.de

**Raffaele Paolino**[*]
Department of Mathematics & MCML,
LMU Munich
paolino@math.lmu.de

**Aras Bacho**
Department of Mathematics,
LMU Munich

**Gitta Kutyniok**
Department of Mathematics & MCML,
LMU Munich

## Abstract

Graph neural networks (GNNs) have shown state-of-the-art performances in various applications. However, GNNs often struggle to capture long-range dependencies in graphs due to oversmoothing. In this paper, we generalize the concept of oversmoothing from undirected to directed graphs. To this aim, we extend the notion of Dirichlet energy by considering a directed symmetrically normalized Laplacian. As vanilla graph convolutional networks are prone to oversmooth, we adopt a neural graph ODE framework. Specifically, we propose fractional graph Laplacian neural ODEs, which describe non-local dynamics. We prove that our approach allows propagating information between distant nodes while maintaining a low probability of long-distance jumps. Moreover, we show that our method is more flexible with respect to the convergence of the graph's Dirichlet energy, thereby mitigating oversmoothing. We conduct extensive experiments on synthetic and real-world graphs, both directed and undirected, demonstrating our method's versatility across diverse graph homophily levels. Our code is available on GitHub.

## 1 Introduction

Graph neural networks (GNNs) (Gori et al., 2005; Scarselli et al., 2009; Bronstein et al., 2017) have emerged as a powerful class of machine learning models capable of effectively learning representations of structured data. GNNs have demonstrated state-of-the-art performance in a wide range of applications, including social network analysis (Monti et al., 2019), molecular property prediction (Gilmer et al., 2017), and recommendation systems (J. Wang et al., 2018; Fan et al., 2019). The majority of existing work on GNNs has focused on undirected graphs (Defferrard et al., 2016; Kipf et al., 2017; Hamilton et al., 2017), where edges have no inherent direction. However, many real-world systems, such as citation networks, transportation systems, and biological pathways, are inherently directed, necessitating the development of methods explicitly tailored to directed graphs.

Despite their success, most existing GNN models struggle to capture long-range dependencies, which can be critical for specific tasks, such as node classification and link prediction, and for specific graphs, such as heterophilic graphs. This shortcoming also arises from the problem of *oversmoothing*, where increasing the depth of GNNs results in the node features converging to similar values that only convey information about the node's degree (Oono et al., 2019; Cai et al., 2020). Consequently,

---

[*]Equal contribution.

37th Conference on Neural Information Processing Systems (NeurIPS 2023).

scaling the depth of GNNs is not sufficient to broaden receptive fields, and other approaches are necessary to address this limitation. While these issues have been extensively studied in undirected graphs (Q. Li et al., 2018; G. Li et al., 2019; Luan, M. Zhao, et al., 2019; D. Chen et al., 2020; Rusch et al., 2022), their implications for directed graphs remain largely unexplored. Investigating these challenges and developing effective solutions is crucial for applying GNNs to real-world scenarios.

Over-smoothing has been shown to be intimately related to the graph's *Dirichlet energy*, defined as

$$\mathscr{E}(\mathbf{x}) := \frac{1}{4} \sum_{i,j=1}^{N} a_{i,j} \left\| \frac{\mathbf{x}_i}{\sqrt{d_i}} - \frac{\mathbf{x}_j}{\sqrt{d_j}} \right\|_2^2,$$

where $\mathbf{A} = (a_{i,j})_{i,j=1}^{N}$ represents the adjacency matrix of the underlying graph, $\mathbf{x} \in \mathbb{R}^{N \times K}$ denotes the node features, and $d_i \in \mathbb{R}$ the degree of node $i$. Intuitively, the Dirichlet energy measures the smoothness of nodes' features. Therefore, a GNN that minimizes the Dirichlet energy is expected to perform well on *homophilic* graphs, where similar nodes are likely to be connected. Conversely, a GNN that ensures high Dirichlet energy should lead to better performances on *heterophilic* graphs, for which the nodes' features are less smooth.

This paper aims to bridge the gap in understanding oversmoothing for directed graphs. To this aim, we generalize the concept of Dirichlet energy, providing a rigorous foundation for analyzing oversmoothing. Specifically, we consider the directed symmetrically normalized Laplacian, which accommodates directed graph structures and recovers the usual definition in the undirected case. Even though the directed symmetrically normalized Laplacian has been already used (Zou et al., 2022), its theoretical properties remain widely unexplored.

However, a vanilla graph convolutional network (GCN) (Kipf et al., 2017) implementing this directed Laplacian alone is not able to prevent oversmoothing. For this reason, we adopt a graph neural ODE framework, which has been shown to effectively alleviate oversmoothing in undirected graphs (Bodnar et al., 2022; Rusch et al., 2022; Di Giovanni et al., 2023).

## 1.1  Graph Neural ODEs

The concept of neural ODE was introduced by Haber et al. (2018) and R. T. Q. Chen et al. (2018), who first interpreted the layers in neural networks as the time variable in ODEs. Building on this foundation, Poli et al. (2021), Chamberlain et al. (2021), and Eliasof et al. (2021) extended the connection to the realm of GNNs, resulting in the development of graph neural ODEs. In this context, each node $i$ of the underlying graph is described by a state variable $\mathbf{x}_i(t) \in \mathbb{R}^K$, representing the node $i$ at time $t$. We can define the dynamics of $\mathbf{x}(t)$ via the node-wise ODE

$$\mathbf{x}'(t) = f_{\mathbf{w}}(\mathbf{x}(t)), \ t \in [0, T],$$

subject to the initial condition $\mathbf{x}(0) = \mathbf{x}_0 \in \mathbb{R}^{N \times K}$, where the function $f_{\mathbf{w}} : \mathbb{R}^{N \times K} \to \mathbb{R}^{N \times K}$ is parametrized by the learnable parameters $\mathbf{w}$.

The graph neural ODE can be seen as a continuous learnable architecture on the underlying graph, which computes the final node representation $\mathbf{x}(T)$ from the input nodes' features $\mathbf{x}_0$. Typical choices for $f_{\mathbf{w}}$ include attention-based functions (Chamberlain et al., 2021), which generalize graph attention networks (GATs) (Velickovic et al., 2018), or convolutional-like functions (Di Giovanni et al., 2023) that generalize GCNs (Kipf et al., 2017).

How can we choose the learnable function $f_{\mathbf{w}}$ to accommodate both directed and undirected graphs, as well as different levels of homophily? We address this question in the following subsection.

## 1.2  Fractional Laplacians

The continuous fractional Laplacian, denoted by $(-\Delta)^\alpha$ for $\alpha > 0$, is used to model non-local interactions. For instance, the fractional heat equation $\partial_t u + (-\Delta)^\alpha u = 0$ provides a flexible and accurate framework for modeling anomalous diffusion processes. Similarly, the fractional diffusion-reaction, quasi-geostrophic, Cahn-Hilliard, porous medium, Schrödinger, and ultrasound equations are more sophisticated models to represent complex anomalous systems (Pozrikidis, 2018).

Similarly to the continuous case, the fractional graph Laplacian (FGL) (Benzi et al., 2020) models non-local network dynamics. In general, the FGL does not inherit the sparsity of the underlying

graph, allowing a random walker to leap rather than walk solely between adjacent nodes. Hence, the FGL is able to build long-range connections, making it well-suited for heterophilic graphs.

## 1.3 Main Contributions

We present a novel approach to the fractional graph Laplacian by defining it in the singular value domain, instead of the frequency domain (Benzi et al., 2020). This formulation bypasses the need for computing the Jordan decomposition of the graph Laplacian, which lacks reliable numerical methods. We show that our version of the FGL can still capture long-range dependencies, and we prove that its entries remain reasonably bounded.

We then propose two FGL-based neural ODEs: the fractional heat equation and the fractional Schrödinger equation. Importantly, we demonstrate that solutions to these FGL-based neural ODEs offer increased flexibility in terms of the convergence of the Dirichlet energy. Notably, the exponent of the fractional graph Laplacian becomes a learnable parameter, allowing our network to adaptively determine the optimal exponent for the given task and graph. We show that this can effectively alleviate oversmoothing in undirected and directed graphs.

To validate the effectiveness of our approach, we conduct extensive experiments on synthetic and real-world graphs, with a specific focus on supervised node classification. Our experimental results indicate the advantages offered by fractional graph Laplacians, particularly in non-homophilic and directed graphs.

## 2 Preliminaries

We denote a graph as $\mathcal{G} = (\mathcal{V}, \mathcal{E})$, where $\mathcal{V}$ is the set of nodes, $\mathcal{E}$ is the set of edges, and $N = |\mathcal{V}|$ is the number of nodes. The adjacency matrix $\mathbf{A} := \{a_{i,j}\}$ encodes the edge information, with $a_{i,j} = 1$ if there is an edge directed from node $j$ to $i$, and $0$ otherwise. The in- and out-degree matrices are then defined as $\mathbf{D}_{\text{in}} = \text{diag}(\mathbf{A}\mathbf{1})$, $\mathbf{D}_{\text{out}} = \text{diag}(\mathbf{A}^\mathsf{T}\mathbf{1})$, respectively. The node feature matrix $\mathbf{x} \in \mathbb{R}^{N \times K}$ contains for every node its feature in $\mathbb{R}^K$.

Given any matrix $\mathbf{M} \in \mathbb{C}^{n \times n}$, we denote its spectrum by $\lambda(\mathbf{M}) := \{\lambda_i(\mathbf{M})\}_{i=1}^n$ in ascending order w.r.t. to the real part, i.e., $\Re\lambda_1(\mathbf{M}) \leq \Re\lambda_2(\mathbf{M}) \leq \ldots \leq \Re\lambda_n(\mathbf{M})$. Furthermore, we denote by $\|\mathbf{M}\|_2$ and $\|\mathbf{M}\|$ the Frobenius and spectral norm of $\mathbf{M}$, respectively. Lastly, we denote by $\mathbf{I}_n$ the identity matrix, where we omit the dimension $n$ when it is clear from the context.

**Homophily and Heterophily**   Given a graph $\mathcal{G} = (\mathcal{V}, \mathcal{E})$ with labels $\mathbf{y} = \{y_i\}_{i \in \mathcal{V}}$, the *homophily* of the graph indicates whether connected nodes are likely to have the same labels; formally,

$$\mathscr{H}(\mathcal{G}) = \frac{1}{N} \sum_{i=1}^N \frac{|\{j \in \{1, \ldots, N\} : a_{i,j} = 1 \wedge y_i = y_j\}|}{|\{j \in \{1, \ldots, N\} : a_{i,j} = 1\}|},$$

where the numerator represents the number of neighbors of node $i \in \mathcal{V}$ that have the same label $y_i$ (Pei et al., 2019). We say that $\mathcal{G}$ is *homophilic* if $\mathscr{H}(\mathcal{G}) \approx 1$ and *heterophilic* if $\mathscr{H}(\mathcal{G}) \approx 0$.

## 3 Dirichlet Energy and Laplacian for (Directed) Graphs

In this section, we introduce the concept of Dirichlet energy and demonstrate its relationship to a directed Laplacian, thereby generalizing well-known results for undirected graphs.

**Definition 3.1.** *The* Dirichlet energy *is defined on the node features* $\mathbf{x} \in \mathbb{R}^{N \times K}$ *of a graph* $\mathcal{G}$ *as*

$$\mathscr{E}(\mathbf{x}) := \frac{1}{4} \sum_{i,j=1}^N a_{i,j} \left\| \frac{\mathbf{x}_i}{\sqrt{d_i^{in}}} - \frac{\mathbf{x}_j}{\sqrt{d_j^{out}}} \right\|_2^2. \tag{1}$$

The Dirichlet energy measures how much the features change over the nodes of $\mathcal{G}$, by quantifying the disparity between the normalized outflow of information from node $j$ and the normalized inflow of information to node $i$.

**Definition 3.2.** *We define the* symmetrically normalized adjacency (SNA) *as* $\mathbf{L} := \mathbf{D}_{in}^{-1/2} \mathbf{A} \mathbf{D}_{out}^{-1/2}$.

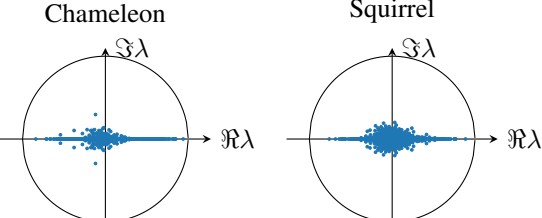

Figure 1: Spectrum $\lambda\left(\mathbf{L}\right)$ of common directed real-world graphs. The Perron-Frobenius eigenvalue is $\lambda_{\mathrm{PF}} \approx 0.94$ for Chameleon, and $\lambda_{\mathrm{PF}} \approx 0.89$ for Squirrel.

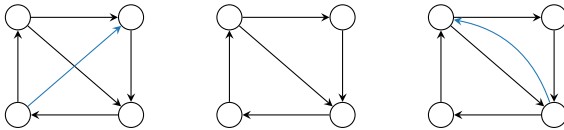

Figure 2: Examples of non-weakly balanced (left), weakly balanced (center), and balanced (right) directed graphs. The Perron-Frobenius eigenvalue of the left graph is $\lambda_{\mathrm{PF}} \approx 0.97 \neq 1$, while for the middle and right graphs $\lambda_{\mathrm{PF}} = 1$.

Note that $\mathbf{L}$ is symmetric if and only if $\mathcal{G}$ is undirected; the term "symmetrically" refers to the both-sided normalization rather than the specific property of the matrix itself.

It is well-known that the SNA's spectrum of a connected undirected graph lies within $[-1, 1]$ (Chung, 1997). We extend this result to directed graphs, which generally exhibit complex-valued spectra.

**Proposition 3.3.** *Let $\mathcal{G}$ be a directed graph with SNA $\mathbf{L}$. For every $\lambda \in \lambda(\mathbf{L})$, it holds $|\lambda| \leq 1$.*

Proposition 3.3 provides an upper bound for the largest eigenvalue of any directed graph, irrespective of its size. However, many other spectral properties do not carry over easily from the undirected to the directed case. For example, the SNA may not possess a one-eigenvalue, even if the graph is strongly connected (see, e.g., Figures 1 to 2). The one-eigenvalue is of particular interest since its eigenvector $\mathbf{v}$ corresponds to zero Dirichlet energy $\mathscr{E}(\mathbf{v}) = 0$. Therefore, studying when $1 \in \lambda(\mathbf{L})$ is crucial to understanding the behavior of the Dirichlet energy. We fully characterize the set of graphs for which $1 \in \lambda(\mathbf{L})$; this is the scope of the following definition.

**Definition 3.4.** *A graph $\mathcal{G} = (\mathcal{V}, \mathcal{E})$ is said to be* balanced *if $d_i^{in} = d_i^{out}$ for all $i \in \{1, \ldots, N\}$, and* weakly balanced *if there exists $\mathbf{k} \in \mathbb{R}^N$ such that $\mathbf{k} \neq 0$ and*

$$\sum_{j=1}^{N} a_{i,j} \left( \frac{k_j}{\sqrt{d_j^{out}}} - \frac{k_i}{\sqrt{d_i^{in}}} \right) = 0 \, , \ \forall i \in \{1, \ldots, N\} \, .$$

It is straightforward to see that a balanced graph is weakly balanced since one can choose $k_i = \sqrt{d_i^{in}}$. Hence, all undirected graphs are also weakly balanced. However, as shown in Figure 2, the set of balanced graphs is a proper subset of the set of weakly balanced graphs.

**Proposition 3.5.** *Let $\mathcal{G}$ be a directed graph with SNA $\mathbf{L}$. Then, $1 \in \lambda(\mathbf{L})$ if and only if the graph is weakly balanced. Suppose the graph is strongly connected, then $-1 \in \lambda(\mathbf{L})$ if and only if the graph is weakly balanced with an even period.*

Proposition 3.5 generalizes a well-known result for undirected graphs: $-1 \in \lambda(\mathbf{L})$ if and only if the graph is bipartite, i.e., has even period. The next result shows that the Dirichlet energy defined in (1) and the SNA are closely connected.

**Proposition 3.6.** *For every $\mathbf{x} \in \mathbb{C}^{N \times K}$, it holds $\mathscr{E}(\mathbf{x}) = \frac{1}{2}\Re\left(\mathrm{trace}\left(\mathbf{x}^{\mathsf{H}}\left(\mathbf{I} - \mathbf{L}\right)\mathbf{x}\right)\right)$. Moreover, there exists $\mathbf{x} \neq \mathbf{0}$ such that $\mathscr{E}(\mathbf{x}) = 0$ if and only if the graph is weakly balanced.*

Proposition 3.6 generalizes the well-known result from the undirected (see, e.g., Cai et al., 2020, Definition 3.1) to the directed case. This result is an important tool for analyzing the evolution of the Dirichlet energy in graph neural networks.

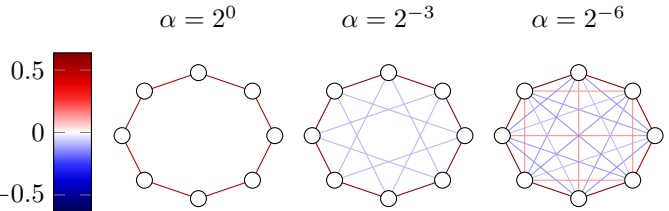

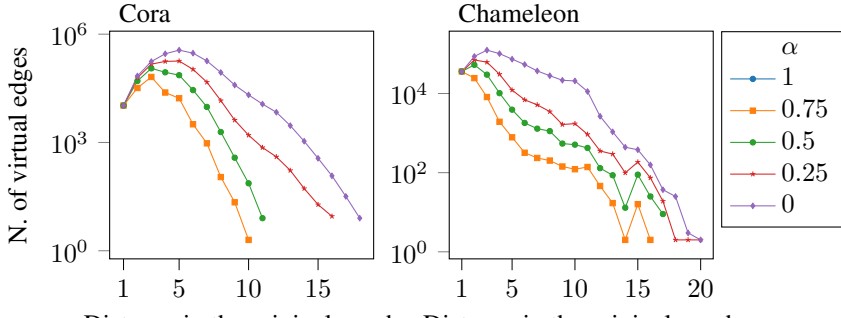

(a) Synthetic cycle graph. The values of the fractional Laplacian can also be negative.

(b) Real-world graphs. We count the number of virtual edges built by the fractional Laplacian based on the distance $d(i, j)$ in the original graph. The number of virtual edges increases as $\alpha$ decreases.

Figure 3: Visual representation of long-range edges built by the fractional Laplacian.

## 4 Fractional Graph Laplacians

We introduce the fractional graph Laplacian through the singular value decomposition (SVD). This approach has two key advantages over the traditional definition (Pozrikidis, 2018; Benzi et al., 2020) in the spectral domain. First, it allows defining the fractional Laplacian based on any choice of graph Laplacian, including those with negative or complex spectrum such as the SNA. Secondly, the SVD is computationally more efficient and numerically more stable than the Jordan decomposition, which would be necessary if the fractional Laplacian was defined in the spectral domain.

Consider a directed graph with SNA $\mathbf{L}$ and its SVD $\mathbf{L} = \mathbf{U\Sigma V}^{\mathsf{H}}$, where $\mathbf{U}, \mathbf{V} \in \mathbb{C}^{N \times N}$ are unitary matrices and $\mathbf{\Sigma} \in \mathbb{R}^{N \times N}$ is a diagonal matrix. Given $\alpha \in \mathbb{R}$, we define the $\alpha$-*fractional graph Laplacian* [2] ($\alpha$-*FGL* in short) as

$$\mathbf{L}^{\alpha} := \mathbf{U\Sigma}^{\alpha}\mathbf{V}^{\mathsf{H}}.$$

In undirected graphs, the $\alpha$-FGL preserves the sign of the eigenvalues $\lambda$ of $\mathbf{L}$ while modifying their magnitudes, i.e., $\lambda \mapsto \text{sign}(\lambda) |\lambda|^{\alpha}$. [3]

The $\alpha$-FGL is generally less sparse than the original SNA, as it connects nodes that are not adjacent in the underlying graph. The next theorem proves that the weight of such "virtual" edges is bounded.

**Theorem 4.1.** *Let $\mathcal{G}$ be a directed graph with SNA $\mathbf{L}$. For $\alpha > 0$, if the distance $d(i, j)$ between nodes $i$ and $j$ is at least 2, then*

$$\left| (\mathbf{L}^{\alpha})_{i,j} \right| \leq \left( 1 + \frac{\pi^2}{2} \right) \left( \frac{\|\mathbf{L}\|}{2 \left( d(i, j) - 1 \right)} \right)^{\alpha}.$$

We provide a proof of Theorem 4.1 in Appendix C. In Figure 3a, we visually represent the cycle graph with eight nodes and the corresponding $\alpha$-FGL entries. We also refer to Figure 3b, where we

---

[2]We employ the term "fractional graph Laplacian" instead of "fractional symmetrically normalized adjacency" to highlight that, in the singular value domain, one can construct fractional powers of any matrix representations of the underlying graph (commonly referred to as "graph Laplacians"). This fact is not generally true in the eigenvalue domain because eigenvalues can be negative or complex numbers.

[3]In fact, this property can be generalized to all directed graphs with a normal SNA matrix, including, among others, directed cycle graphs. See Lemma C.1 for more details.

depict the distribution of $\alpha$-FGL entries for the real-world graphs Cora (undirected) and Chameleon (directed) with respect to the distance in the original graph. Our empirical findings align with our theoretical results presented in Theorem 4.1.

# 5   Fractional Graph Laplacian Neural ODE

This section explores two fractional Laplacian-based graph neural ODEs. First, we consider the fractional heat equation,

$$\mathbf{x}'(t) = -\mathbf{L}^{\alpha}\mathbf{x}(t)\mathbf{W}\,, \ \mathbf{x}(0) = \mathbf{x}_0\,, \tag{2}$$

where $\mathbf{x}_0 \in \mathbb{R}^{N \times K}$ is the *initial condition*, $\mathbf{x}(t) \in \mathbb{R}^{N \times K}$ for $t > 0$ and $\alpha \in \mathbb{R}$. We assume that the *channel mixing matrix* $\mathbf{W} \in \mathbb{R}^{K \times K}$ is a symmetric matrix. Second, we consider the fractional Schrödinger equation,

$$\mathbf{x}'(t) = i\,\mathbf{L}^{\alpha}\mathbf{x}(t)\mathbf{W}\,, \ \mathbf{x}(0) = \mathbf{x}_0\,, \tag{3}$$

where $\mathbf{x}_0, \mathbf{x}(t) \in \mathbb{C}^{N \times K}$ and $\mathbf{W} \in \mathbb{C}^{K \times K}$ is unitary diagonalizable. Both (2) and (3) can be analytically solved. For instance, the solution of (2) is given by $\mathrm{vec}(\mathbf{x})(t) = \exp(-t\,\mathbf{W} \otimes \mathbf{L}^{\alpha})\mathrm{vec}(\mathbf{x}_0)$, where $\otimes$ denotes the Kronecker product and $\mathrm{vec}(\cdot)$ represents the vectorization operation. However, calculating the exact solution is computationally infeasible since the memory required to store $\mathbf{W} \otimes \mathbf{L}^{\alpha}$ alone grows as $(NK)^2$. Therefore, we rely on numerical schemes to solve (2) and (3).

In the remainder of this section, we analyze the Dirichlet energy for solutions to (2) and (3). We begin with the definition of oversmoothing.

**Definition 5.1.** *Neural ODE-based GNNs are said to* oversmooth *if the normalized Dirichlet energy decays exponentially fast. That is, for any initial value $\mathbf{x}_0$, the solution $\mathbf{x}(t)$ satisfies for every $t > 0$*

$$\left| \mathscr{E}\left( \frac{\mathbf{x}(t)}{\|\mathbf{x}(t)\|_2} \right) - \min \lambda(\mathbf{I} - \mathbf{L}) \right| \leq \exp\left(-Ct\right)\,, \ C > 0\,.$$

Definition 5.1 captures the *actual* smoothness of features by considering the normalized Dirichlet energy, which mitigates the impact of feature amplitude (Cai et al., 2020; Di Giovanni et al., 2023). Additionally, Proposition 3.6 shows that the normalized Dirichlet energy is intimately related to the numerical range of $\mathbf{I} - \mathbf{L}$ of the underlying graph. This shows that the Dirichlet energy and eigenvalues (or *frequencies*) of the SNA are intertwined, and one can equivalently talk about Dirichlet energy or frequencies (see also Lemma D.2). In particular, it holds that

$$0 \leq \mathscr{E}\left( \frac{\mathbf{x}(t)}{\|\mathbf{x}(t)\|_2} \right) \leq \frac{\|\mathbf{I} - \mathbf{L}\|}{2}\,.$$

As seen in Section 3, the minimal possible value attained by the normalized Dirichlet energy is often strictly greater than 0 for directed graphs. This indicates that GNNs on general directed graphs inherently cannot oversmooth to the same extent as in undirected. However, we prove that a vanilla GCN implementing the directed SNA oversmooths with respect to Definition 5.1, see Appendix E.3.

## 5.1   Frequency Analysis for Graphs with Normal SNA

This subsection focuses on the frequency analysis of FGL-based Neural ODEs for undirected graphs. Most classical GNNs (Kipf et al., 2017; Velickovic et al., 2018) and also graph neural ODEs (Chamberlain et al., 2021; Eliasof et al., 2021) have been shown to oversmooth. Di Giovanni et al. (2023) proved that the normalized Dirichlet energy for GNNs based on (2) with $\alpha = 1$ can not only converge to its minimal value but also to its maximal possible value. A GNN exhibiting this property is then termed *Highest-Frequency-Dominant (HFD)*.

However, in real-world scenarios, most graphs are not purely homophilic nor purely heterophilic but fall somewhere in between. Intuitively, this suggests that mid-range frequencies might be more suitable. To illustrate this intuition, consider the cycle graph as an example. If we have a homophily of 1, low frequencies are optimal; with a homophily equal to 0, high frequencies are optimal. Interestingly, for a homophily of $1/2$, the mid-range frequency is optimal, even though the eigendecomposition is label-independent. More information on this example can be found in Figure 4 and Appendix F. Based on this observation, we propose the following definition to generalize the concept of HFD, accommodating not only the lowest or highest frequency but all possible frequencies.

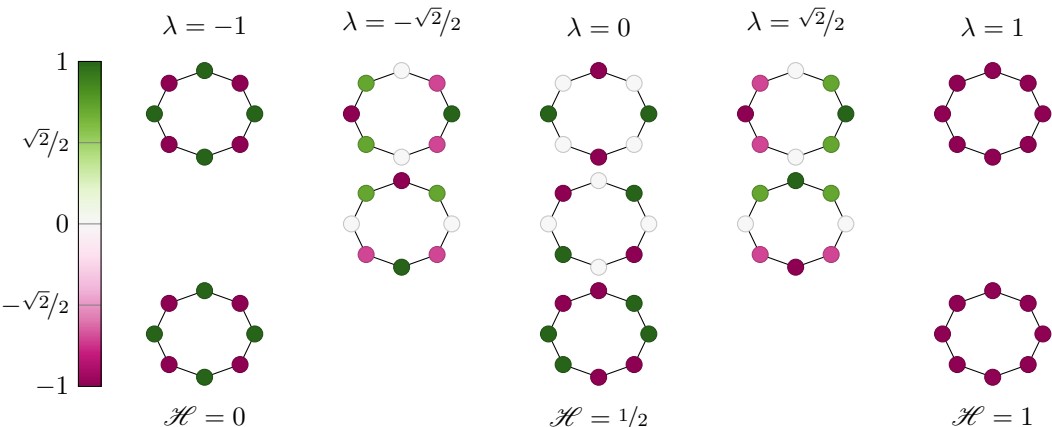

Figure 4: Eigendecomposition of $\mathbf{L}$ for the cycle graph $C_8$ (see Appendix F). The first two rows show the eigenvectors corresponding to the eigenvalues $\lambda$. The last row shows how the (label-unaware) eigendecomposition can be used to study homophily, whose definition requires the labels.

**Definition 5.2.** *Let $\lambda \geq 0$. Neural ODE-based GNNs initialized at $\mathbf{x}_0$ are $\lambda$-Frequency-Dominant ($\lambda$-FD) if the solution $\mathbf{x}(t)$ satisfies*

$$\mathscr{E}\left(\frac{\mathbf{x}(t)}{\|\mathbf{x}(t)\|_2}\right) \xrightarrow{t \to \infty} \frac{\lambda}{2}.$$

*Suppose $\lambda$ is the smallest or the largest eigenvalue with respect to the real part. In that case, we call it Lowest-Frequency-Dominant (LFD) or Highest-Frequency-Dominant (HFD), respectively.*

In the following theorem, we show that (2) and (3) are not limited to being LFD or HFD, but can also be mid-frequency dominant.

**Theorem 5.3.** *Let $\mathcal{G}$ be an undirected graph with SNA $\mathbf{L}$. Consider the initial value problem in (2) with $\mathbf{W} \in \mathbb{R}^{K \times K}$ and $\alpha \in \mathbb{R}$. Then, for almost all initial values $\mathbf{x}_0 \in \mathbb{R}^{N \times K}$ the following holds.*

$(\alpha > 0)$ *The solution to (2) is either HFD or LFD.*

$(\alpha < 0)$ *Let $\lambda_+(\mathbf{L})$ and $\lambda_-(\mathbf{L})$ be the smallest positive and negative non-zero eigenvalue of $\mathbf{L}$, respectively. The solution to (2) is either $(1 - \lambda_+(\mathbf{L}))$-FD or $(1 - \lambda_-(\mathbf{L}))$-FD.*

*Furthermore, the previous results also hold for solutions to the Schrödinger equation (3) if $\mathbf{W} \in \mathbb{C}^{K \times K}$ has at least one eigenvalue with non-zero imaginary part.*

Theorem 5.3 $(\alpha > 0)$ generalizes the result by Di Giovanni et al. (2023) for $\alpha = 1$ to arbitrary positive values of $\alpha$. The convergence speed in Theorem 5.3 $(\alpha > 0)$ depends on the choice of $\alpha \in \mathbb{R}$. By selecting a variable $\alpha$ (e.g., as a learnable parameter), we establish a flexible learning framework capable of adapting the convergence speed of the Dirichlet energy. A slower or more adjustable convergence speed facilitates broader frequency exploration as it converges more gradually to its maximal or minimal value. Consequently, the frequency component contributions (for finite time, i.e., in practice) are better balanced, which is advantageous for graphs with different homophily levels. Theorem 5.3 $(\alpha < 0)$ shows that solutions of the fractional neural ODEs in (2) and (3) are not limited to be LFD or HFD. To demonstrate this and the other results of Theorem 5.3, we solve (2) using an explicit Euler scheme for different choices of $\alpha$ and $\mathbf{W}$ on the Cora and Chameleon graphs. The resulting evolution of the Dirichlet energy with respect to time is illustrated in Figure 5. Finally, we refer to Theorem D.5 in Appendix D.1 for the full statement and proof of Theorem 5.3.

**Remark 5.4.** *Theorem 5.3 is stated for the analytical solutions of (2) and (3), respectively. As noted in Section 5, calculating the analytical solution is infeasible in practice. However, we show in Appendices D.2 to D.3 that approximations of the solution of (2) and (3) via explicit Euler schemes satisfy the same Dirichlet energy convergence properties if the step size is sufficiently small.*

**Remark 5.5.** *Theorem 5.3 can be generalized to all directed graphs with normal SNA, i.e., satisfying the condition $\mathbf{L}\mathbf{L}^\top = \mathbf{L}^\top \mathbf{L}$. For the complete statement, see Appendix D.1.*

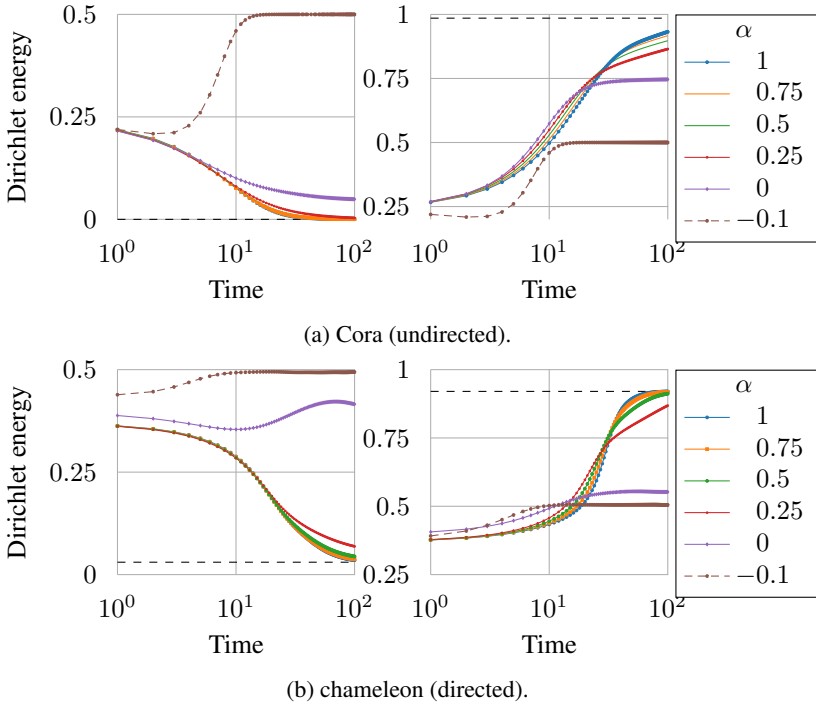

(a) Cora (undirected).

(b) chameleon (directed).

Figure 5: Convergence of Dirichlet energy for the solution of equation (2) using an explicit Euler scheme with a step size of $h = 10^{-1}$. We consider different $\alpha$-FGL in (2) and choose $\mathbf{W}$ as a random diagonal matrix. In the left plot, $\mathbf{W}$ has only a negative spectrum, while in the right plot, $\mathbf{W}$ has only a positive spectrum. The black horizontal line represents the theoretical limit based on Theorem 5.3.

## 5.2 Frequency Dominance for Directed Graphs

Section 5.1 analyzes the Dirichlet energy in graphs with normal SNA. However, the situation becomes significantly more complex when considering generic directed graphs. In our experiments (see Figure 5), we observe that the solution to (2) and (3) does not necessarily lead to oversmoothing. On the contrary, the solution can be controlled to exhibit either LFD or HFD for $\alpha > 0$, and mid-frequency-dominance for $\alpha < 0$ as proven for undirected graphs in Theorem 5.3. We present an initial theoretical result for directed graphs, specifically in the case of $\alpha = 1$.

**Theorem 5.6.** *Let $\mathcal{G}$ be a directed graph with SNA $\mathbf{L}$. Consider the initial value problem in (2) with diagonal channel mixing matrix $\mathbf{W} \in \mathbb{R}^{K \times K}$ and $\alpha = 1$. Suppose $\lambda_1(\mathbf{L})$ is unique. For almost all initial values $\mathbf{x}_0 \in \mathbb{R}^{N \times K}$, the solution to (2) is either HFD or LFD.*

The proof of Theorem 5.6 is given in Appendix E.1. Finally, we refer to Appendix E.2 for the analogous statement and proof when the solution of (2) is approximated via an explicit Euler scheme.

## 6 Numerical Experiments

This section evaluates the fractional Laplacian ODEs in node classification by approximating (2) and (3) with an explicit Euler scheme. This leads to the following update rules

$$\mathbf{x}_{t+1} = \mathbf{x}_t - h\,\mathbf{L}^\alpha \mathbf{x}_t \mathbf{W}\,,\ \ \mathbf{x}_{t+1} = \mathbf{x}_t + i\,h\,\mathbf{L}^\alpha \mathbf{x}_t \mathbf{W}\,, \tag{4}$$

for the heat and Schrödinger equation, respectively. In both cases, $\mathbf{W}$, $\alpha$ and $h$ are learnable parameters, $t$ is the layer, and $\mathbf{x}_0$ is the initial nodes' feature matrix. In accordance with the results in Section 5, we select $\mathbf{W}$ as a diagonal matrix. The initial features $\mathbf{x}_0$ in (4) are encoded through a MLP, and the output is decoded using a second MLP. We refer to the resulting model as *FLODE* (fractional Laplacian ODE). In Appendix A, we present details on the baseline models, the training setup, and the exact hyperparameters.

Table 1: Test accuracy (Film, Squirrel, Chameleon, Citeseer) and test AUROC (Minesweeper, Tolokers, Questions) on node classification, top three models. The thorough comparison is reported in Table 4, Appendix A: FLODE consistently outperforms the baseline models GCN and GRAFF, and it achieves results comparable to state-of-the-art.

(a) Undirected graphs.

|  | Squirrel | Chameleon | Citeseer |
|---|---|---|---|
| 1st | **FLODE** | **FLODE** | **FLODE** |
|  | $64.23 \pm 1.84$ | $73.60 \pm 1.55$ | $78.07 \pm 1.62$ |
| 2nd | GREAD | GREAD | Geom-GCN |
|  | $59.22 \pm 1.44$ | $71.38 \pm 1.30$ | $78.02 \pm 1.15$ |
| 3rd | $\text{GRAFF}_{\text{NL}}$ | $\text{GRAFF}_{\text{NL}}$ | GREAD |
|  | $59.01 \pm 1.31$ | $71.38 \pm 1.47$ | $77.60 \pm 1.81$ |

(b) Heterophily-specific graphs.

|  | Minesweeper | Tolokers | Questions |
|---|---|---|---|
| 1st | GAT-sep | **FLODE** | FSGNN |
|  | $93.91 \pm 0.35$ | $84.17 \pm 0.58$ | $78.86 \pm 0.92$ |
| 2nd | GraphSAGE | GAT-sep | **FLODE** |
|  | $93.51 \pm 0.57$ | $83.78 \pm 0.43$ | $78.39 \pm 1.22$ |
| 3rd | **FLODE** | GAT | GT-sep |
|  | $92.43 \pm 0.51$ | $83.70 \pm 0.47$ | $78.05 \pm 0.93$ |

(c) Directed graphs.

|  | Film | Squirrel | Chameleon |
|---|---|---|---|
| 1st | **FLODE** | HLP | FSGNN |
|  | $37.41 \pm 1.06$ | $74.17 \pm 1.83$ | $78.14 \pm 1.25$ |
| 2nd | GRAFF | **FLODE** | **FLODE** |
|  | $37.11 \pm 1.08$ | $74.03 \pm 1.58$ | $77.98 \pm 1.05$ |
| 3rd | ACM | FSGNN | HLP |
|  | $36.89 \pm 1.18$ | $73.48 \pm 2.13$ | $77.48 \pm 1.50$ |

**Ablation Study.** In Appendix A.3, we investigate the influence of each component (learnable exponent, ODE framework, directionality via the SNA) on the performance of FLODE. The adjustable fractional power in the FGL is a crucial component of FLODE, as it alone outperforms the model employing the ODE framework with a fixed $\alpha = 1$. Further, Appendix A.3 includes ablation studies that demonstrate FLODE's capability to efficiently scale to large depths, as depicted in Figure 8.

**Real-World Graphs.** We report results on 6 undirected datasets consisting of both homophilic graphs, i.e., Cora (McCallum et al., 2000), Citeseer (Sen et al., 2008) and Pubmed (Namata et al., 2012), and heterophilic graphs, i.e., Film (Tang et al., 2009), Squirrel and Chameleon (Rozemberczki et al., 2021). We evaluate our method on the directed and undirected versions of Squirrel, Film, and Chameleon. In all datasets, we use the standard 10 splits from (Pei et al., 2019). The choice of the baseline models and their results are taken from (Di Giovanni et al., 2023). Further, we test our method on heterophily-specific graph datasets, i.e., Roman-empire, Minesweeper, Tolokers, and Questions (Platonov et al., 2023). The splits, baseline models, and results are taken from (Platonov et al., 2023). The top three models are shown in Table 1, and the thorough comparison is reported in Table 4. Due to memory limitations, we compute only 30% of singular values for Pubmed, Roman-Empire, and Questions, which serve as the best low-rank approximation of the original SNA.

**Synthetic Directed Graph.** We consider the directed stochastic block model (DSBM) datasets (Zhang et al., 2021). The DSBM divides nodes into 5 clusters and assigns probabilities for interactions between vertices. It considers two sets of probabilities: $\{\alpha_{i,j}\}$ for undirected edge creation and $\{\beta_{i,j}\}$ for assigning edge directions, $i, j \in \{1, \ldots 5\}$. The objective is to classify vertices based on their clusters. In the first experiment, $\alpha_{i,j} = \alpha^*$ varies, altering neighborhood information's importance. In the second experiment, $\beta_{i,j} = \beta^*$ varies, changing directional information. The results are shown in Figure 6 and Table 6. The splits, baseline models, and results are taken from (Zhang et al., 2021).

**Results.** The experiments showcase the flexibility of FLODE, as it can accommodate various types of graphs, both directed and undirected, as well as a broad range of homophily levels. While other methods, such as MagNet (Zhang et al., 2021), perform similarly to our approach, they face limitations when applied to certain graph configurations. For instance, when applied to undirected graphs, MagNet reduces to ChebNet, making it unsuitable for heterophilic graphs. Similarly, GRAFF

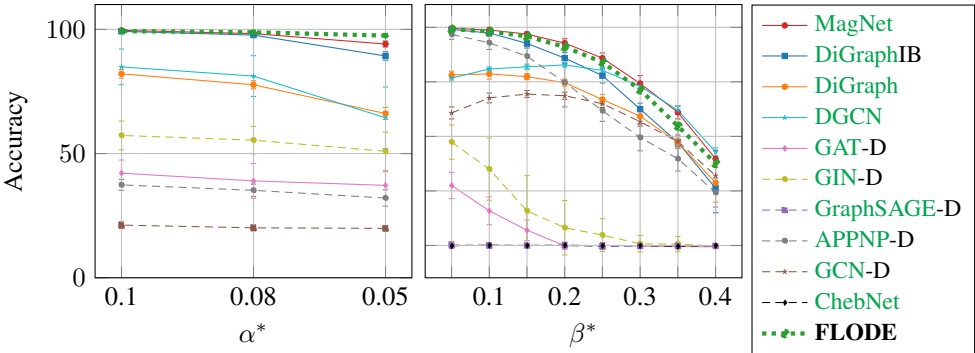

Figure 6: Experiments on directed stochastic block model. Unlike other models, FLODE's performances do not deteriorate as much when changing the inter-cluster edge density $\alpha^*$.

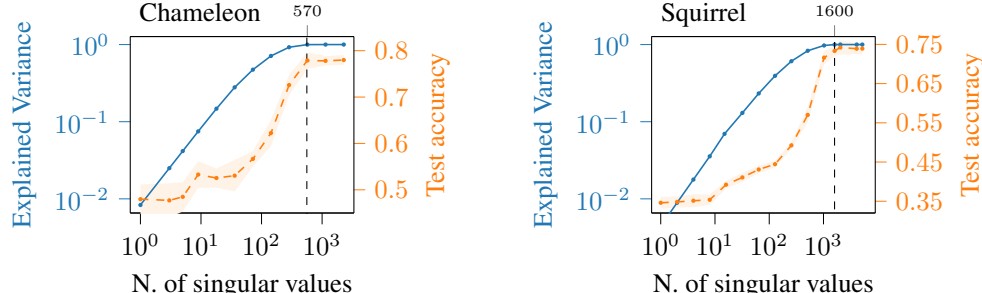

Figure 7: Effect of truncated SVD on test accuracy (orange) for standard directed real-world graphs. The explained variance, defined as $\sum_{i=1}^{k} \sigma_i^2 / \sum_{j=1}^{N} \sigma_j^2$, measures the variability the first $k$ singular values explain. For chameleon, the accuracy stabilized after $570$ ($25\%$) singular values, corresponding to an explained variance of $0.998$. For squirrel, after $1600$ ($31\%$) singular values, which correspond to an explained variance $0.999$, the improvement in test accuracy is only marginal.

(Di Giovanni et al., 2023) performs well on undirected graphs but falls short on directed graphs. We note that oftentimes FLODE learns a non-trivial exponent $\alpha \neq 1$, highlighting the advantages of FGL-based GNNs (see, e.g., Table 5). Furthermore, as shown in Table 9 and Appendix A.3, our empirical results align closely with the theoretical results in Section 5.

# 7 Conclusion

In this work, we introduce the concepts of Dirichlet energy and oversmoothing for directed graphs and demonstrate their relation with the SNA. Building upon this foundation, we define fractional graph Laplacians in the singular value domain, resulting in matrices capable of capturing long-range dependencies. To address oversmoothing in directed graphs, we propose fractional Laplacian-based graph ODEs, which are provably not limited to LFD behavior. We finally show the flexibility of our method to accommodate various graph structures and homophily levels in node-level tasks.

**Limitations and Future Work.** The computational cost of the SVD grows cubically in $N$, while the storage of the singular vectors grows quadratically in $N$. Both costs can be significantly reduced by computing only $k \ll N$ singular values via truncated SVD (Figure 7), giving the best $k$-rank approximation of the SNA. Moreover, the SVD can be computed offline as a preprocessing step.

The frequency analysis of $\alpha$-FGL neural ODEs in directed graphs is an exciting future direction. It would also be worthwhile to investigate the impact of choosing $\alpha \neq 1$ on the convergence speed of the Dirichlet energy. Controlling the speed could facilitate the convergence of the Dirichlet energy to an *optimal* value, which has been shown to exist in synthetic settings (Keriven, 2022; X. Wu et al., 2022). Another interesting future direction would be to analyze the dynamics when approximating the solution to the FGL neural ODEs using alternative numerical solvers, such as adjoint methods.

## Acknowledgments

S. M. acknowledges partial support by the NSF-Simons Research Collaboration on the Mathematical and Scientific Foundations of Deep Learning (MoDL) (NSF DMS 2031985) and DFG SPP 1798, KU 1446/27-2.

G. K. acknowledges partial support by the DAAD programme Konrad Zuse Schools of Excellence in Artificial Intelligence, sponsored by the Federal Ministry of Education and Research. G. Kutyniok also acknowledges support from the Munich Center for Machine Learning (MCML) as well as the German Research Foundation under Grants DFG-SPP-2298, KU 1446/31-1 and KU 1446/32-1 and under Grant DFG-SFB/TR 109 and Project C09.

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

# Acronyms

| | |
|---|---|
| AUROC | Area under the ROC curve |
| DSBM | Directed Stochastic Block Model |
| FD | Frequency Dominant |
| FGL | Fractional Graph Laplacian |
| GAT | Graph Attention Network |
| GCN | Graph Convolutional Network |
| GNN | Graph Neural Network |
| HFD | Highest Frequency Dominant |
| LCC | Largest Connected Components |
| LFD | Lowest Frequency Dominant |
| MLP | Multi-Layer Perceptron |
| ODE | Ordinary Differential Equation |
| SNA | Symmetrically Normalized Adjacency |
| SVD | Singular Value Decomposition |

# Notation

| | |
|---|---|
| $i$ | Imaginary unit |
| $\Re(z)$ | Real part of $z \in \mathbb{C}$ |
| $\Im(z)$ | Imaginary part of $z \in \mathbb{C}$ |
| $\operatorname{diag}(\mathbf{x})$ | Diagonal matrix with $\mathbf{x}$ on the diagonal. |
| $\mathbf{1}$ | Constant vector of all 1s. |
| $\mathbf{M}^{\mathsf{T}}$ | Transpose of $\mathbf{M}$ |
| $\mathbf{M}^{*}$ | Conjugate of $\mathbf{M}$ |
| $\mathbf{M}^{\mathsf{H}}$ | Conjugate transpose of $\mathbf{M}$ |
| $\|\mathbf{M}\|$ | Spectral norm of $\mathbf{M}$ |
| $\|\mathbf{M}\|_2$ | Frobenius norm of $\mathbf{M}$ |
| $\lambda(\mathbf{M})$ | Spectrum of $\mathbf{M}$ |
| $\sigma(\mathbf{M})$ | Singular values of $\mathbf{M}$ |
| $\mathscr{E}(\mathbf{x})$ | Dirichlet energy computed on $\mathbf{x}$ |
| $\mathscr{H}(\mathcal{G})$ | Homophily coefficient of the graph $\mathcal{G}$ |
| $\mathbf{A} \otimes \mathbf{B}$ | Kronecker product between $\mathbf{A}$ and $\mathbf{B}$ |
| $\operatorname{vec}(\mathbf{M})$ | Vector obtained stacking columns of $\mathbf{M}$. |

# A  Implementation Details

In this section, we give the details on the numerical results in Section 6. We begin by describing the exact model.

**Model architecture.** Let $\mathcal{G}$ be a directed graphs and $\mathbf{x}_0 \in \mathbb{R}^{N \times K}$ the node features. Our architecture first embeds the input node features $\mathbf{x}$ via a multi-layer perceptron (MLP). We then evolve the features $\mathbf{x}_0$ according to a slightly modified version of (3), i.e, $\mathbf{x}'(t) = -i\, \mathbf{L}^\alpha \mathbf{x}(t) \mathbf{W}$ for some time $t \in [0, T]$. In our experiments, we approximate the solution with an Explicit Euler scheme with step size $h > 0$. This leads to the following update rule

$$\mathbf{x}_{t+1} = \mathbf{x}_t - ih\mathbf{L}^\alpha \mathbf{x}_t \mathbf{W}\,.$$

The channel mixing matrix is a diagonal learnable matrix $\mathbf{W} \in \mathbb{C}^{K \times K}$, and $\alpha \in \mathbb{R}$, $h \in \mathbb{C}$ are also learnable parameters. The features at the last time step $\mathbf{x}_T$ are then fed into a second MLP, whose output is used as the final output. Both MLPs use LeakyReLU as non-linearity and dropout (Srivastava et al., 2014). On the contrary, the graph layers do not use any dropout nor non-linearity. A sketch of the algorithm is reported in fLode.

---

**Algorithm 1:** fLode

```
% A, x₀ are given.
% Preprocessing
```
1   $\mathbf{D}_{\text{in}}\ = \text{diag}\,(\mathbf{A}\mathbf{1})$
2   $\mathbf{D}_{\text{out}} = \text{diag}\,(\mathbf{A}^\mathsf{T}\mathbf{1})$
3   $\mathbf{L}\ = \mathbf{D}_{\text{in}}^{-1/2}\mathbf{A}\mathbf{D}_{\text{out}}^{-1/2}$
4   $\mathbf{U}, \mathbf{\Sigma}, \mathbf{V}^{\mathsf{H}}\ = \text{svd}(\mathbf{L})$
```
% The core of the algorithm is very simple
```
5   `def` training_step($\mathbf{x}_0$):
6    $\mathbf{x}_0 = $ input_MLP($\mathbf{x}_0$)
7    `for` $t \in \{1, \ldots, T\}$ `do`
8     $\mathbf{x}_t\ = \mathbf{x}_{t-1} - i\,h\,\mathbf{U}\mathbf{\Sigma}^\alpha \mathbf{V}^\mathsf{H}\mathbf{x}_{t-1}\mathbf{W}$
9    $\mathbf{x}_T = $ output_MLP($\mathbf{x}_T$)
10   `return` $\mathbf{x}_T$

---

**Complexity.** The computation of the SVD is $\mathcal{O}(N^3)$. However, one can compute only the first $p \ll N$ singular values: this cuts down the cost to $\mathcal{O}(N^2\,p)$. The memory required to store the singular vectors is $\mathcal{O}(N^2)$, since they are not sparse in general. Each training step has a cost of $\mathcal{O}(N^2\,K)$.

**Experimental details.** Our model is implemented in `PyTorch` (Paszke et al., 2019), using `PyTorch geometric` (Fey et al., 2019). The computation of the SVD for the fractional Laplacian is implemented using the library `linalg` provided by `PyTorch`. In the case of truncated SVD, we use the function `randomized_svd` provided by the library `extmath` from `sklearn`. The code and instructions to reproduce the experiments are available on GitHub. Hyperparameters were tuned using grid search. All experiments were run on an internal cluster with `NVIDIA GeForce RTX 2080 Ti` and `NVIDIA TITAN RTX` GPUs with 16 and 24 GB of memory, respectively.

**Training details.** All models were trained for 1000 epochs using Adam (Kingma et al., 2015) as optimizer with a fixed learning rate. We perform early stopping if the validation metric does not increase for 200 epochs.

## A.1  Real-World Graphs

**Undirected graphs** We conducted 10 repetitions using data splits obtained from (Pei et al., 2019). For each split, 48% of the nodes are used for training, 32% for validation and 20% for testing. In all datasets, we considered the largest connected component (LCC). Chameleon, Squirrel, and Film are

directed graphs; hence, we converted them to undirected. Cora, Citeseer, and Pubmed are already undirected graphs: to these, we added self-loops. We normalized the input node features for all graphs.

As baseline models, we considered the same models as in (Di Giovanni et al., 2023). The results were provided by Pei et al. (2019) and include standard GNNs, such as GAT (Velickovic et al., 2018), GCN (Kipf et al., 2017), and GraphSAGE (Hamilton et al., 2017). We also included models designed to address oversmoothing and heterophilic graphs, such as PairNorm (L. Zhao et al., 2019), GGCN (Yan et al., 2022), Geom-GCN (Pei et al., 2019), $H_2$GCN (Zhu, Yan, et al., 2020), GPRGNN (Chien et al., 2021), and Sheaf (Bodnar et al., 2022). Furthermore, we included the graph neural ODE-based approaches, CGNN (Xhonneux et al., 2020) and GRAND (Chamberlain et al., 2021), as in (Di Giovanni et al., 2023), and the model GRAFF from (Di Giovanni et al., 2023) itself. Finally, we included GREAD (Choi et al., 2023), GraphCON (Rusch et al., 2022), ACMP (Y. Wang et al., 2022) and GCN and GAT equipped with DropEdge (Rong et al., 2020).

**Heterophily-specific Models**  For heterophily-specific datasets, we use the same models and results as in (Platonov et al., 2023). As baseline models we considered the topology-agnostic ResNet (He et al., 2016) and two graph-aware modifications: ResNet+SGC(F. Wu et al., 2019) where the initial node features are multiplied by powers of the SNA, and ResNet+adj, where rows of the adjacency matrix are used as additional node features; GCN (Kipf et al., 2017), GraphSAGE (Hamilton et al., 2017); GAT (Velickovic et al., 2018) and GT (Shi et al., 2021) as well as their modification GAT-sep and GT-sep which separate ego- and neighbor embeddings; $H_2$GCN (Zhu, Yan, et al., 2020), CPGNN (Zhu, Rossi, et al., 2021), GPRGNN (Chien et al., 2021), FSGNN (Maurya et al., 2021), GloGNN (X. Li et al., 2022), FAGCN (Bo et al., 2021), GBK-GNN (Du et al., 2022), and JacobiConv (X. Wang et al., 2022).

The exact hyperparameters for FLODE are provided in Table 5.

## A.2 Synthetic Directed Graphs

The dataset and code are taken from (Zhang et al., 2021). As baseline models, we considered the ones in (Zhang et al., 2021) for which we report the corresponding results. The baseline models include standard GNNs, such as ChebNet (Defferrard et al., 2016), GCN (Kipf et al., 2017), GraphSAGE (Hamilton et al., 2017), APPNP (Gasteiger et al., 2018), GIN (Xu et al., 2018), GAT (Velickovic et al., 2018), but also models specifically designed for directed graphs, such as DGCN (Tong, Liang, Sun, Rosenblum, et al., 2020), DiGraph and DiGraphIB (Tong, Liang, Sun, X. Li, et al., 2020), MagNet (Zhang et al., 2021)).

**The DSBM dataset.**  The directed stochastic block model (DSBM) is described in detail in (Zhang et al., 2021, Section 5.1.1). To be self-contained, we include a short explanation.

The DSBM model is defined as follows. There are $N$ vertices, which are divided into $n_c$ clusters $(C_1, C_2, ...C_{n_c})$, each having an equal number of vertices. An interaction is defined between any two distinct vertices, $u$ and $v$, based on two sets of probabilities: $\{\alpha_{i,j}\}_{i,j=1}^{n_c}$ and $\{\beta_{i,j}\}_{i,j=1}^{n_c}$.

The set of probabilities $\{\alpha_{i,j}\}$ is used to create an undirected edge between any two vertices $u$ and $v$, where $u$ belongs to cluster $C_i$ and $v$ belongs to cluster $C_j$. The key property of this probability set is that $\alpha_{i,j} = \alpha_{j,i}$, which means the chance of forming an edge between two clusters is the same in either direction.

The set of probabilities $\{\beta_{i,j}\}$ is used to assign a direction to the undirected edges. For all $i, j \in \{1, \ldots, n_c\}$, we assume that $\beta_{i,j} + \beta_{j,i} = 1$ holds. Then, to the undirected edge $(u, v)$ is assigned the direction from $u$ to $v$ with probability $\beta_{i,j}$ if $u$ belongs to cluster $C_i$ and $v$ belongs to cluster $C_j$, and the direction from $v$ to $u$ with probability $\beta_{j,i}$.

The primary objective here is to classify the vertices based on their respective clusters.

There are several scenarios designed to test different aspects of the baseline models and our model. In the experiments, the total number of nodes is fixed at $N = 2500$ and the number of clusters is fixed at $n_c = 5$. In all experiments, the training set contains 20 nodes per cluster, 500 nodes for validation, and the rest for testing. The results are averaged over 5 different seeds and splits.

Table 4: Test accuracy on node classification: top three models indicated as 1st , 2nd , 3rd.

(a) Undirected graphs.

| | Film | Squirrel | Chameleon | Citeseer | Pubmed | Cora |
|---|---|---|---|---|---|---|
| GGCN | 37.54 ± 1.56 | 55.17 ± 1.58 | 71.14 ± 1.84 | 77.14 ± 1.45 | 89.15 ± 0.37 | 87.95 ± 1.05 |
| GPRGNN | 34.63 ± 1.22 | 31.61 ± 1.24 | 46.58 ± 1.71 | 77.13 ± 1.67 | 87.54 ± 0.38 | 87.95 ± 1.18 |
| FAGCN | 35.70 ± 1.00 | 36.48 ± 1.86 | 60.11 ± 2.15 | 77.11 ± 1.57 | 89.49 ± 0.38 | 87.87 ± 1.20 |
| GCNII | 37.44 ± 1.30 | 38.47 ± 1.58 | 63.86 ± 3.04 | 77.33 ± 1.48 | 90.15 ± 0.43 | 88.37 ± 1.25 |
| Geom-GCN | 31.59 ± 1.15 | 38.15 ± 0.92 | 60.00 ± 2.81 | 78.02 ± 1.15 | 89.95 ± 0.47 | 85.35 ± 1.57 |
| PairNorm | 27.40 ± 1.24 | 50.44 ± 2.04 | 62.74 ± 2.82 | 73.59 ± 1.47 | 87.53 ± 0.44 | 85.79 ± 1.01 |
| GraphSAGE | 34.23 ± 0.99 | 41.61 ± 0.74 | 58.73 ± 1.68 | 76.04 ± 1.30 | 88.45 ± 0.50 | 86.90 ± 1.04 |
| GCN | 27.32 ± 1.10 | 53.43 ± 2.01 | 64.82 ± 2.24 | 76.50 ± 1.36 | 88.42 ± 0.50 | 86.98 ± 1.27 |
| GAT | 27.44 ± 0.89 | 40.72 ± 1.55 | 60.26 ± 2.50 | 76.55 ± 1.23 | 87.30 ± 1.10 | 86.33 ± 0.48 |
| MLP | 36.53 ± 0.70 | 28.77 ± 1.56 | 46.21 ± 2.99 | 74.02 ± 1.90 | 75.69 ± 2.00 | 87.16 ± 0.37 |
| CGNN | 35.95 ± 0.86 | 29.24 ± 1.09 | 46.89 ± 1.66 | 76.91 ± 1.81 | 87.70 ± 0.49 | 87.10 ± 1.35 |
| GRAND | 35.62 ± 1.01 | 40.05 ± 1.50 | 54.67 ± 2.54 | 76.46 ± 1.77 | 89.02 ± 0.51 | 87.36 ± 0.96 |
| Sheaf (max) | 37.81 ± 1.15 | 56.34 ± 1.32 | 68.04 ± 1.58 | 76.70 ± 1.57 | 89.49 ± 0.40 | 86.90 ± 1.13 |
| GRAFF$_{NL}$ | 35.96 ± 0.95 | 59.01 ± 1.31 | 71.38 ± 1.47 | 76.81 ± 1.12 | 89.81 ± 0.50 | 87.81 ± 1.13 |
| GREAD | 37.90 ± 1.17 | 59.22 ± 1.44 | 71.38 ± 1.30 | 77.60 ± 1.81 | 90.23 ± 0.55 | 88.57 ± 0.66 |
| GraphCON | 35.58 ± 1.24 | 35.51 ± 1.40 | 49.63 ± 1.89 | 76.36 ± 2.67 | 88.01 ± 0.47 | 87.22 ± 1.48 |
| ACMP | 34.93 ± 1.26 | 40.05 ± 1.53 | 57.59 ± 2.09 | 76.71 ± 1.77 | 87.79 ± 0.47 | 87.71 ± 0.95 |
| GCN+DropEdge | 29.93 ± 0.80 | 41.30 ± 1.77 | 59.06 ± 2.04 | 76.57 ± 2.68 | 86.97 ± 0.42 | 83.54 ± 1.06 |
| GAT+DropEdge | 28.95 ± 0.76 | 41.27 ± 1.76 | 58.95 ± 2.13 | 76.13 ± 2.20 | 86.91 ± 0.45 | 83.54 ± 1.06 |
| **FLODE** | 37.16 ± 1.42 | 64.23 ± 1.84 | 73.60 ± 1.55 | 78.07 ± 1.62 | 89.02 ± 0.38 | 86.44 ± 1.17 |

(b) Directed graphs.

| | Film | Squirrel | Chameleon |
|---|---|---|---|
| ACM | 36.89 ± 1.18 | 54.4 ± 1.88 | 67.08 ± 2.04 |
| HLP | 34.59 ± 1.32 | 74.17 ± 1.83 | 77.48 ± 1.50 |
| FSGNN | 35.67 ± 0.69 | 73.48 ± 2.13 | 78.14 ± 1.25 |
| GRAFF | 37.11 ± 1.08 | 58.72 ± 0.84 | 71.08 ± 1.75 |
| **FLODE** | 37.41 ± 1.06 | 74.03 ± 1.58 | 77.98 ± 1.05 |

(c) Heterophily-specific graphs. For Minesweeper, Tolokers and Questions the evaluation metric is the AUROC.

| | Roman-empire | Minesweeper | Tolokers | Questions |
|---|---|---|---|---|
| ResNet | 65.88 ± 0.38 | 50.89 ± 1.39 | 72.95 ± 1.06 | 70.34 ± 0.76 |
| ResNet+SGC | 73.90 ± 0.51 | 70.88 ± 0.90 | 80.70 ± 0.97 | 75.81 ± 0.96 |
| ResNet+adj | 52.25 ± 0.40 | 50.42 ± 0.83 | 78.78 ± 1.11 | 75.77 ± 1.24 |
| GCN | 73.69 ± 0.74 | 89.75 ± 0.52 | 83.64 ± 0.67 | 76.09 ± 1.27 |
| GraphSAGE | 85.74 ± 0.67 | 93.51 ± 0.57 | 82.43 ± 0.44 | 76.44 ± 0.62 |
| GAT | 80.87 ± 0.30 | 92.01 ± 0.68 | 83.70 ± 0.47 | 77.43 ± 1.20 |
| GAT-sep | 88.75 ± 0.41 | 93.91 ± 0.35 | 83.78 ± 0.43 | 76.79 ± 0.71 |
| GT | 86.51 ± 0.73 | 91.85 ± 0.76 | 83.23 ± 0.64 | 77.95 ± 0.68 |
| GT-sep | 87.32 ± 0.39 | 92.29 ± 0.47 | 82.52 ± 0.92 | 78.05 ± 0.93 |
| FAGCN | 60.11 ± 0.52 | 89.71 ± 0.31 | 73.35 ± 1.01 | 63.59 ± 1.46 |
| CPGNN | 63.96 ± 0.62 | 52.03 ± 5.46 | 73.36 ± 1.01 | 65.96 ± 1.95 |
| H$_2$GCN | 64.85 ± 0.27 | 86.24 ± 0.61 | 72.94 ± 0.97 | 55.48 ± 0.91 |
| FSGNN | 79.92 ± 0.56 | 90.08 ± 0.70 | 82.76 ± 0.61 | 78.86 ± 0.92 |
| GloGNN | 59.63 ± 0.69 | 51.08 ± 1.23 | 73.39 ± 1.17 | 65.74 ± 1.19 |
| FAGCN | 65.22 ± 0.56 | 88.17 ± 0.73 | 77.75 ± 1.05 | 77.24 ± 1.26 |
| GBK-GNN | 74.57 ± 0.47 | 90.85 ± 0.58 | 81.01 ± 0.67 | 74.47 ± 0.86 |
| JacobiConv | 71.14 ± 0.42 | 89.66 ± 0.40 | 68.66 ± 0.65 | 73.88 ± 1.16 |
| **FLODE** | 74.97 ± 0.53 | 92.43 ± 0.51 | 84.17 ± 0.58 | 78.39 ± 1.22 |

Table 5: Selected hyperparameters, learned exponent, step size, and Dirichlet energy in the last layer for real-world datasets.

(a) Undirected.

| | Dataset | | | | | |
|---|---|---|---|---|---|---|
| | **Film** | **Squirrel** | **Chameleon** | **Citeseer** | **Pubmed** | **Cora** |
| learning rate | $10^{-3}$ | $2.5 \cdot 10^{-3}$ | $5 \cdot 10^{-3}$ | $10^{-2}$ | $10^{-2}$ | $10^{-2}$ |
| weight decay | $5 \cdot 10^{-4}$ | $5 \cdot 10^{-4}$ | $10^{-3}$ | $5 \cdot 10^{-3}$ | $10^{-3}$ | $5 \cdot 10^{-3}$ |
| hidden channels | 256 | 64 | 64 | 64 | 64 | 64 |
| num. layers | 1 | 6 | 4 | 2 | 3 | 2 |
| encoder layers | 3 | 1 | 1 | 1 | 3 | 1 |
| decoder layers | 2 | 2 | 2 | 1 | 1 | 2 |
| input dropout | 0 | $1.5 \cdot 10^{-1}$ | 0 | 0 | $5 \cdot 10^{-2}$ | 0 |
| decoder dropout | $10^{-1}$ | $10^{-1}$ | 0 | 0 | $10^{-1}$ | 0 |
| exponent | $1.001 \pm 0.003$ | $0.17 \pm 0.03$ | $0.35 \pm 0.15$ | $0.92 \pm 0.03$ | $0.82 \pm 0.07$ | $0.90 \pm 0.02$ |
| step size | $0.991 \pm 0.002$ | $1.08 \pm 0.01$ | $1.22 \pm 0.03$ | $1.04 \pm 0.02$ | $1.12 \pm 0.02$ | $1.06 \pm 0.01$ |
| Dirichlet energy | $0.246 \pm 0.006$ | $0.40 \pm 0.02$ | $0.13 \pm 0.03$ | $0.021 \pm 0.001$ | $0.015 \pm 0.001$ | $0.0227 \pm 0.0006$ |

(b) Directed.

| | Dataset | | |
|---|---|---|---|
| | **Film** | **Squirrel** | **Chameleon** |
| learning rate | $10^{-3}$ | $2.5 \cdot 10^{-3}$ | $10^{-2}$ |
| weight decay | $5 \cdot 10^{-4}$ | $5 \cdot 10^{-4}$ | $10^{-3}$ |
| hidden channels | 256 | 64 | 64 |
| num. layers | 1 | 6 | 5 |
| encoder layers | 3 | 1 | 1 |
| decoder layers | 2 | 2 | 2 |
| input dropout | 0 | $10^{-1}$ | 0 |
| decoder dropout | 0.1 | $10^{-1}$ | 0 |
| exponent | $1.001 \pm 0.005$ | $0.28 \pm 0.06$ | $0.30 \pm 0.11$ |
| step size | $0.990 \pm 0.002$ | $1.22 \pm 0.02$ | $1.22 \pm 0.05$ |
| Dirichlet energy | $0.316 \pm 0.005$ | $0.38 \pm 0.02$ | $0.27 \pm 0.04$ |

(c) Heterophily-specific graphs.

| | Dataset | | | |
|---|---|---|---|---|
| | **Roman-empire** | **Minesweeper** | **Tolokers** | **Questions** |
| learning rate | $10^{-3}$ | $10^{-3}$ | $10^{-3}$ | $10^{-2}$ |
| weight decay | 0 | 0 | 0 | $5 \cdot 10^{-4}$ |
| hidden channels | 512 | 512 | 512 | 128 |
| num. layers | 4 | 4 | 4 | 5 |
| encoder layers | 2 | 2 | 1 | 2 |
| decoder layers | 2 | 2 | 2 | 2 |
| input dropout | 0 | 0 | 0 | 0 |
| decoder dropout | 0 | 0 | 0 | 0 |
| exponent | $0.689 \pm 0.038$ | $0.749 \pm 0.017$ | $1.053 \pm 0.041$ | $1.090 \pm 0.046$ |
| step size | $0.933 \pm 0.015$ | $0.984 \pm 0.004$ | $0.993 \pm 0.009$ | $0.789 \pm 0.062$ |
| Dirichlet energy | $0.059 \pm 0.003$ | $0.173 \pm 0.019$ | $0.155 \pm 0.013$ | $0.092 \pm 0.039$ |

Table 6: Node classification accuracy of ordered DSBM graphs: top three models as 1st, 2nd and 3rd.

(a) Varying edge density.

| | $\alpha^*$ | | |
| --- | --- | --- | --- |
| | 0.1 | 0.08 | 0.05 |
| ChebNet | $19.9 \pm 0.6$ | $20.0 \pm 0.7$ | $20.0 \pm 0.7$ |
| GCN-D | $68.9 \pm 2.1$ | $67.6 \pm 2.7$ | $58.5 \pm 2.0$ |
| APPNP-D | $97.7 \pm 1.7$ | $95.9 \pm 2.2$ | $90.3 \pm 2.4$ |
| GraphSAGE-D | $20.1 \pm 1.1$ | $19.9 \pm 0.8$ | $19.9 \pm 1.0$ |
| GIN-D | $57.3 \pm 5.8$ | $55.4 \pm 5.5$ | $50.9 \pm 7.7$ |
| GAT-D | $42.1 \pm 5.3$ | $39.0 \pm 7.0$ | $37.2 \pm 5.5$ |
| DGCN | $84.9 \pm 7.2$ | $81.2 \pm 8.2$ | $64.4 \pm 12.4$ |
| DiGraph | $82.1 \pm 1.7$ | $77.7 \pm 1.6$ | $66.1 \pm 2.4$ |
| DiGraphIB | $99.2 \pm 0.5$ | $97.7 \pm 0.7$ | $89.3 \pm 1.7$ |
| MagNet | $99.6 \pm 0.2$ | $98.3 \pm 0.8$ | $94.1 \pm 1.2$ |
| **FLODE** | $99.3 \pm 0.1$ | $98.8 \pm 0.1$ | $97.5 \pm 0.1$ |

(b) Varying net flow.

| | $\beta^*$ | | | | | | | |
| --- | --- | --- | --- | --- | --- | --- | --- | --- |
| | 0.05 | 0.10 | 0.15 | 0.20 | 0.25 | 0.30 | 0.35 | 0.40 |
| ChebNet | $19.9 \pm 0.7$ | $20.1 \pm 0.6$ | $20.0 \pm 0.6$ | $20.1 \pm 0.8$ | $19.9 \pm 0.9$ | $20.0 \pm 0.5$ | $19.7 \pm 0.9$ | $20.0 \pm 0.5$ |
| GCN-D | $68.6 \pm 2.2$ | $74.1 \pm 1.8$ | $75.5 \pm 1.3$ | $74.9 \pm 1.3$ | $72.0 \pm 1.4$ | $65.4 \pm 1.6$ | $58.1 \pm 2.4$ | $45.6 \pm 4.7$ |
| APPNP-D | $97.4 \pm 1.8$ | $94.3 \pm 2.4$ | $89.4 \pm 3.6$ | $79.8 \pm 9.0$ | $69.4 \pm 3.9$ | $59.6 \pm 4.9$ | $51.8 \pm 4.5$ | $39.4 \pm 5.3$ |
| GraphSAGE-D | $20.2 \pm 1.2$ | $20.0 \pm 1.0$ | $20.0 \pm 0.8$ | $20.0 \pm 0.7$ | $19.6 \pm 0.9$ | $19.8 \pm 0.7$ | $19.9 \pm 0.9$ | $19.9 \pm 0.8$ |
| GIN-D | $57.9 \pm 6.3$ | $48.0 \pm 11.4$ | $32.7 \pm 12.9$ | $26.5 \pm 10.0$ | $23.8 \pm 6.0$ | $20.6 \pm 3.0$ | $20.5 \pm 2.8$ | $19.8 \pm 0.5$ |
| GAT-D | $42.0 \pm 4.8$ | $32.7 \pm 5.1$ | $25.6 \pm 3.8$ | $19.9 \pm 1.4$ | $20.0 \pm 1.0$ | $19.8 \pm 0.8$ | $19.6 \pm 0.2$ | $19.5 \pm 0.2$ |
| DGCN | $81.4 \pm 1.1$ | $84.7 \pm 0.7$ | $85.5 \pm 1.0$ | $86.2 \pm 0.8$ | $84.2 \pm 1.1$ | $78.4 \pm 1.3$ | $69.6 \pm 1.5$ | $54.3 \pm 1.5$ |
| DiGraph | $82.5 \pm 1.4$ | $82.9 \pm 1.9$ | $81.9 \pm 1.1$ | $79.7 \pm 1.3$ | $73.5 \pm 1.9$ | $67.4 \pm 2.8$ | $57.8 \pm 1.6$ | $43.0 \pm 7.1$ |
| DiGraphIB | $99.2 \pm 0.4$ | $97.9 \pm 0.6$ | $94.1 \pm 1.7$ | $88.7 \pm 2.0$ | $82.3 \pm 2.7$ | $70.0 \pm 2.2$ | $57.8 \pm 6.4$ | $41.0 \pm 9.0$ |
| MagNet | $99.6 \pm 0.2$ | $99.0 \pm 1.0$ | $97.5 \pm 0.8$ | $94.2 \pm 1.6$ | $88.7 \pm 1.9$ | $79.4 \pm 2.9$ | $68.8 \pm 2.4$ | $51.8 \pm 3.1$ |
| **FLODE** | $99.3 \pm 0.1$ | $98.5 \pm 0.1$ | $96.7 \pm 0.2$ | $92.8 \pm 0.1$ | $87.2 \pm 0.3$ | $77.1 \pm 0.5$ | $63.8 \pm 0.3$ | $50.1 \pm 0.5$ |

Following Zhang et al. (2021), we train our model in both experiments for 3000 epochs and use early-stopping if the validation accuracy does not increase for 500 epochs. We select the best model based on the validation accuracy after sweeping over a few hyperparameters. We give exact numerical values for the experiments with the standard error in Table 6a and refer to Appendix A.2 for the chosen hyperparameters.

**DSBM with varying edge density.** In the first experiment, the model is evaluated based on its performance on the DSBM with varying $\alpha_{i,j} = \alpha^*$, $\alpha^* \in \{0.1, 0.08, 0.05\}$ for $i \neq j$, which essentially changes the density of edges between different clusters. The other probabilities are fixed at $\alpha_{i,i} = 0.5$, $\beta_{i,i} = 0.5$ and $\beta_{i,j} = 0.05$ for $i > j$. The results are shown in Figure 6 with exact numerical values in Table 6a.

**DSBM with varying net flow.** In the other scenario, the model is tested on how it performs when the net flow from one cluster to another varies. This is achieved by keeping $\alpha_{i,j} = 0.1$ constant for all $i$ and $j$, and allowing $\beta_{i,j}$ to vary from 0.05 to 0.4. The other probabilities are fixed at $\alpha_{i,i} = 0.5$ and $\beta_{i,i} = 0.5$. The results are shown in Figure 6 with exact numerical values in Table 6b.

### A.3 Ablation Study

We perform an ablation study on Chameleon and Squirrel (directed, heterophilic), and Citeseer (undirected, homophilic). For this, we sweep over different model options using the same hyperparameters

Table 7: Selected hyperparameters for DSBM dataset.

(a) Varying edge density.

| | $\alpha^*$ | | |
| --- | --- | --- | --- |
| | 0.1 | 0.08 | 0.05 |
| learning rate | $5 \cdot 10^{-3}$ | $5 \cdot 10^{-3}$ | $5 \cdot 10^{-3}$ |
| decay | $1 \cdot 10^{-3}$ | $1 \cdot 10^{-3}$ | $5 \cdot 10^{-4}$ |
| input dropout | $1 \cdot 10^{-1}$ | $2 \cdot 10^{-1}$ | $1 \cdot 10^{-1}$ |
| decoder dropout | $1 \cdot 10^{-1}$ | $5 \cdot 10^{-2}$ | $1 \cdot 10^{-1}$ |
| hidden channels | 256 | 256 | 256 |

(b) Varying net flow.

| | $\beta^*$ | | | | | | | |
| --- | --- | --- | --- | --- | --- | --- | --- | --- |
| | 0.05 | 0.1 | 0.15 | 0.2 | 0.25 | 0.3 | 0.35 | 0.4 |
| learning rate | $5 \cdot 10^{-3}$ | $5 \cdot 10^{-3}$ | $1 \cdot 10^{-3}$ | $1 \cdot 10^{-3}$ | $1 \cdot 10^{-3}$ | $1 \cdot 10^{-3}$ | $1 \cdot 10^{-3}$ | $1 \cdot 10^{-3}$ |
| decay | $1 \cdot 10^{-3}$ | $1 \cdot 10^{-3}$ | $5 \cdot 10^{-4}$ | $1 \cdot 10^{-3}$ | $1 \cdot 10^{-3}$ | $1 \cdot 10^{-3}$ | $5 \cdot 10^{-4}$ | $1 \cdot 10^{-3}$ |
| input dropout | $1 \cdot 10^{-1}$ | $1 \cdot 10^{-1}$ | $2 \cdot 10^{-1}$ | $1 \cdot 10^{-1}$ | $1 \cdot 10^{-1}$ | $2 \cdot 10^{-1}$ | $5 \cdot 10^{-2}$ | $2 \cdot 10^{-1}$ |
| decoder dropout | $1 \cdot 10^{-1}$ | $1 \cdot 10^{-1}$ | $5 \cdot 10^{-2}$ | $5 \cdot 10^{-2}$ | $5 \cdot 10^{-2}$ | $1 \cdot 10^{-1}$ | $2 \cdot 10^{-1}$ | $1 \cdot 10^{-1}$ |
| hidden channels | 256 | 256 | 256 | 256 | 256 | 256 | 256 | 256 |

via grid search. The test accuracy corresponding to the hyperparameters that yielded maximum validation accuracy is reported in Table 8.

The ablation study on Chameleon demonstrates that all the components of the model (learnable exponent, ODE framework with the Schrödinger equation, and directionality via the SNA) contribute to the performance of FLODE. The fact that performance drops when any of these components are not used suggests that they all play crucial roles in the model's ability to capture the structure and evolution of heterophilic graphs. It is important to note that the performance appears to be more dependent on the adjustable fraction in the FGL than on the use of the ODE framework, illustrating that the fractional Laplacian alone can effectively capture long-range dependencies. However, when the ODE framework is additionally employed, a noticeable decrease in variance is observed.

**From Theory to Practice.** We conduct an ablation study to investigate the role of depth on Chameleon, Citeseer, Cora, and Squirrel datasets. The results, depicted in Figure 8, demonstrate that the neural ODE framework enables GNNs to scale to large depths (256 layers). Moreover, we see that the fractional Laplacian improves over the standard Laplacian in the heterophilic graphs which is supported by our claims in Section 5.2. We highlight that using only the fractional Laplacian without the neural ODE framework oftentimes outperforms the standard Laplacian with the neural ODE framework. This indicates the importance of the long-range connections built by the fractional Laplacian.

We further demonstrate the close alignment of our theoretical and experimental results, which enables us to precisely anticipate when the models will exhibit HFD or LFD behaviors. In this context, we calculate parameters (according to Theorem D.5) and illustrate at each depth the expected and observed behaviors. For Squirrel and Chameleon, which are heterophilic graphs, we observe that both their theoretical and empirical behaviors are HFD. Additionally, the learned exponent is small. In contrast, for Cora and Citeseer, we see the opposite.

Finally, we employ the best hyperparameters in Table 5a to solve both fractional heat and Schrödinger graph ODEs, further substantiating the intimate link between our theoretical advancements and practical applications.

Table 8: Ablation study on node classification task: top two models are indicated as 1$^{\text{st}}$ and 2$^{\text{nd}}$

(a) Chameleon (directed, heterophilic).

| | Update Rule | Test Accuracy | Dirichlet Energy |
|---|---|---|---|
| D | $\mathbf{x}_{t+1} = \mathbf{x}_t - ih\mathbf{L}^\alpha\mathbf{x}_t\mathbf{W}$ | $77.79 \pm 1.42$ | 0.213 (t=5) |
| | $\mathbf{x}_{t+1} = \mathbf{x}_t - ih\mathbf{L}\ \mathbf{x}_t\mathbf{W}$ | $75.72 \pm 1.13$ | 0.169 (t=6) |
| | $\mathbf{x}_{t+1} = \quad -i\ \mathbf{L}^\alpha\mathbf{x}_t\mathbf{W}$ | $77.35 \pm 2.22$ | 0.177 (t=4) |
| | $\mathbf{x}_{t+1} = \quad -i\ \mathbf{L}\ \mathbf{x}_t\mathbf{W}$ | $69.61 \pm 1.59$ | 0.178 (t=4) |
| U | $\mathbf{x}_{t+1} = \mathbf{x}_t - ih\mathbf{L}^\alpha\mathbf{x}_t\mathbf{W}$ | $73.60 \pm 1.68$ | 0.131 (t=4) |
| | $\mathbf{x}_{t+1} = \mathbf{x}_t - ih\mathbf{L}\ \mathbf{x}_t\mathbf{W}$ | $70.15 \pm 0.86$ | 0.035 (t=4) |
| | $\mathbf{x}_{t+1} = \quad -i\ \mathbf{L}^\alpha\mathbf{x}_t\mathbf{W}$ | $71.25 \pm 3.04$ | 0.118 (t=4) |
| | $\mathbf{x}_{t+1} = \quad -i\ \mathbf{L}\ \mathbf{x}_t\mathbf{W}$ | $67.19 \pm 2.49$ | 0.040 (t=4) |
| D | $\mathbf{x}_{t+1} = \mathbf{x}_t - \ h\mathbf{L}^\alpha\mathbf{x}_t\mathbf{W}$ | $77.33 \pm 1.47$ | 0.378 (t=6) |
| | $\mathbf{x}_{t+1} = \mathbf{x}_t - \ h\mathbf{L}\ \mathbf{x}_t\mathbf{W}$ | $73.55 \pm 0.94$ | 0.165 (t=6) |
| | $\mathbf{x}_{t+1} = \quad - \ \mathbf{L}^\alpha\mathbf{x}_t\mathbf{W}$ | $74.12 \pm 3.60$ | 0.182 (t=4) |
| | $\mathbf{x}_{t+1} = \quad - \ \mathbf{L}\ \mathbf{x}_t\mathbf{W}$ | $68.47 \pm 2.77$ | 0.208 (t=4) |

(b) Squirrel (directed, heterophilic).

| | Update Rule | Test Accuracy | Dirichlet Energy |
|---|---|---|---|
| D | $\mathbf{x}_{t+1} = \mathbf{x}_t - ih\mathbf{L}^\alpha\mathbf{x}_t\mathbf{W}$ | $74.03 \pm 1.58$ | $0.38 \pm 0.02$ |
| | $\mathbf{x}_{t+1} = \mathbf{x}_t - ih\mathbf{L}\ \mathbf{x}_t\mathbf{W}$ | $64.04 \pm 2.25$ | $0.35 \pm 0.02$ |
| | $\mathbf{x}_{t+1} = \quad -i\ \mathbf{L}^\alpha\mathbf{x}_t\mathbf{W}$ | $64.25 \pm 1.85$ | $0.46 \pm 0.01$ |
| | $\mathbf{x}_{t+1} = \quad -i\ \mathbf{L}\ \mathbf{x}_t\mathbf{W}$ | $42.04 \pm 1.58$ | $0.29 \pm 0.05$ |
| U | $\mathbf{x}_{t+1} = \mathbf{x}_t - ih\mathbf{L}^\alpha\mathbf{x}_t\mathbf{W}$ | $64.23 \pm 1.84$ | $0.40 \pm 0.02$ |
| | $\mathbf{x}_{t+1} = \mathbf{x}_t - ih\mathbf{L}\ \mathbf{x}_t\mathbf{W}$ | $55.19 \pm 1.52$ | $0.26 \pm 0.03$ |
| | $\mathbf{x}_{t+1} = \quad -i\ \mathbf{L}^\alpha\mathbf{x}_t\mathbf{W}$ | $61.40 \pm 2.15$ | $0.43 \pm 0.01$ |
| | $\mathbf{x}_{t+1} = \quad -i\ \mathbf{L}\ \mathbf{x}_t\mathbf{W}$ | $41.19 \pm 1.95$ | $0.20 \pm 0.02$ |
| D | $\mathbf{x}_{t+1} = \mathbf{x}_t - \ h\mathbf{L}^\alpha\mathbf{x}_t\mathbf{W}$ | $71.86 \pm 1.65$ | $0.50 \pm 0.01$ |
| | $\mathbf{x}_{t+1} = \mathbf{x}_t - \ h\mathbf{L}\ \mathbf{x}_t\mathbf{W}$ | $59.34 \pm 1.78$ | $0.43 \pm 0.03$ |
| | $\mathbf{x}_{t+1} = \quad - \ \mathbf{L}^\alpha\mathbf{x}_t\mathbf{W}$ | $42.91 \pm 7.86$ | $0.32 \pm 0.08$ |
| | $\mathbf{x}_{t+1} = \quad - \ \mathbf{L}\ \mathbf{x}_t\mathbf{W}$ | $35.37 \pm 1.69$ | $0.25 \pm 0.05$ |
| U | $\mathbf{x}_{t+1} = \mathbf{x}_t - \ h\mathbf{L}^\alpha\mathbf{x}_t\mathbf{W}$ | $62.95 \pm 2.02$ | $0.61 \pm 0.08$ |
| | $\mathbf{x}_{t+1} = \mathbf{x}_t - \ h\mathbf{L}\ \mathbf{x}_t\mathbf{W}$ | $52.19 \pm 1.17$ | $0.51 \pm 0.07$ |
| | $\mathbf{x}_{t+1} = \quad - \ \mathbf{L}^\alpha\mathbf{x}_t\mathbf{W}$ | $59.04 \pm 0.02$ | $0.44 \pm 0.02$ |
| | $\mathbf{x}_{t+1} = \quad - \ \mathbf{L}\ \mathbf{x}_t\mathbf{W}$ | $39.69 \pm 1.54$ | $0.20 \pm 0.02$ |

(c) Citeseer (undirected, homphilic).

| Update Rule | Test Accuracy | Dirichlet Energy |
|---|---|---|
| $\mathbf{x}_{t+1} = \mathbf{x}_t - ih\mathbf{L}^\alpha\mathbf{x}_t\mathbf{W}$ | $78.07 \pm 1.62$ | 0.021 (t=5) |
| $\mathbf{x}_{t+1} = \mathbf{x}_t - ih\mathbf{L}\ \mathbf{x}_t\mathbf{W}$ | $77.97 \pm 2.29$ | 0.019 (t=4) |
| $\mathbf{x}_{t+1} = \quad -i\ \mathbf{L}^\alpha\mathbf{x}_t\mathbf{W}$ | $77.27 \pm 2.10$ | 0.011 (t=6) |
| $\mathbf{x}_{t+1} = \quad -i\ \mathbf{L}\ \mathbf{x}_t\mathbf{W}$ | $77.97 \pm 2.23$ | 0.019 (t=4) |

Table 9: Learned $\alpha$ and spectrum of $\mathbf{W}$. According to Theorem 5.3, we denote FD $\coloneqq \lambda_K(\mathbf{W}) \mathrm{f}_\alpha(\lambda_1(\mathbf{L})) - \lambda_1(\mathbf{W})$ and FD $\coloneqq \Im(\lambda_K(\mathbf{W})) \mathrm{f}_\alpha(\lambda_1(\mathbf{L})) - \Im(\lambda_1(\mathbf{W}))$ for the fractional heat (H) and Schrödinger (S) graph ODEs, respectively. The heterophilic graphs Squirrel and Chameleon exhibit HFD since FD $< 0$, while the homophilic Cora, Citeseer, Pubmed exhibit LFD since FD $> 0$.

| | | Film | Squirrel | Chameleon | Citeseer | Pubmed | Cora |
|---|---|---|---|---|---|---|---|
| | $\lambda_1(\mathbf{L})$ | $-0.9486$ | $-0.8896$ | $-0.9337$ | $-0.5022$ | $-0.6537$ | $-0.4826$ |
| H | $\alpha$ | $1.008 \pm 0.007$ | $0.19 \pm 0.05$ | $0.37 \pm 0.14$ | $0.89 \pm 0.06$ | $1.15 \pm 0.08$ | $0.89 \pm 0.01$ |
| | $\lambda_1(\mathbf{W})$ | $-2.774 \pm 0.004$ | $-1.62 \pm 0.03$ | $-1.81 \pm 0.02$ | $-1.76 \pm 0.01$ | $-1.66 \pm 0.06$ | $-1.81 \pm 0.01$ |
| | $\lambda_K(\mathbf{W})$ | $2.858 \pm 0.009$ | $2.21 \pm 0.03$ | $2.29 \pm 0.05$ | $2.28 \pm 0.06$ | $1.1 \pm 0.3$ | $2.32 \pm 0.01$ |
| | FD | $0.367 \pm 0.001$ | $-0.54 \pm 0.02$ | $-0.42 \pm 0.04$ | $0.52 \pm 0.02$ | $0.97 \pm 0.09$ | $0.60 \pm 0.01$ |
| S | $\alpha$ | $1.000 \pm 0.002$ | $0.17 \pm 0.03$ | $0.34 \pm 0.11$ | $0.90 \pm 0.07$ | $0.76 \pm 0.07$ | $0.90 \pm 0.02$ |
| | $\Im(\lambda_1(\mathbf{W}))$ | $-2.795 \pm 0.001$ | $-1.68 \pm 0.01$ | $-1.79 \pm 0.01$ | $-1.70 \pm 0.04$ | $-1.74 \pm 0.01$ | $-1.78 \pm 0.01$ |
| | $\Im(\lambda_K(\mathbf{W}))$ | $2.880 \pm 0.002$ | $2.21 \pm 0.03$ | $2.46 \pm 0.02$ | $2.29 \pm 0.07$ | $0.98 \pm 0.09$ | $2.30 \pm 0.02$ |
| | FD | $0.4945 \pm 0.0001$ | $-0.48 \pm 0.03$ | $-0.62 \pm 0.03$ | $0.46 \pm 0.06$ | $1.03 \pm 0.05$ | $0.59 \pm 0.01$ |

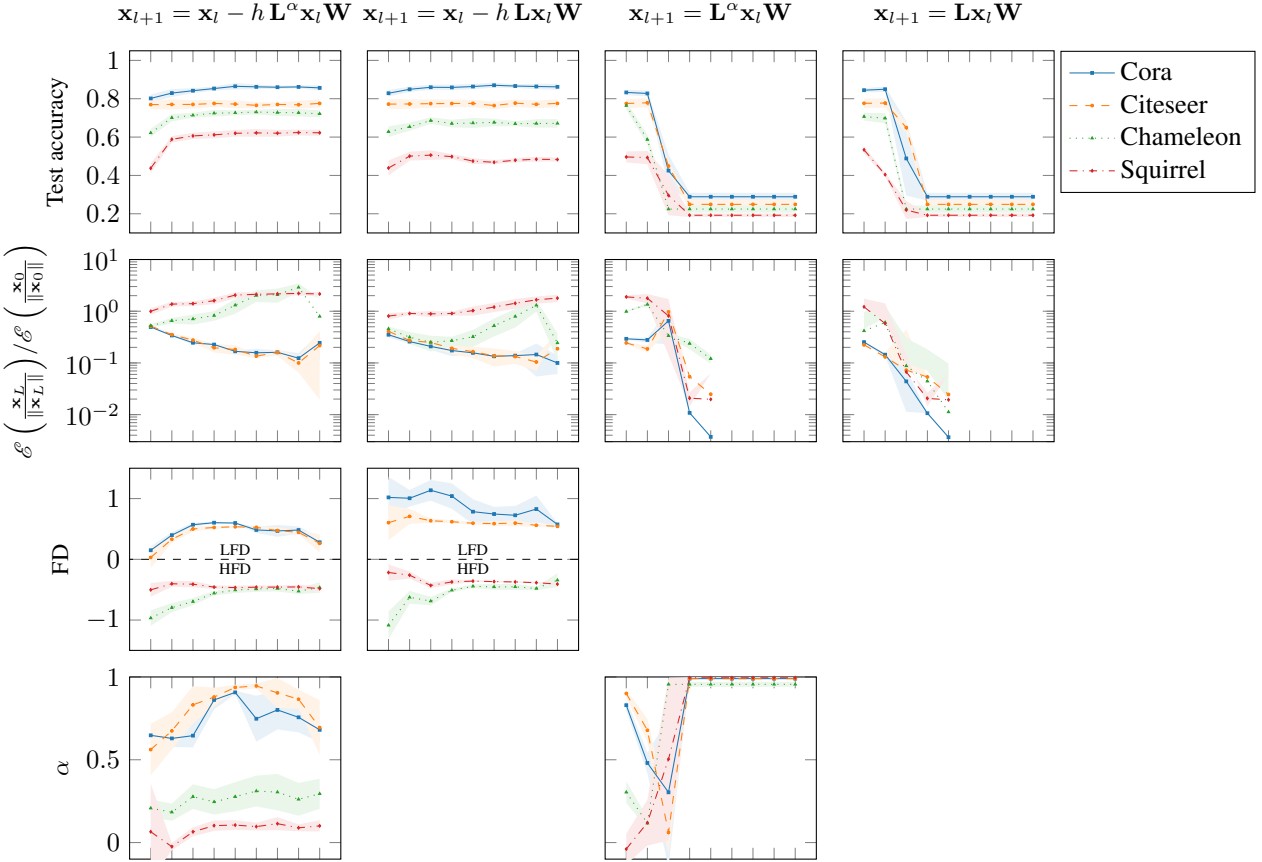

Figure 8: Ablation study on the effect of different update rules and different number of layers on undirected datasets. The x-axis shows the number of layers $2^L$ for $L \in \{0, \ldots, 8\}$. FD is calculated according to Theorem 5.3.

# B   Appendix for Section 3

**Proposition 3.3.** *Let $\mathcal{G}$ be a directed graph with SNA $\mathbf{L}$. For every $\lambda \in \lambda(\mathbf{L})$, it holds $|\lambda| \leq 1$ and $\lambda(\mathbf{I} - \mathbf{L}) = 1 - \lambda(\mathbf{L})$.*

*Proof.* We show that the numerical range $\mathcal{W}(\mathbf{L}) = \left\{ \mathbf{x}^{\mathsf{H}} \mathbf{L} \mathbf{x} : \mathbf{x}^{\mathsf{H}} \mathbf{x} = 1 \right\}$ satisfies $\mathcal{W}(\mathbf{L}) \subset [-1, 1]$. As $\mathcal{W}(\mathbf{L})$ contains all eigenvalues of $\mathbf{L}$ the thesis follows.

Let $\mathbf{A}$ be the adjacency matrix of $\mathcal{G}$ and $\mathbf{x} \in \mathbb{C}^N$ with $\mathbf{x}^{\mathsf{H}} \mathbf{x} = 1$. Applying the Cauchy-Schwartz inequality in $(2)$ and $(3)$, we get

$$
\begin{aligned}
\left| \mathbf{x}^{\mathsf{H}} \mathbf{L} \mathbf{x} \right| &\overset{(1)}{\leq} \sum_{i=1}^{N} \sum_{j=1}^{N} a_{i,j} \frac{|x_i|\,|x_j|}{\sqrt{d_i^{in} d_j^{out}}} \\
&= \sum_{i=1}^{N} \frac{|x_i|}{\sqrt{d_i^{in}}} \sum_{j=1}^{N} a_{i,j} \frac{|x_j|}{\sqrt{d_j^{out}}} \\
&\overset{(2)}{\leq} \sum_{i=1}^{N} \frac{|x_i|}{\sqrt{d_i^{in}}} \sqrt{\sum_{j=1}^{N} a_{i,j} \frac{|x_j|^2}{d_j^{out}} \sum_{j=1}^{N} a_{i,j}} \\
&= \sum_{i=1}^{N} |x_i| \sqrt{\sum_{j=1}^{N} a_{i,j} \frac{|x_j|^2}{d_j^{out}}} \\
&\overset{(3)}{\leq} \sqrt{\sum_{i=1}^{N} |x_i|^2 \sum_{i=1}^{N} \sum_{j=1}^{N} a_{i,j} \frac{|x_j|^2}{d_j^{out}}} \\
&= \sum_{i=1}^{N} |x_i|^2 \;,
\end{aligned}
$$

where we used $a_{i,j}^2 = a_{i,j}$. We have $\sum_{i=1}^{N} |x_i|^2 = \mathbf{x}^{\mathsf{H}} \mathbf{x} = 1$ such that $\mathcal{W}(\mathbf{L}) \subset [-1, 1]$ follows. The second claim follows directly by $(\mathbf{I} - \mathbf{L})\mathbf{v} = \mathbf{v} - \lambda \mathbf{v} = (1 - \lambda)\mathbf{v}$. $\qquad\square$

**Proposition 3.5.** *Let $\mathcal{G}$ be a directed graph with SNA $\mathbf{L}$. Then $1 \in \lambda(\mathbf{L})$ if and only if the graph is weakly balanced. Suppose the graph is strongly connected; then $-1 \in \lambda(\mathbf{L})$ if and only if the graph is weakly balanced with an even period.*

*Proof.* Since the numerical range is only a superset of the set of eigenvalues, we cannot simply consider when the inequalities $(1) - (3)$ in the previous proof are actual equalities. Therefore, we have to find another way to prove the statement. Suppose that the graph is weakly balanced, then

$$
\sum_{j=1}^{N} a_{i,j} \left( \frac{k_j}{\sqrt{d_j^{\text{out}}}} - \frac{k_i}{\sqrt{d_i^{\text{in}}}} \right) = 0 \,, \ \forall j \in \{1, \ldots, N\} \;.
$$

We will prove that $\mathbf{k} = (k_i)_{i=1}^{N}$ is an eigenvector corresponding to the eigenvalue 1,

$$
(\mathbf{Lk})_i = \sum_{j=1}^{N} \frac{a_{i,j}}{\sqrt{d_i^{\text{in}} d_j^{\text{out}}}} k_j = \frac{1}{\sqrt{d_i^{\text{in}}}} \sum_{j=1}^{N} \frac{a_{i,j}}{\sqrt{d_j^{\text{out}}}} k_j = \frac{1}{\sqrt{d_i^{\text{in}}}} \sum_{j=1}^{N} \frac{a_{i,j}}{\sqrt{d_i^{\text{in}}}} k_i = \frac{1}{d_i^{\text{in}}} \left( \sum_{j=1}^{N} a_{i,j} \right) k_i = k_i \,.
$$

For the other direction, suppose that there exists $\mathbf{x} \in \mathbb{R}^N$ such that $\mathbf{x} \neq 0$ and $\mathbf{x} = \mathbf{Lx}$. Then for all $i \in \{1, \ldots, N\}$

$$
0 = (\mathbf{Lx})_i - x_i = \sum_{j=1}^{N} \frac{a_{i,j}}{\sqrt{d_i^{\text{in}} d_j^{\text{out}}}} x_j - x_i = \sum_{j=1}^{N} \frac{a_{i,j}}{\sqrt{d_i^{\text{in}} d_j^{\text{out}}}} x_j - \sum_{j=1}^{N} \frac{a_{i,j}}{d_i^{\text{in}}} x_i
$$

$$= \sum_{j=1}^{N} \frac{a_{i,j}}{\sqrt{d_i^{\text{in}}}} \left( \frac{x_j}{\sqrt{d_j^{\text{out}}}} - \frac{x_i}{\sqrt{d_i^{\text{in}}}} \right),$$

hence, the graph is weakly balanced.

By Perron-Frobenius theorem for irreducible non-negative matrices, one gets that $\mathbf{L}$ has exactly $h$ eigenvalues with maximal modulus corresponding to the $h$ roots of the unity, where $h$ is the period of $\mathbf{L}$. Hence, $-1$ is an eigenvalue of $\mathbf{L}$ if and only if the graph is weakly balanced and $h$ is even. $\qquad\square$

**Proposition 3.6.** *For every* $\mathbf{x} \in \mathbb{C}^{N \times K}$, *we have*

$$\Re \left( \text{trace} \left( \mathbf{x}^{\mathsf{H}} \left( \mathbf{I} - \mathbf{L} \right) \mathbf{x} \right) \right) = \frac{1}{2} \sum_{i,j=1}^{N} a_{i,j} \left\| \frac{\mathbf{x}_i}{\sqrt{d_i^{in}}} - \frac{\mathbf{x}_j}{\sqrt{d_j^{out}}} \right\|_2^2 ,$$

*Moreover, there exists* $\mathbf{x} \neq 0$ *such that* $\mathscr{E}(\mathbf{x}) = 0$ *if and only if the graph is weakly balanced.*

*Proof.* By direct computation, it holds

$$\frac{1}{2} \sum_{i,j=1}^{N} a_{i,j} \left\| \frac{x_{i,:}}{\sqrt{d_i^{\text{in}}}} - \frac{x_{j,:}}{\sqrt{d_j^{\text{out}}}} \right\|_2^2$$

$$= \frac{1}{2} \sum_{i,j=1}^{N} a_{i,j} \sum_{k=1}^{K} \left| \frac{x_{i,k}}{\sqrt{d_i^{\text{in}}}} - \frac{x_{j,k}}{\sqrt{d_j^{\text{out}}}} \right|^2$$

$$= \frac{1}{2} \sum_{i,j=1}^{N} a_{i,j} \sum_{k=1}^{K} \left( \frac{x_{i,k}}{\sqrt{d_i^{\text{in}}}} - \frac{x_{j,k}}{\sqrt{d_j^{\text{out}}}} \right)^* \left( \frac{x_{i,k}}{\sqrt{d_i^{\text{in}}}} - \frac{x_{j,k}}{\sqrt{d_j^{\text{out}}}} \right)$$

$$= \frac{1}{2} \sum_{i,j=1}^{N} \sum_{k=1}^{K} a_{i,j} \frac{|x_{i,k}|^2}{d_i^{\text{in}}} + \frac{1}{2} \sum_{i,j=1}^{N} \sum_{k=1}^{K} a_{i,j} \frac{|x_{j,k}|^2}{d_j^{\text{out}}}$$

$$- \frac{1}{2} \sum_{i,j=1}^{N} \sum_{k=1}^{K} a_{i,j} \frac{x_{i,k}^* x_{j,k}}{\sqrt{d_i^{\text{in}} d_j^{\text{out}}}} - \frac{1}{2} \sum_{i,j=1}^{N} \sum_{k=1}^{K} a_{i,j} \frac{x_{i,k} x_{j,k}^*}{\sqrt{d_i^{\text{in}} d_j^{\text{out}}}}$$

$$= \frac{1}{2} \sum_{i=1}^{N} \sum_{k=1}^{K} |x_{i,k}|^2 + \frac{1}{2} \sum_{j=1}^{N} \sum_{k=1}^{K} |x_{j,k}|^2 - \frac{1}{2} \sum_{i,j=1}^{N} \sum_{k=1}^{K} a_{i,j} \frac{x_{i,k}^* x_{j,k}}{\sqrt{d_i^{\text{in}} d_j^{\text{out}}}} - \frac{1}{2} \sum_{i,j=1}^{N} \sum_{k=1}^{K} a_{i,j} \frac{x_{i,k} x_{j,k}^*}{\sqrt{d_i^{\text{in}} d_j^{\text{out}}}}$$

$$= \sum_{i=1}^{N} \sum_{k=1}^{K} |x_{i,k}|^2 - \frac{1}{2} \sum_{i,j=1}^{N} \sum_{k=1}^{K} a_{i,j} \frac{(\mathbf{x}^{\mathsf{H}})_{k,i} x_{j,k}}{\sqrt{d_i^{\text{in}} d_j^{\text{out}}}} - \frac{1}{2} \sum_{i,j=1}^{N} \sum_{k=1}^{K} a_{i,j} \frac{x_{i,k} (\mathbf{x}^{\mathsf{H}})_{k,j}}{\sqrt{d_i^{\text{in}} d_j^{\text{out}}}}$$

$$= \sum_{i=1}^{N} \sum_{k=1}^{K} |x_{i,k}|^2 - \frac{1}{2} \sum_{i,j=1}^{N} \sum_{k=1}^{K} a_{i,j} \frac{(\mathbf{x}^{\mathsf{H}})_{k,i} x_{j,k}}{\sqrt{d_i^{\text{in}} d_j^{\text{out}}}} - \frac{1}{2} \left( \sum_{i,j=1}^{N} \sum_{k=1}^{K} a_{i,j} \frac{(\mathbf{x}^{\mathsf{H}})_{k,i} x_{j,k}}{\sqrt{d_i^{\text{in}} d_j^{\text{out}}}} \right)^*$$

$$= \Re \left( \sum_{i=1}^{N} \sum_{k=1}^{K} |x_{i,k}|^2 - \sum_{i,j=1}^{N} \sum_{k=1}^{K} a_{i,j} \frac{x_{i,k}^* x_{j,k}}{\sqrt{d_i^{\text{in}} d_j^{\text{out}}}} \right)$$

$$= \Re \left( \text{trace} \left( \mathbf{x}^{\mathsf{H}} \left( \mathbf{I} - \mathbf{L} \right) \mathbf{x} \right) \right).$$

The last claim can be proved as follows. For simplicity, suppose $\mathbf{x} \in \mathbb{R}^N$. The " $\Longleftarrow$ " is clear since one can choose $\mathbf{x}$ to be $\mathbf{k}$. To prove the " $\Longrightarrow$ ", we reason by contradiction. Suppose there exists a $\mathbf{x} \neq 0$ such that $\mathscr{E}(\mathbf{x}) = 0$ and the underlying graph is not weakly connected, i.e.,

$$\forall \tilde{\mathbf{x}} \neq \mathbf{0}, \ \left| \sum_{j=1}^{N} a_{i,j} \left( \frac{\tilde{x}_j}{\sqrt{d_j^{\text{out}}}} - \frac{\tilde{x}_i}{\sqrt{d_i^{\text{in}}}} \right) \right| > 0, \ \forall i \in \{1, \ldots, N\},$$

Then, since $\mathbf{x} \neq 0$,

$$
\begin{aligned}
0 = \mathscr{E}(\mathbf{x}) &= \frac{1}{4} \sum_{i,j=1}^{N} a_{i,j} \left| \frac{x_i}{\sqrt{d_i^{\text{in}}}} - \frac{x_j}{\sqrt{d_j^{\text{out}}}} \right|^2 \\
&\geq \frac{1}{4} \sum_{i=1}^{N} \frac{1}{d_i^{\text{in}}} \left( \sum_{j=1}^{N} a_{i,j} \left| \frac{x_i}{\sqrt{d_i^{\text{in}}}} - \frac{x_j}{\sqrt{d_j^{\text{out}}}} \right|^2 \right) \left( \sum_{j=1}^{N} a_{i,j} \right) \\
&\geq \frac{1}{4} \sum_{i=1}^{N} \frac{1}{d_i^{\text{in}}} \left( \sum_{j=1}^{N} a_{i,j} \left| \frac{x_i}{\sqrt{d_i^{\text{in}}}} - \frac{x_j}{\sqrt{d_j^{\text{out}}}} \right| \right)^2 \\
&\geq \frac{1}{4} \sum_{i=1}^{N} \frac{1}{d_i^{\text{in}}} \left| \sum_{j=1}^{N} a_{i,j} \left( \frac{x_i}{\sqrt{d_i^{\text{in}}}} - \frac{x_j}{\sqrt{d_j^{\text{out}}}} \right) \right|^2 \\
&> 0 \,,
\end{aligned}
$$

where we used Cauchy-Schwartz and triangle inequalities. $\qquad \square$

We give the following simple corollary.

**Corollary B.1.** *For every* $\mathbf{x} \in \mathbb{R}^{N \times K}$, *it holds* $\mathscr{E}(\mathbf{x}) = \frac{1}{2} \Re \left( \operatorname{vec}(\mathbf{x})^{\mathsf{H}} (\mathbf{I} \otimes (\mathbf{I} - \mathbf{L})) \operatorname{vec}(\mathbf{x}) \right)$.

## C   Appendix for Section 4

In this section, we provide some properties about FGLs. The first statement shows that the FGL of a normal SNA $\mathbf{L}$ only changes the magnitude of the eigenvalues of $\mathbf{L}$.

**Lemma C.1.** *Let* $\mathbf{M}$ *be a normal matrix with eigenvalues* $\lambda_1, \ldots, \lambda_N$ *and corresponding eigenvectors* $\mathbf{v}_1, \ldots, \mathbf{v}_N$. *Suppose* $\mathbf{M} = \mathbf{L} \boldsymbol{\Sigma} \mathbf{R}^{\mathsf{H}}$ *is its singular value decomposition. Then it holds*

$$
\boldsymbol{\Sigma} = |\boldsymbol{\Lambda}| \,, \ \mathbf{L} = \mathbf{V}, \ \mathbf{R} = \mathbf{V} \exp(i \boldsymbol{\Theta}) \,, \ \boldsymbol{\Theta} = \operatorname{diag}\left( \{\theta_i\}_{i=1}^{N} \right) \,, \ \theta_i = \operatorname{atan2}\left( \Re \lambda_i, \Im \lambda_i \right) \,.
$$

*Proof.* By hypothesis, there exist a unitary matrix $\mathbf{V}$ such that $\mathbf{M} = \mathbf{V} \boldsymbol{\Lambda} \mathbf{V}^{\mathsf{H}}$, then

$$
\begin{aligned}
\mathbf{M}^{\mathsf{H}} \mathbf{M} &= \mathbf{V} \boldsymbol{\Lambda}^* \mathbf{V}^{\mathsf{H}} \mathbf{V} \boldsymbol{\Lambda} \mathbf{V}^{\mathsf{H}} = \mathbf{V} |\boldsymbol{\Lambda}|^2 \mathbf{V}^{\mathsf{H}} \,, \\
\mathbf{M}^{\mathsf{H}} \mathbf{M} &= \mathbf{R} \boldsymbol{\Sigma} \mathbf{L}^{\mathsf{H}} \mathbf{L} \boldsymbol{\Sigma} \mathbf{R}^{\mathsf{H}} = \mathbf{L} \boldsymbol{\Sigma}^2 \mathbf{L}^{\mathsf{H}} \,.
\end{aligned}
$$

Therefore, $\boldsymbol{\Sigma} = |\boldsymbol{\Lambda}|$ and $\mathbf{L} = \mathbf{V}$

$$
\mathbf{M} = \mathbf{R} |\boldsymbol{\Lambda}| \mathbf{V}^{\mathsf{H}}
$$

Finally, we note that it must hold $\mathbf{R} = \mathbf{V} \exp(i \boldsymbol{\Theta})$ where $\boldsymbol{\Theta} = \operatorname{diag}\left( \{\operatorname{atan2}(\Re \lambda_i, \Im \lambda_i)\}_{i=1}^{N} \right)$ and atan2 is the 2-argument arctangent. $\qquad \square$

We proceed by proving Theorem 4.1, which follows the proof of a similar result given in (Benzi et al., 2020) for the fractional Laplacian defined in the spectral domain of an in-degree normalized graph Laplacian. However, our result also holds for directed graphs and in particular for fractional Laplacians that are defined via the SVD of a graph SNA.

**Lemma C.2.** *Let* $\mathbf{M} \in \mathbb{R}^{n \times n}$ *with singular values* $\sigma(\mathbf{M}) \subset [a, b]$. *For* $f : [a, b] \to \mathbb{R}$, *define* $f(\mathbf{M}) = \mathbf{U} f(\boldsymbol{\Sigma}) \mathbf{V}^{\mathsf{H}}$, *where* $\mathbf{M} = \mathbf{U} \boldsymbol{\Sigma} \mathbf{V}^{\mathsf{H}}$ *is the singular value decomposition of* $\mathbf{M}$. *If* $f$ *has modulus of continuity* $\omega$ *and* $d(i, j) \geq 2$, *it holds*

$$
|f(\mathbf{M})|_{i,j} \leq \left( 1 + \frac{\pi^2}{2} \right) \omega \left( \frac{b-a}{2} |d(i,j) - 1|^{-1} \right) \,.
$$

*Proof.* Let $g : [a, b] \to \mathbb{R}$ be any function, then

$$
\begin{aligned}
\|f(\mathbf{M}) - g(\mathbf{M})\|_2 &= \left\| \mathbf{U} f(\mathbf{\Sigma}) \mathbf{V}^\mathsf{H} - \mathbf{U} g(\mathbf{\Sigma}) \mathbf{V}^\mathsf{H} \right\|_2 \\
&= \|f(\mathbf{\Sigma}) - g(\mathbf{\Sigma})\|_2 \\
&= \|f(\lambda) - g(\lambda)\|_{\infty, \sigma(M)}.
\end{aligned}
$$

The second equation holds since the 2-norm is invariant under unitary transformations. By Jackson's Theorem, there exists for every $m \geq 1$ a polynomial $p_m$ of order $m$ such that

$$
\|f(\mathbf{M}) - p_m(\mathbf{M})\|_2 \leq \|f - p_m\|_{\infty, [a,b]} \leq \left(1 + \frac{\pi^2}{2}\right) \omega \left(\frac{b-a}{2m}\right).
$$

Fix $i, j \in \{1, \ldots, n\}$. If $d(i, j) = m + 1$, then any power of $\mathbf{M}$ up to order $m$ has a zero entry in $(i, j)$, i.e., $(\mathbf{M}^m)_{i,j} = 0$. Hence, $f(\mathbf{M})_{i,j} = f(\mathbf{M})_{i,j} - p_m(\mathbf{M})_{i,j}$, and we get

$$
|f(\mathbf{M})_{i,j}| \leq \|f(\mathbf{M}) - g(\mathbf{M})\|_2 \leq \omega \left(1 + \frac{\pi^2}{2}\right) \left(\frac{b-a}{2m}\right) = \left(1 + \frac{\pi^2}{2}\right) \omega \left(\frac{b-a}{2} |d(i,j) - 1|^{-1}\right)
$$

from which the thesis follows. $\qquad\square$

Finally, we give a proof of Theorem 4.1, which is a consequence of the previous statement.

*Proof of Theorem 4.1.* The eigenvalues of $\mathbf{L}$ are in the unit circle, i.e., $\|\mathbf{L}\| \leq 1$. Hence, $\left\|\mathbf{L}\mathbf{L}^\mathsf{H}\right\| \leq 1$, and the singular values of $\mathbf{L}$ are in $[0, 1]$. By Lemma C.2 and the fact that $f(x) = x^\alpha$ has modulus of continuity $\omega(t) = t^\alpha$ the thesis follows. $\qquad\square$

# D  Appendix for Section 5

In this section, we provide the appendix for Section 5. We begin by analyzing the solution of linear matrix ODEs. For this, let $\mathbf{M} \in \mathbb{C}^{N \times N}$. For $\mathbf{x}_0 \in \mathbb{C}^N$, consider the initial value problem

$$
\mathbf{x}'(t) = -\mathbf{M}\mathbf{x}(t), \quad \mathbf{x}(0) = \mathbf{x}_0. \tag{5}
$$

**Theorem D.1** (Existence and uniqueness of linear ODE solution). *The initial value problem given by (5) has a unique solution $\mathbf{x}(t) \in \mathbb{C}^N$ for any initial condition $\mathbf{x}_0 \in \mathbb{C}^N$.*

The solution of (5) can be expressed using matrix exponentials, even if $\mathbf{M}$ is not symmetric. The matrix exponential is defined as:

$$
\exp(-\mathbf{M}t) = \sum_{k=0}^{\infty} \frac{(-\mathbf{M})^k t^k}{k!},
$$

where $\mathbf{M}^k$ is the $k$-th power of the matrix $\mathbf{M}$. The solution of (5) can then be written as

$$
\mathbf{x}(t) = \exp(-\mathbf{M}t)\mathbf{x}_0. \tag{6}
$$

## D.1  Appendix for Section 5.1

In this section, we analyze the solution to (2) and (3). We further provide a proof for Theorem 5.3. We begin by considering the solution to the fractional heat equation (2). The analysis for the Schrödinger equation (3) follows analogously.

The fractional heat equation $\mathbf{x}'(t) = -\mathbf{L}^\alpha \mathbf{x} \mathbf{W}$ can be vectorized and rewritten via the Kronecker product as

$$
\mathrm{vec}(\mathbf{x})'(t) = -\mathbf{W} \otimes \mathbf{L}^\alpha \mathrm{vec}(\mathbf{x})(t). \tag{7}
$$

In the undirected case $\mathbf{L}$ and $\mathbf{I} - \mathbf{L}$ are both symmetric, and the eigenvalues satisfy the relation $\lambda_i(\mathbf{I} - \mathbf{L}) = 1 - \lambda_i(\mathbf{L})$. The corresponding eigenvectors $\psi_i(\mathbf{L})$ and $\psi_i(\mathbf{I} - \mathbf{L})$ can be chosen to be the same for $\mathbf{L}$ and $\mathbf{I} - \mathbf{L}$. In the following, we assume that these eigenvectors are orthonormalized.

If $\mathbf{L}$ is symmetric, we can decompose it via the spectral theorem into $\mathbf{L} = \mathbf{U}\mathbf{D}\mathbf{U}^T$, where $\mathbf{U} = [\psi_1(\mathbf{L}), \ldots, \psi_N(\mathbf{L})]$ is an orthogonal matrix containing the eigenvectors of $\mathbf{L}$, and $\mathbf{D}$ is the diagonal matrix of eigenvalues.

Due to Lemma C.1, the fractional Laplacian $\mathbf{L}^\alpha$ can be written as $\mathbf{L}^\alpha = \mathbf{U}\,\mathrm{f}_\alpha(\mathbf{D})\mathbf{U}^T$, where $\mathrm{f}_\alpha : \mathbb{R} \to \mathbb{R}$ is the map $x \mapsto \mathrm{sign}(x)\,|x|^\alpha$ and is applied element-wise. Clearly, the eigendecomposition of $\mathbf{L}^\alpha$ is given by the eigenvalues $\{\mathrm{f}_\alpha(\lambda_1(\mathbf{L})), \dots, \mathrm{f}_\alpha(\lambda_N(\mathbf{L}))\}$ and the corresponding eigenvectors $\{\psi_1(\mathbf{L}), \dots, \psi_N(\mathbf{L})\}$.

Now, by well-known properties of the Kronecker product, one can write the eigendecomposition of $\mathbf{W} \otimes \mathbf{L}^\alpha$ as

$$\{\lambda_r(\mathbf{W})\,\mathrm{f}_\alpha\,(\lambda_l(\mathbf{L}))\}_{r\in\{1,\dots,K\},\,l\in\{1,\dots,N\}} \,, \ \{\psi_r(\mathbf{W}) \otimes \psi_l(\mathbf{L})\}_{r\in\{1,\dots,K\},\,l\in\{1,\dots,N\}} \,.$$

Note that $1 \in \lambda(\mathbf{L})$ and, since $\mathrm{trace}(\mathbf{L}) = 0$, the SNA has at least one negative eigenvalue. This property is useful since it allows to retrieve of the indices $(r, l)$ corresponding to eigenvalues with minimal real (or imaginary) parts in a simple way.

The initial condition $\mathrm{vec}(\mathbf{x}_0)$ can be decomposed as

$$\mathrm{vec}(\mathbf{x}_0) = \sum_{r=1}^{K}\sum_{l=1}^{N} c_{r,l}\,\psi_r(\mathbf{W}) \otimes \psi_l(\mathbf{L})\,, \ c_{r,l} = \langle \mathrm{vec}(\mathbf{x}_0)\,,\ \psi_r(\mathbf{W}) \otimes \psi_l(\mathbf{W}) \rangle \,.$$

Then, the solution $\mathrm{vec}(\mathbf{x})(t)$ of (7) can be written as

$$\mathrm{vec}(\mathbf{x})(t) = \sum_{r=1}^{K}\sum_{l=1}^{N} c_{r,l}\,\exp\left(-t\lambda_r(\mathbf{W})\,\mathrm{f}_\alpha\,(\lambda_l(\mathbf{L}))\right)\,\psi_r(\mathbf{W}) \otimes \psi_l(\mathbf{L}). \tag{8}$$

The following result shows the relationship between the frequencies of $\mathbf{I} - \mathbf{L}$ and the Dirichlet energy and serves as a basis for the following proofs.

**Lemma D.2.** *Let $\mathcal{G}$ be a graph with SNA $\mathbf{L}$. Consider $\mathbf{x}(t) \in \mathbb{C}^{N\times K}$ such that there exists $\boldsymbol{\varphi} \in \mathbb{C}^{N\times K} \setminus \{0\}$ with*

$$\frac{\mathrm{vec}(\mathbf{x})(t)}{\|\mathrm{vec}(\mathbf{x})(t)\|_2} \xrightarrow{t\to\infty} \mathrm{vec}(\boldsymbol{\varphi})\,,$$

*and $(\mathbf{I} \otimes (\mathbf{I} - \mathbf{L}))\mathrm{vec}(\boldsymbol{\varphi}) = \lambda\mathrm{vec}(\boldsymbol{\varphi})$. Then,*

$$\mathscr{E}\left(\frac{\mathbf{x}(t)}{\|\mathbf{x}(t)\|_2}\right) \xrightarrow{t\to\infty} \frac{\Re(\lambda)}{2}\,.$$

*Proof.* As $\mathrm{vec}(\boldsymbol{\varphi})$ is the limit of unit vectors, $\mathrm{vec}(\boldsymbol{\varphi})$ is a unit vector itself. We calculate its Dirichlet energy,

$$\mathscr{E}(\mathrm{vec}(\boldsymbol{\varphi})) = \frac{1}{2}\Re\left(\mathrm{vec}(\boldsymbol{\varphi})^{\mathsf{H}}(\mathbf{I} \otimes (\mathbf{I} - \mathbf{L}))\mathrm{vec}(\boldsymbol{\varphi})\right) = \frac{1}{2}\Re\left(\lambda\,\mathrm{vec}(\boldsymbol{\varphi})^{\mathsf{H}}\mathrm{vec}(\boldsymbol{\varphi})\right) = \frac{1}{2}\Re(\lambda)\,.$$

Since $\mathbf{x} \mapsto \mathscr{E}(\mathbf{x})$ is continuous, the thesis follows. $\qquad\square$

Another useful result that will be extensively used in proving Theorem 5.3 is presented next.

**Lemma D.3.** *Suppose $\mathbf{x}(t)$ can be expressed as*

$$\mathbf{x}(t) = \sum_{k=1}^{K}\sum_{n=1}^{N} c_{k,n}\,\exp\left(-t\,\lambda_{k,n}\right)\mathbf{v}_k \otimes \mathbf{w}_n\,,$$

*for some choice of $c_{k,n}$, $\lambda_{k,n}$, $\{\mathbf{v}_k\}$, $\{\mathbf{w}_n\}$. Let $(a, b)$ be the unique index of $\lambda_{k,n}$ with minimal real part and corresponding non-null coefficient $c_{k,n}$, i.e.*

$$(a, b) := \argmin_{(k,n)\in[K]\times[N]} \{\Re(\lambda_{k,n}) : c_{k,n} \neq 0\}\,.$$

*Then*

$$\frac{\mathbf{x}(t)}{\|\mathbf{x}(t)\|_2} \xrightarrow{t\to\infty} \frac{c_{a,b}\,\mathbf{v}_a \otimes \mathbf{w}_b}{\|c_{a,b}\,\mathbf{v}_a \otimes \mathbf{w}_b\|_2}\,.$$

*Proof.* The key insight is to separate the addend with index $(a, b)$. It holds

$$\mathbf{x}(t) = \sum_{k=1}^{K} \sum_{n=1}^{N} c_{k,n} \exp\left(-t\,\lambda_{k,n}\right) \mathbf{v}_n \otimes \mathbf{w}_m$$

$$= \exp\left(-t\,\lambda_{a,b}\right) \left( c_{a,b}\mathbf{v}_a \otimes \mathbf{w}_b + \sum_{\substack{(k,n)\in[K]\times[N] \\ (k,n)\neq(a,b)}} c_{k,n} \exp\left(-t\,\left(\lambda_{k,n} - \lambda_{a,b}\right)\right) \mathbf{v}_k \otimes \mathbf{w}_n \right).$$

We note that

$$\lim_{t\to\infty} \left|\exp\left(-t\,\left(\lambda_{k,n} - \lambda_{a,b}\right)\right)\right| = \lim_{t\to\infty} \left|\exp\left(-t\,\Re\left(\lambda_{k,n} - \lambda_{a,b}\right)\right) \exp\left(-i\,t\,\Im\left(\lambda_{k,n} - \lambda_{a,b}\right)\right)\right|$$

$$= \lim_{t\to\infty} \exp\left(-t\,\Re\left(\lambda_{k,n} - \lambda_{a,b}\right)\right)$$

$$= 0\,,$$

for all $(k, n) \neq (a, b)$, since $\Re\left(\lambda_{k,n} - \lambda_{a,b}\right) > 0$. Therefore, one gets

$$\frac{\mathbf{x}(t)}{\|\mathbf{x}(t)\|_2} \xrightarrow{t\to\infty} \frac{c_{a,b}\,\mathbf{v}_a \otimes \mathbf{w}_b}{\|c_{a,b}\,\mathbf{v}_a \otimes \mathbf{w}_b\|_2}\,,$$

where the normalization removes the dependency on $\exp\left(-t\,\lambda_{a,b}\right)$ □

When $\lambda_{a,b}$ is not unique, it is still possible to derive a convergence result. In this case, $\mathbf{x}$ will converge to an element in the span generated by vectors corresponding to $\lambda_{a,b}$, i.e.,

$$\frac{\mathbf{x}(t)}{\|\mathbf{x}(t)\|_2} \xrightarrow{t\to\infty} \frac{\displaystyle\sum_{(a,b)\in\mathcal{A}} c_{a,b}\,\mathbf{v}_a \otimes \mathbf{w}_b}{\left\|\displaystyle\sum_{(a,b)\in\mathcal{A}} c_{a,b}\,\mathbf{v}_a \otimes \mathbf{w}_b\right\|_2}\,,$$

where $\mathcal{A} := \{(k, n) : \Re(\lambda_{k,n}) = \Re(\lambda_{a,b})\,,\ c_{k,n} \neq 0\}$.

A similar result to Lemma D.3 holds for a slightly different representation of $\mathbf{x}(t)$.

**Lemma D.4.** *Suppose* $\mathbf{x}(t)$ *can be expressed as*

$$\mathbf{x}(t) = \sum_{k=1}^{K} \sum_{n=1}^{N} c_{k,n} \exp\left(i\,t\,\lambda_{k,n}\right) \mathbf{v}_k \otimes \mathbf{w}_n\,,$$

*for some choice of* $c_{k,n}$, $\lambda_{k,n}$, $\{\mathbf{v}_k\}$, $\{\mathbf{w}_n\}$. *Let* $(a, b)$ *be the unique index of* $\lambda_{k,n}$ *with minimal imaginary part and corresponding non-null coefficient* $c_{k,n}$, *i.e.*

$$(a, b) := \underset{(k,n)\in[K]\times[N]}{\arg\min}\ \left\{\Im\left(\lambda_{k,n}\right) : c_{k,n} \neq 0\right\}\,.$$

*Then*

$$\frac{\mathbf{x}(t)}{\|\mathbf{x}(t)\|_2} \xrightarrow{t\to\infty} \frac{c_{a,b}\,\mathbf{v}_a \otimes \mathbf{w}_b}{\|c_{a,b}\,\mathbf{v}_a \otimes \mathbf{w}_b\|_2}\,.$$

*Proof.* The proof follows the same reasoning as in the proof of Lemma D.3. The difference is that the dominating frequency is the one with the minimal imaginary part, since

$$\Re\left(i\,\lambda_{k,n}\right) = -\Im\left(\lambda_{k,n}\right)\,,$$

and, consequently,

$$\underset{(k,n)\in[K]\times[N]}{\arg\max}\ \left\{\Re\left(i\,\lambda_{k,n}\right)\right\} = \underset{(k,n)\in\in[K]\times[N]}{\arg\min}\ \left\{\Im\left(\lambda_{k,n}\right)\right\}\,.$$

□

### D.1.1 Proof of Theorem 5.3

We denote the eigenvalues of $\mathbf{L}$ closest to $0$ from above and below as

$$\lambda_+(\mathbf{L}) := \arg\min_l \{\lambda_l(\mathbf{L}) \ : \ \lambda_l(\mathbf{L}) > 0\}\,,$$
$$\lambda_-(\mathbf{L}) := \arg\max_l \{\lambda_l(\mathbf{L}) \ : \ \lambda_l(\mathbf{L}) < 0\}\,. \tag{9}$$

We assume that the channel mixing $\mathbf{W} \in \mathbb{R}^{K \times K}$ and the graph Laplacians $\mathbf{L}, \mathbf{I} - \mathbf{L} \in \mathbb{R}^{N \times N}$ are real matrices. Finally, we suppose the eigenvalues of a generic matrix $\mathbf{M}$ are sorted in ascending order, i.e., $\lambda_i(\mathbf{M}) \le \lambda_j(\mathbf{M})$, $i < j$.

We now reformulate Theorem 5.3 for the fractional heat equation (2) and provide its full proof, which follows a similar frequency analysis to the one in (Di Giovanni et al., 2023, Theorem B.3)

**Theorem D.5.** *Let $\mathcal{G}$ be an undirected graph with SNA $\mathbf{L}$. Consider the initial value problem in (2) with channel mixing matrix $\mathbf{W} \in \mathbb{R}^{K \times K}$ and $\alpha \in \mathbb{R}$. Then, for almost all initial conditions $\mathbf{x}_0 \in \mathbb{R}^{N \times K}$ the following is satisfied.*

($\alpha > 0$) *The solution to (2) is HFD if*

$$\lambda_K(\mathbf{W})\,\mathrm{f}_\alpha\left(\lambda_1(\mathbf{L})\right) < \lambda_1(\mathbf{W})\,,$$

*and LFD otherwise.*

($\alpha < 0$) *The solution to (2) is $(1 - \lambda_-(\mathbf{L}))$-FD if*

$$\lambda_K(\mathbf{W})\,\mathrm{f}_\alpha\left(\lambda_-(\mathbf{L})\right) < \lambda_1(\mathbf{W})\,\mathrm{f}_\alpha\left(\lambda_+(\mathbf{L})\right)\,,$$

*and $(1 - \lambda_+(\mathbf{L}))$-FD otherwise.*

*Proof of ($\alpha > 0$).* As derived in (8), the solution of (2) with initial condition $\mathbf{x}_0$ can be written in a vectorized form as

$$\mathrm{vec}(\mathbf{x})(t) = \exp\left(-t\,\mathbf{W}^\mathsf{T} \otimes \mathbf{L}^\alpha\right)\mathrm{vec}(\mathbf{x}_0)$$
$$= \sum_{r=1}^{K}\sum_{l=1}^{N} c_{r,l}\,\exp\left(-t\,\lambda_r(\mathbf{W})\,\mathrm{f}_\alpha\left(\lambda_l(\mathbf{L})\right)\right)\,\psi_r(\mathbf{W}) \otimes \psi_l(\mathbf{L}),$$

where $\lambda_r(\mathbf{W})$ are the eigenvalues of $\mathbf{W}$ with corresponding eigenvectors $\psi_r(\mathbf{W})$, and $\lambda_l(\mathbf{L})$ are the eigenvalues of $\mathbf{L}$ with corresponding eigenvectors $\psi_l(\mathbf{L})$. The coefficients $c_{r,l}$ are the Fourier coefficients of $\mathbf{x}_0$, i.e.,

$$c_{r,l} := \langle \mathrm{vec}(\mathbf{x}_0),\ \psi_r(\mathbf{W}) \otimes \psi_l(\mathbf{L})\rangle\,.$$

The key insight is to separate the eigenprojection corresponding to the most negative frequency. By Lemma D.3, this frequency component dominates for $t$ going to infinity.

Suppose

$$\lambda_K(\mathbf{W})\,\mathrm{f}_\alpha\left(\lambda_1(\mathbf{L})\right) < \lambda_1(\mathbf{W})\,\mathrm{f}_\alpha\left(\lambda_N(\mathbf{L})\right) = \lambda_1(\mathbf{W})\,.$$

In this case, $\lambda_K(\mathbf{W})\,\mathrm{f}_\alpha\left(\lambda_1(\mathbf{L})\right)$ is the most negative frequency. Assume for simplicity that $\lambda_K(\mathbf{W})$ has multiplicity one; the argument can be applied even if this is not the case, since the corresponding eigenvectors are orthogonal for higher multiplicities.

For almost all initial conditions $\mathbf{x}_0$, the coefficient $c_{K,1}$ is not null; hence

$$\frac{\mathrm{vec}(\mathbf{x})(t)}{\|\mathrm{vec}(\mathbf{x})(t)\|_2} \xrightarrow{t \to \infty} \frac{c_{K,1}\,\psi_K(\mathbf{W}) \otimes \psi_1(\mathbf{L})}{\|c_{K,1}\,\psi_K(\mathbf{W}) \otimes \psi_1(\mathbf{L})\|_2}\,.$$

By standard properties of the Kronecker product, we have

$$(\mathbf{I} \otimes \mathbf{L})\left(\psi_K(\mathbf{W}) \otimes \psi_1(\mathbf{L})\right) = (\mathbf{I}\,\psi_K(\mathbf{W})) \otimes (\mathbf{L}\,\psi_1(\mathbf{L})) = \lambda_1(\mathbf{L})\,\psi_K(\mathbf{W}) \otimes \psi_1(\mathbf{L})\,, \tag{10}$$

i.e., $\psi_K(\mathbf{W}) \otimes \psi_1(\mathbf{L})$ is an eigenvector of $\mathbf{I} \otimes \mathbf{L}$ corresponding to the eigenvalue $\lambda_1(\mathbf{L})$. Then, by Proposition 3.3, $\psi_K(\mathbf{W}) \otimes \psi_1(\mathbf{L})$ is also an eigenvector of $\mathbf{I} \otimes \mathbf{I} - \mathbf{L}$ corresponding to the eigenvalue $1 - \lambda_1(\mathbf{L}) = \lambda_N(\mathbf{I} - \mathbf{L})$. An application of Lemma D.2 finishes the proof.

Similarly, we can show that if $\alpha > 0$ and $\lambda_K(\mathbf{W})\,\mathrm{f}_\alpha\left(\lambda_1(\mathbf{L})\right) > \lambda_1(\mathbf{W})$ the lowest frequency component $\lambda_1(\mathbf{I} - \mathbf{L})$ is dominant.

*Proof of* $(\alpha < 0)$. In this case either $f_\alpha\left(\lambda_+\left(\mathbf{L}\right)\right)\lambda_1\left(\mathbf{W}\right)$ or $f_\alpha\left(\lambda_-\left(\mathbf{L}\right)\right)\lambda_K\left(\mathbf{W}\right)$ are the most negative frequency components. Hence, if $f_\alpha\left(\lambda_-\left(\mathbf{L}\right)\right)\lambda_K\left(\mathbf{W}\right) > f_\alpha\left(\lambda_+\left(\mathbf{L}\right)\right)\lambda_1\left(\mathbf{W}\right)$ the frequency $f_\alpha\left(\lambda_+\left(\mathbf{L}\right)\right)\lambda_1\left(\mathbf{W}\right)$ is dominating and otherwise the frequency $f_\alpha\left(\lambda_-\left(\mathbf{L}\right)\right)\lambda_K\left(\mathbf{W}\right)$. We can see this by following the exact same reasoning of *(i)*. □

**Remark D.6.** *In the proof of* $(\alpha < 0)$*, we are tacitly assuming that* $\mathbf{L}$ *has only non-zero eigenvalues. If not, we can truncate the* SVD *and remove all zeros singular values (which correspond to zeros eigenvalues). In doing so, we obtain the best invertible approximation of* $\mathbf{L}$ *to which the theorem can be applied.*

We now generalize the previous result to all directed graphs with normal SNA.

**Theorem D.7.** *Let* $\mathcal{G}$ *be a a strongly connected directed graph with normal* SNA $\mathbf{L}$ *such that* $\lambda_1(\mathbf{L}) \in \mathbb{R}$*. Consider the initial value problem in* (2) *with channel mixing matrix* $\mathbf{W} \in \mathbb{R}^{K\times K}$ *and* $\alpha > 0$*. Then, for almost all initial values* $\mathbf{x}_0 \in \mathbb{R}^{N\times K}$ *the solution to* (2) *is* HFD *if*

$$\lambda_K(\mathbf{W})|\lambda_1(\mathbf{L})|^\alpha < \lambda_1(\mathbf{W})|\lambda_N(\mathbf{L})|^\alpha,$$

*and* LFD *otherwise.*

*Proof.* Any normal matrix is unitary diagonalizable, i.e., there exist eigenvalues $\lambda_1, \ldots, \lambda_N$ and corresponding eigenvectors $\mathbf{v}_1, \ldots, \mathbf{v}_N$ such that $\mathbf{L} = \mathbf{V}\boldsymbol{\Lambda}\mathbf{V}^H$. Then, by Lemma C.1, the singular value decomposition of $\mathbf{L}$ is given by $\mathbf{L} = \mathbf{U}\boldsymbol{\Sigma}\mathbf{V}^H$, where

$$\boldsymbol{\Sigma} = |\boldsymbol{\Lambda}| \; , \;\; \mathbf{U} = \mathbf{V}\exp\left(i\boldsymbol{\Theta}\right) \; , \;\; \boldsymbol{\Theta} = \mathrm{diag}\left(\{\theta_i\}_{i=1}^N\right) \; , \;\; \theta_i = \mathrm{atan2}\left(\Re\lambda_i, \Im\lambda_i\right) .$$

Hence,

$$\mathbf{L}^\alpha = \mathbf{U}\boldsymbol{\Sigma}^\alpha\mathbf{V}^H = \mathbf{V}\left|\boldsymbol{\Lambda}\right|^\alpha\exp\left(i\boldsymbol{\Theta}\right)\mathbf{V}^H.$$

Then, equivalent to the derivation of (8), the solution to the vectorized fractional heat equation

$$\mathrm{vec}(\mathbf{x})'(t) = -\mathbf{W}\otimes\mathbf{L}^\alpha\mathrm{vec}(\mathbf{x})(t)$$

is given by

$$\mathrm{vec}(\mathbf{x})(t) = \sum_{r=1}^K\sum_{l=1}^N c_{r,l}\,\exp\left(-t\lambda_r\left(\mathbf{W}\right)f_\alpha\left(\lambda_l\left(\mathbf{L}\right)\right)\right)\,\psi_r(\mathbf{W})\otimes\psi_l(\mathbf{L}).$$

with

$$f_\alpha(\lambda_l(\mathbf{L})) = |\lambda(\mathbf{L})_l|^\alpha\exp(i\theta_l).$$

Now, equivalent to the proof of Theorem 5.3, we apply Lemma D.3. Therefore, the dominating frequency is given by the eigenvalue of $\mathbf{W}\otimes\mathbf{L}^\alpha$ with the most negative real part. The eigenvalues of $\mathbf{W}\otimes\mathbf{L}^\alpha$ are given by $\lambda_r(\mathbf{W})f_\alpha(\lambda_l(\mathbf{L}))$ for $r = 1, \ldots, K$, $l = 1, \ldots, N$. The corresponding real parts are given by

$$\Re(\lambda_r(\mathbf{W})f_\alpha(\lambda_l(\mathbf{L}))) = \lambda_r(\mathbf{W})\left|\lambda(\mathbf{L})_i\right|^\alpha\cos(\theta_i) = \lambda_r(\mathbf{W})\left|\lambda(\mathbf{L})_i\right|^{\alpha-1}\Re(\lambda(\mathbf{L})_i).$$

By Perron-Frobenius, the eigenvalue of $\mathbf{L}$ with the largest eigenvalues is given by $\lambda_N(\mathbf{L}) \in \mathbb{R}$. Hence, for all $l = 1, \ldots, N$,

$$|\lambda(\mathbf{L})_l|^\alpha\cos(\theta_l) \le |\lambda(\mathbf{L})_N|^\alpha .$$

Similarly, for all $l = 1, \ldots, N$ with $\Re(\lambda(\mathbf{L})_l) < 0$,

$$-|\lambda(\mathbf{L})_l|^\alpha\cos(\theta_l) \le -|\lambda(\mathbf{L})_1|^\alpha .$$

Thus, the frequency with the most negative real part is either given by $\lambda_K(\mathbf{W})f_\alpha\left(\lambda_1(\mathbf{L})\right)$ or $\lambda_1(\mathbf{W})f_\alpha\left(\lambda_N(\mathbf{L})\right)$. The remainder of the proof is analogous to the proof of Theorem D.7.

□

In the following, we provide the complete statement and proof for the claims made in Theorem 5.3 when the underlying ODE is the Schrödinger equation as presented in (3).

**Theorem D.8.** *Let $\mathcal{G}$ be a undirected graph with SNA $\mathbf{L}$. Consider the initial value problem in (3) with channel mixing matrix $\mathbf{W} \in \mathbb{C}^{K \times K}$ and $\alpha \in \mathbb{R}$. Suppose that $\mathbf{W}$ has at least one eigenvalue with non-zero imaginary part and sort the eigenvalues of $\mathbf{W}$ in ascending order with respect to their imaginary part. Then, for almost initial values $\mathbf{x}_0 \in \mathbb{C}^{N \times K}$, the following is satisfied.*

$(\alpha > 0)$ *Solutions of (3) are HFD if*

$$\Im\left(\lambda_K(\mathbf{W})\right) f_\alpha\left(\lambda_1(\mathbf{L})\right) < \Im\left(\lambda_1(\mathbf{W})\right),$$

*and LFD otherwise.*

$(\alpha < 0)$ *Let $\lambda_+(\mathbf{L})$ and $\lambda_-(\mathbf{L})$ be the smallest positive and biggest negative non-zero eigenvalue of $\mathbf{L}$, respectively. Solutions of (3) are $(1 - \lambda_-(\mathbf{L}))$-FD if*

$$\Im\left(\lambda_K(\mathbf{W})\right) f_\alpha\left(\lambda_-(\mathbf{L})\right) < \Im\left(\lambda_1(\mathbf{W})\right) f_\alpha\left(\lambda_+(\mathbf{L})\right).$$

*Otherwise, solutions of (3) are $(1 - \lambda_+(\mathbf{L}))$-FD.*

*Proof.* The proof follows the same reasoning as the proof for the heat equation in Theorem D.5. The difference is that we now apply Lemma D.4 instead of Lemma D.3.

Therefore, the dominating frequency is either $\lambda_K(\mathbf{W}) f_\alpha\left(\lambda_1(\mathbf{L})\right)$ or $\lambda_1(\mathbf{W}) f_\alpha\left(\lambda_N(\mathbf{L})\right)$ if $\alpha > 0$, and $\lambda_K(\mathbf{W}) f_\alpha\left(\lambda_-(\mathbf{L})\right)$ or $\lambda_1(\mathbf{W}) f_\alpha\left(\lambda_+(\mathbf{L})\right)$ if $\alpha < 0$. $\qquad\square$

## D.2 Frequency Dominance for Numerical Approximations of the Heat Equation

For $n \in \mathbb{N}$ and $h \in \mathbb{R}$, $h > 0$, the solution of (2) at time $nh > 0$ can be approximated with an explicit Euler scheme

$$\mathrm{vec}(\mathbf{x})(n\,h) = \sum_{k=0}^{n} \binom{n}{k} h^k (-\mathbf{W} \otimes \mathbf{L}^\alpha)^k \mathrm{vec}(\mathbf{x}_0),$$

which can be further simplified via the binomial theorem as

$$\mathrm{vec}(\mathbf{x})(n\,h) = \left(\mathbf{I} - h\left(\mathbf{W} \otimes \mathbf{L}^\alpha\right)\right)^n \mathrm{vec}(\mathbf{x}_0). \tag{11}$$

Hence, it holds the representation formula

$$\mathrm{vec}(\mathbf{x})(n\,h) = \sum_{r,l} c_{r,l} \left(1 - h\,\lambda_r\left(\mathbf{W}\right) f_\alpha\left(\lambda_l(\mathbf{L})\right)\right)^n \psi_r\left(\mathbf{W}\right) \otimes \psi_l\left(\mathbf{L}\right).$$

In this case, the dominating frequency maximizes $|1 - h\,\lambda_r\left(\mathbf{W}\right) f_\alpha\left(\lambda_l(\mathbf{L})\right)|$. When $h < \|\mathbf{W}\|^{-1}$, the product $h\,\lambda_r\left(\mathbf{W}\right) f_\alpha\left(\lambda_l(\mathbf{L})\right)$ is guaranteed to be in $[-1, 1]$, and

$$|1 - h\,\lambda_r\left(\mathbf{W}\right) f_\alpha\left(\lambda_l(\mathbf{L})\right)| = 1 - h\,\lambda_r\left(\mathbf{W}\right) f_\alpha\left(\lambda_l(\mathbf{L})\right) \in [0, 2].$$

Therefore, the dominating frequency minimizes $h\,\lambda_r\left(\mathbf{W}\right) f_\alpha\left(\lambda_l(\mathbf{L})\right)$. This is the reasoning behind the next result.

**Proposition D.9.** *Let $h \in \mathbb{R}$, $h > 0$. Consider the fractional heat equation (2) with $\alpha \in \mathbb{R}$. Let $\{\mathbf{x}(n\,h)\}_{n \in \mathbb{N}}$ be the trajectory of vectors derived by approximating (2) with an explicit Euler scheme with step size $h$. Suppose $h < \|\mathbf{W}\|^{-1}$. Then, for almost all initial values $\mathbf{x}_0$*

$$\mathscr{E}\left(\frac{\mathbf{x}(n\,h)}{\|\mathbf{x}(n\,h)\|_2}\right) \xrightarrow{n \to \infty} \begin{cases} \dfrac{\lambda_N\left(\mathbf{I} - \mathbf{L}\right)}{2}, & \text{if } \lambda_K(\mathbf{W}) f_\alpha\left(\lambda_1(\mathbf{L})\right) < \lambda_1(\mathbf{W}), \\ 0, & \text{otherwise}. \end{cases}$$

*Proof.* Define

$$(\lambda_a, \lambda_b) := \arg\max_{r,l} \left\{|1 - h\lambda_r\left(\mathbf{W}\right) f_\alpha\left(\lambda_l(\mathbf{L})\right)| : r \in \{1, \dots, K\}, l \in \{1, \dots, N\}\right\}.$$

By the hypothesis on $h$, this is equivalent to

$$(\lambda_a, \lambda_b) = \arg\min_{r,l} \left\{\lambda_r\left(\mathbf{W}\right) f_\alpha\left(\lambda_l(\mathbf{L})\right) : r \in \{1, \dots, K\}, l \in \{1, \dots, N\}\right\}.$$

Therefore, $(\lambda_a, \lambda_b)$ is either $(\lambda_1(\mathbf{W}), \lambda_N(\mathbf{L}))$ or $(\lambda_K(\mathbf{W}), \lambda_1(\mathbf{L}))$. Hence,

$$\frac{\mathrm{vec}(\mathbf{x})(n\,h)}{\|\mathrm{vec}(\mathbf{x})(n\,h)\|_2} \xrightarrow{n\to\infty} \frac{c_{a,b}\,\psi_a\,(\mathbf{W}) \otimes \psi_b\,(\mathbf{L})}{\|c_{a,b}\,\psi_a\,(\mathbf{W}) \otimes \psi_b\,(\mathbf{L})\|_2}\,.$$

If the condition $\lambda_K(\mathbf{W})\,\mathrm{f}_\alpha\,(\lambda_1(\mathbf{L})) < \lambda_1(\mathbf{W})$ is satisfied, we have $b = 1$. Then by (10), the normalized $\mathrm{vec}(\mathbf{x})$ converges to the eigenvector of $\mathbf{I} \otimes \mathbf{I} - \mathbf{L}$ corresponding to the largest frequency $1 - \lambda_1(\mathbf{L}) = \lambda_N(\mathbf{I} - \mathbf{L})$. An application of Lemma D.2 finishes the proof.

If $\lambda_K(\mathbf{W})\,\mathrm{f}_\alpha\,(\lambda_1(\mathbf{L})) < \lambda_1(\mathbf{W})$ is not satisfied, we have $b = N$, and the other direction follows with the same argument. □

Similarly to Proposition D.9 one can prove the following results for negative fractions.

**Proposition D.10.** *Let* $h \in \mathbb{R}$, $h > 0$. *Consider the fractional heat equation* (2) *with* $\alpha < 0$. *Let* $\{\mathbf{x}(n\,h)\}_{n\in\mathbb{N}}$ *be the trajectory of vectors derived by approximating the solution of* (2) *with an explicit Euler scheme with step size* $h$. *Suppose that* $h < \|\mathbf{W}\|^{-1}$. *The approximated solution is* $(1 - \lambda_-(\mathbf{L}))$-FD *if*

$$\lambda_1(\mathbf{W})\,\mathrm{f}_\alpha\,(\lambda_+(\mathbf{L})) < \lambda_K(\mathbf{W})\,\mathrm{f}_\alpha\,(\lambda_-(\mathbf{L}))\,,$$

*and* $(1 - \lambda_+(\mathbf{L}))$-FD *otherwise.*

*Proof.* The proof follows the same reasoning as the proof of Proposition D.9 by realizing that the dominating frequencies $(\lambda_a, \lambda_b)$ are either given by $(\lambda_1(\mathbf{W}), \lambda_+(\mathbf{L}))$ or $(\lambda_K(\mathbf{W}), \lambda_-(\mathbf{L}))$. □

### D.3 Frequency Dominance for Numerical Approximations of the Schrödinger Equation

For $n \in \mathbb{N}$ and $h \in \mathbb{R}$, $h > 0$, the solution of (3) at time $n\,h > 0$ can be approximated with an explicit Euler scheme as well. Similarly to the previous section, we can write

$$\mathrm{vec}(\mathbf{x})(n\,h) = (\mathbf{I} + i\,h\,(\mathbf{W} \otimes \mathbf{L}^\alpha))^n\,\mathrm{vec}(\mathbf{x}_0)\,.$$

and

$$\mathrm{vec}(\mathbf{x})(n\,h) = \sum_{r,l} c_{r,l}\,(1 + i\,h\,\lambda_r\,(\mathbf{W})\,\mathrm{f}_\alpha\,(\lambda_l(\mathbf{L})))^n\,\psi_r\,(\mathbf{W}) \otimes \psi_l\,(\mathbf{L})\,.$$

The dominating frequency will be discussed in the following theorem.

**Proposition D.11.** *Let* $h \in \mathbb{R}$, $h > 0$. *Let* $\{\mathbf{x}(n\,h)\}_{n\in\mathbb{N}}$ *be the trajectory of vectors derived by approximating* (3) *with an explicit Euler scheme with sufficiently small step size* $h$. *Sort the eigenvalues of* $\mathbf{W}$ *in ascending order with respect to their imaginary part. Then, for almost all initial values* $\mathbf{x}_0$

$$\mathscr{E}\left(\frac{\mathbf{x}(n\,h)}{\|\mathbf{x}(n\,h)\|_2}\right) \xrightarrow{n\to\infty} \begin{cases} \dfrac{\lambda_N(\mathbf{I} - \mathbf{L})}{2}\,, & \text{if } \mathrm{f}_\alpha\,(\lambda_1(\mathbf{L}))\,\Im\,(\lambda_K(\mathbf{W})) < \mathrm{f}_\alpha\,(\lambda_N(\mathbf{L}))\,\Im\,(\lambda_1(\mathbf{W})) \\ 0\,, & \text{otherwise.} \end{cases}$$

*Proof.* Define

$$(\lambda_a, \lambda_b) := \arg\max_{r,l} \{|1 + i\,h\lambda_r\,(\mathbf{W})\,\mathrm{f}_\alpha\,(\lambda_l(\mathbf{L}))| \; : \; r \in \{1, \ldots, K\}\,, \; l \in \{1, \ldots, N\}\}\,.$$

By definition of $a$ and $b$, for all $r$ and $l$ it holds

$$|1 + i\,h\,\lambda_a\,(\mathbf{W})\,\mathrm{f}_\alpha\,(\lambda_b(\mathbf{L}))| > |1 + i\,h\,\lambda_r\,(\mathbf{W})\,\mathrm{f}_\alpha\,(\lambda_l(\mathbf{L}))|\,. \tag{12}$$

Hence,

$$\frac{\mathrm{vec}(\mathbf{x})(t)}{\|\mathrm{vec}(\mathbf{x})(t)\|_2} \xrightarrow{t\to\infty} \frac{c_{a,b}\psi_a\,(\mathbf{W}) \otimes \psi_b\,(\mathbf{L})}{\|c_{a,b}\psi_a\,(\mathbf{W}) \otimes \psi_b\,(\mathbf{L})\|_2}\,.$$

We continue by determining the indices $a$ and $b$. To do so, we note that (12) is equivalent to

$$\mathrm{f}_\alpha\,(\lambda_l\,(\mathbf{L}))\,\Im\,(\lambda_r\,(\mathbf{W})) - \mathrm{f}_\alpha\,(\lambda_b\,(\mathbf{L}))\,\Im\,(\lambda_a\,(\mathbf{W}))$$

$$> \frac{h}{2}\left(\mathrm{f}_\alpha\,(\lambda_l\,(\mathbf{L}))^2\,|\lambda_r\,(\mathbf{W})|^2 - \mathrm{f}_\alpha\,(\lambda_b(\mathbf{L}))^2\,|\lambda_a\,(\mathbf{W})|^2\right)$$

for all $r$, $l$. Denote by $\varepsilon$ the gap

$$0 < \varepsilon := \min_{(r,l) \neq (a,b)} \left\{ f_\alpha \left( \lambda_l \left( \mathbf{L} \right) \right) \Im \left( \lambda_r \left( \mathbf{W} \right) \right) - f_\alpha \left( \lambda_b \left( \mathbf{L} \right) \right) \Im \left( \lambda_a \left( \mathbf{W} \right) \right) \right\} .$$

Noting that

$$\begin{cases} \dfrac{h}{2} \left( f_\alpha \left( \lambda_l \left( \mathbf{L} \right) \right)^2 \left| \lambda_r \left( \mathbf{W} \right) \right|^2 - f_\alpha \left( \lambda_b (\mathbf{L}) \right)^2 \left| \lambda_a \left( \mathbf{W} \right) \right|^2 \right) \leq h \left\| \mathbf{W} \right\|^2 \left\| \mathbf{L} \right\|^{2\alpha} = h \left\| \mathbf{W} \right\|^2 , \\[2mm] \dfrac{h}{2} \left( f_\alpha \left( \lambda_l \left( \mathbf{L} \right) \right)^2 \left| \lambda_r \left( \mathbf{W} \right) \right|^2 - f_\alpha \left( \lambda_b (\mathbf{L}) \right)^2 \left| \lambda_a \left( \mathbf{W} \right) \right|^2 \right) < \varepsilon \end{cases}$$

one gets that (12) is satisfied for $h < \varepsilon \left\| \mathbf{W} \right\|^{-2}$. Therefore, for sufficiently small $h$, the dominating frequencies are the ones with minimal imaginary part, i.e., either $f_\alpha \left( \lambda_1 (\mathbf{L}) \right) \Im \left( \lambda_K (\mathbf{W}) \right)$ or $f_\alpha \left( \lambda_N (\mathbf{L}) \right) \Im \left( \lambda_1 (\mathbf{W}) \right)$. If $f_\alpha \left( \lambda_1 (\mathbf{L}) \right) \Im \left( \lambda_K (\mathbf{W}) \right) < f_\alpha \left( \lambda_N (\mathbf{L}) \right) \Im \left( \lambda_1 (\mathbf{W}) \right)$, then $b = 1$, and the normalized $\mathrm{vec} \left( \mathbf{x} \right)$ converges to the eigenvector corresponding to the smallest frequency $\lambda_1 (\mathbf{L})$. By (10), this is also the eigenvector of $\mathbf{I} \otimes \mathbf{I} - \mathbf{L}$ corresponding to the largest frequency $1 - \lambda_1 (\mathbf{L}) = \lambda_N (\mathbf{I} - \mathbf{L})$. An application of Lemma D.2 finishes the proof. $\qquad \square$

Finally, we present a similar result for negative powers.

**Proposition D.12.** *Let $h \in \mathbb{R}$, $h > 0$. Consider the fractional Schrödinger equation (3) with $\alpha < 0$. Let $\{ \mathbf{x}(n\, h) \}_{n \in \mathbb{N}}$ be the trajectory of vectors derived by approximating the solution of (3) with an explicit Euler scheme with step size $h$. Suppose that $h$ is sufficiently small. Sort the eigenvalues of $\mathbf{W}$ in ascending order with respect to their imaginary part. The approximated solution is $(1 - \lambda_+ (\mathbf{L}))$-FD if*

$$\lambda_1 (\mathbf{W}) \, f_\alpha \left( \lambda_+ (\mathbf{L}) \right) < \lambda_K (\mathbf{W}) \, f_\alpha \left( \lambda_- (\mathbf{L}) \right) ,$$

*and $(1 - \lambda_- (\mathbf{L}))$-FD otherwise.*

*Proof.* Similar to Proposition D.11, we can prove the statement by realizing that the dominating frequencies $(\lambda_a, \lambda_b)$ in (12) are either given by $(\lambda_1 (\mathbf{W}), \lambda_+ (\mathbf{L}))$ or $(\lambda_K (\mathbf{W}), \lambda_- (\mathbf{L}))$. $\qquad \square$

# E   Appendix for Section 5.2

We begin this section by describing the solution of general linear matrix ODEs of the form (6) in terms of the Jordan decomposition of $\mathbf{M}$. This is required when $\mathbf{M}$ is not diagonalizable. For instance, the SNA of a directed graph is not in general a symmetric matrix, hence, not guaranteed to be diagonalizable. We then proceed in Appendix E.1 with the proof of Theorem 5.6.

For a given matrix $\mathbf{M} \in \mathbb{C}^{N \times N}$, the Jordan normal form is given by

$$\mathbf{M} = \mathbf{P} \mathbf{J} \mathbf{P}^{-1},$$

where $\mathbf{P} \in \mathbb{C}^{N \times N}$ is an invertible matrix whose columns are the generalized eigenvectors of $\mathbf{M}$, and $\mathbf{J} \in \mathbb{C}^{N \times N}$ is a block-diagonal matrix with Jordan blocks along its diagonal. Denote with $\lambda_1, \ldots, \lambda_m$ the eigenvalues of $\mathbf{M}$ and with $\mathbf{J}_1, \ldots, \mathbf{J}_m$ the corresponding Jordan blocks. Let $k_l$ be the algebraic multiplicity of the eigenvalue $\lambda_l$, and denote with $\left\{ \psi_l^i (\mathbf{M}) \right\}_{i \in \{1, \ldots, k_l\}}$ the generalized eigenvectors of the Jordan block $\mathbf{J}_l$.

We begin by giving the following well-known result, which fully characterizes the frequencies for the solution of a linear matrix ODE.

**Lemma E.1.** *Let $\mathbf{M} = \mathbf{P} \mathbf{J} \mathbf{P}^{-1} \in \mathbb{C}^{N \times N}$ be the Jordan normal form of $\mathbf{M}$. Let $\mathbf{x} : [0, T] \to \mathbb{R}^n$ be a solution to*

$$\mathbf{x}'(t) = \mathbf{M} \mathbf{x}(t) , \ \mathbf{x}(0) = \mathbf{x}_0.$$

*Then, $\mathbf{x}$ is given by*

$$\mathbf{x}(t) = \sum_{l=1}^m \exp \left( \lambda_l (\mathbf{M}) t \right) \sum_{i=1}^{k_l} c_l^j \sum_{j=1}^i \frac{t^{i-j}}{(i-j)!} \psi_l^j (\mathbf{M}),$$

*where*

$$\mathbf{x}_0 = \sum_{l=1}^m \sum_{i=1}^{k_l} c_l^i \mathbf{P} \mathbf{e}_l^i ,$$

*and $\left\{ \mathbf{e}_l^i : i \in \{1, \ldots k_l\} \, , \ l \in \{1, \ldots, m\} \right\}$ is the standard basis satisfying $\mathbf{P} \mathbf{e}_l^i = \psi_l^i (\mathbf{M})$.*

*Proof.* By (Perko, 2001, Section 1.8), the solution can be written as

$$\exp\left(\mathbf{M}\,t\right)\mathbf{x}_0 = \mathbf{P}\exp\left(\mathbf{J}\,t\right)\mathbf{P}^{-1}\left(\sum_{l=1}^{m}\sum_{i=1}^{k_l}c_l^i\mathbf{P}\mathbf{e}_l^i\right) = \mathbf{P}\exp\left(\mathbf{J}\,t\right)\left(\sum_{l=1}^{m}\sum_{i=1}^{k_l}c_l^i\mathbf{e}_l^i\right),$$

where $\exp\left(\mathbf{J}\,t\right) = \operatorname{diag}\left(\{\exp\left(\mathbf{J}_l\,t\right)\}_{l=1}^{m}\right)$ and

$$\exp\left(\mathbf{J}_l\,t\right) = \exp\left(\lambda_l(\mathbf{M})\,t\right)\begin{bmatrix} 1 & t & \frac{t^2}{2!} & \cdots & \frac{t^{k_l}}{(k_l-1)!} \\ & 1 & t & & \vdots \\ & & 1 & \ddots & \frac{t^2}{2!} \\ & & & \ddots & t \\ & & & & 1 \end{bmatrix}.$$

Since $\exp\left(\mathbf{J}\,t\right) = \bigoplus_{l=1}^{m}\exp\left(\mathbf{J}_l\,t\right)$, we can focus on a single Jordan block. Fix $l \in \{1,\ldots,m\}$, it holds

$$\mathbf{P}\exp\left(\mathbf{J}_l\,t\right)\left(\sum_{i=1}^{k_l}c_l^i\mathbf{e}_l^i\right)$$

$$= \mathbf{P}\exp\left(\lambda_l(\mathbf{M})\,t\right)\left(c_l^1\mathbf{e}_l^1 + c_l^2\left(t\,\mathbf{e}_l^1 + \mathbf{e}_l^2\right) + c_l^3\left(\frac{t^2}{2!}\mathbf{e}_l^1 + t\,\mathbf{e}_l^2 + \mathbf{e}_l^3\right) + \ldots\right)$$

$$= \exp\left(\lambda_l(\mathbf{M})\,t\right)\left(c_l^1\psi_l^1(\mathbf{M}) + c_l^2\left(t\,\psi_l^1(\mathbf{M}) + \psi_l^2(\mathbf{M})\right)\right)$$

$$+ c_l^3\left(\frac{t^2}{2!}\psi_l^1(\mathbf{M}) + t\,\psi_l^2(\mathbf{M}) + \psi_l^3(\mathbf{M})\right) + \ldots\right)$$

$$= \exp\left(\lambda_l(\mathbf{M})\,t\right)\sum_{i=1}^{k_l}c_l^i\sum_{j=1}^{i}\frac{t^{i-j}}{(i-j)!}\psi_l^j(\mathbf{M})\,.$$

Bringing the direct sums together, we get

$$\exp\left(\mathbf{M}\,t\right)\mathbf{x}_0 = \sum_{l=1}^{m}\exp\left(\lambda_l(\mathbf{M})\,t\right)\sum_{i=1}^{k_l}c_l^i\sum_{j=1}^{i}\frac{t^{i-j}}{(i-j)!}\psi_l^j(\mathbf{M})\,,$$

from which the thesis follows. $\square$

In the following, we derive a formula for the solution of ODEs of the form

$$\mathbf{x}'(t) = \mathbf{M}\mathbf{x}(t)\mathbf{W}\,,\quad \mathbf{x}(0) = \mathbf{x}_0\,, \tag{13}$$

for a diagonal matrix $\mathbf{W} \in \mathbb{C}^{K\times K}$ and a general square matrix $\mathbf{M} \in \mathbb{C}^{N\times N}$ with Jordan normal form $\mathbf{P}\mathbf{J}\mathbf{P}^{-1}$. By vectorizing, we obtain the equivalent linear system

$$\operatorname{vec}(\mathbf{x})'(t) = \mathbf{W}\otimes\mathbf{M}\operatorname{vec}(\mathbf{x})(t)\,,\quad \operatorname{vec}(\mathbf{x})(0) = \operatorname{vec}(\mathbf{x}_0)\,. \tag{14}$$

Then, by properties of the Kronecker product, there holds

$$\mathbf{W}\otimes\mathbf{M} = \mathbf{W}\otimes(\mathbf{P}\mathbf{J}\mathbf{P}^{-1}) = (\mathbf{I}\otimes\mathbf{P})(\mathbf{W}\otimes\mathbf{J})(\mathbf{I}\otimes\mathbf{P}^{-1}) = (\mathbf{I}\otimes\mathbf{P})(\mathbf{W}\otimes\mathbf{J})(\mathbf{I}\otimes\mathbf{P})^{-1}.$$

Note that $(\mathbf{I}\otimes\mathbf{P})(\mathbf{W}\otimes\mathbf{J})(\mathbf{I}\otimes\mathbf{P})^{-1}$ is not the Jordan normal form of $\mathbf{D}\otimes\mathbf{M}$. However, we can characterize the Jordan form of $\mathbf{W}\otimes\mathbf{M}$ as follows.

**Lemma E.2.** *The Jordan decomposition of $\mathbf{W}\otimes\mathbf{J}$ is given by $\mathbf{W}\otimes\mathbf{J} = \tilde{\mathbf{P}}\tilde{\mathbf{J}}\tilde{\mathbf{P}}^{-1}$ where $\tilde{\mathbf{J}}$ is a block diagonal matrix with blocks*

$$\tilde{\mathbf{J}}_{j,l} = \begin{bmatrix} w_j\lambda_l(\mathbf{J}) & 1 \\ & w_j\lambda_l(\mathbf{J}) & 1 \\ & & \ddots \\ & & & w_j\lambda_l(\mathbf{J}) & 1 \\ & & & & w_j\lambda_l(\mathbf{J}) \end{bmatrix},$$

*and $\tilde{\mathbf{P}}$ is a diagonal matrix obtained by concatenating $\tilde{\mathbf{P}}_{j,l} = \operatorname{diag}\left(\{w_j^{-n+1}\}_{n=1}^{k_l}\right)$.*

*Proof.* As $\mathbf{J} = \bigoplus_{l=1}^{m} \mathbf{J}_l$, we can focus on a single Jordan block. Fix $l \in \{1, \ldots, m\}$. We have

$$\mathbf{W} \otimes \mathbf{J}_l = \operatorname{diag}\left(\{w_j\,\mathbf{J}_l\}_{j=1}^{K}\right) = \bigoplus_{j=1}^{K} w_j\mathbf{J}_l\,,$$

hence, we can focus once again on a single block. Fix $j \in \{1, \ldots, K\}$; the Jordan decomposition of $w_j\mathbf{J}_l$ is given by $\tilde{\mathbf{P}}_l = \operatorname{diag}\left(\{w_j^{-n+1}\}_{n=1}^{k_l}\right)$ and

$$\tilde{\mathbf{J}}_l = \begin{bmatrix} w_j\lambda_l(\mathbf{J}) & 1 \\ & w_j\lambda_l(\mathbf{J}) & 1 \\ & & \ddots & \ddots \\ & & & w_j\lambda_l(\mathbf{J}) & 1 \\ & & & & w_j\lambda_l(\mathbf{J}) \end{bmatrix}.$$

To verify it, compute the $(n, m)$ element

$$\left(\tilde{\mathbf{P}}_l\tilde{\mathbf{J}}_l\tilde{\mathbf{P}}_l^{-1}\right)_{n,m} = \sum_{i,k}\left(\tilde{\mathbf{P}}_l\right)_{n,i}\left(\tilde{\mathbf{J}}_l\right)_{i,k}\left(\tilde{\mathbf{P}}_l^{-1}\right)_{k,m}.$$

Since $\tilde{\mathbf{P}}_l$ is a diagonal matrix, the only non-null entries are on the diagonal; therefore, $i = n$ and $k = m$

$$= \left(\tilde{\mathbf{P}}_l\right)_{n,n}\left(\tilde{\mathbf{J}}_l\right)_{n,m}\left(\tilde{\mathbf{P}}_l^{-1}\right)_{m,m}$$

and the only non-null entries of $\tilde{\mathbf{J}}_l$ are when $m = n$ or $m = n + 1$, hence

$$= \begin{cases} \left(\tilde{\mathbf{P}}_l\right)_{n,n}\left(\tilde{\mathbf{J}}_l\right)_{n,n}\left(\tilde{\mathbf{P}}_l^{-1}\right)_{n,n} = w_j\lambda_l\left(\mathbf{J}\right), & m = n\,, \\ \left(\tilde{\mathbf{P}}_l\right)_{n,n}\left(\tilde{\mathbf{J}}_l\right)_{n,n+1}\left(\tilde{\mathbf{P}}_l^{-1}\right)_{n+1,n+1} = w_j^{-n+1}w_j^n = w_j\,, & m = n + 1\,. \end{cases}$$

The thesis follows from assembling the direct sums back. $\qquad\square$

Lemma E.2 leads to the following result that fully characterizes the solution of (14) in terms of the generalized eigenvectors and eigenvalues of $\mathbf{M}$ and $\mathbf{W}$.

**Proposition E.3.** *Consider* (14) *with* $\mathbf{M} = \mathbf{PJP}^{-1}$ *and* $\mathbf{W} \otimes \mathbf{J} = \tilde{\mathbf{P}}\tilde{\mathbf{J}}\tilde{\mathbf{P}}^{-1}$*, where* $\tilde{\mathbf{J}}$ *and* $\tilde{\mathbf{P}}$ *are given in Lemma E.2. The solution of* (14) *is*

$$\operatorname{vec}(\mathbf{x})(t) = \sum_{l_1=1}^{K}\sum_{l_2=1}^{m}\exp\left(\lambda_{l_1}(\mathbf{W})\lambda_{l_2}(\mathbf{M})t\right)\sum_{i=1}^{k_{l_2}}c_{l_1,l_2}^{i}\sum_{j=1}^{i}\frac{t^{i-j}}{(i-j)!}\left(\lambda_{l_1}(\mathbf{W})\right)^{1-j}\mathbf{e}_{l_1}\otimes\psi_{l_2}^{j}(\mathbf{M})\,,$$

*where the coefficients* $c_{l_1,l_2}^{i}$ *are given by*

$$\operatorname{vec}(\mathbf{x}_0) = \sum_{l_1=1}^{K}\sum_{l_2=1}^{m}\sum_{i=1}^{k_{l_2}}c_{l_1,l_2}^{i}(\mathbf{I}\otimes\mathbf{P})\tilde{\mathbf{P}}\mathbf{e}_{l_1}\otimes\mathbf{e}_{l_2}^{i}$$

*where* $\left\{\mathbf{e}_{l_2}^{i}\ :\ l_2\in\{1,\ldots,m\}\,,\ i\in\{1,\ldots,k_{l_2}\}\right\}$ *is the standard basis satisfying* $\mathbf{Pe}_{l_2}^{i} = \psi_{l_2}^{i}(\mathbf{M})$.

*Proof.* By Lemma E.2, the eigenvalues of $\mathbf{W}\otimes\mathbf{M}$ and the corresponding eigenvectors and generalized eigenvectors are

$$\lambda_{l_1}(\mathbf{W})\lambda_{l_2}(\mathbf{M})\,,\ \mathbf{e}_{l_1}\otimes\psi_{l_2}^{1}(\mathbf{M})\,,\ (\lambda_{l_1}(\mathbf{W}))^{-i+1}\mathbf{e}_{l_1}\otimes\psi_{l_2}^{i}(\mathbf{M})$$

for $l_1 \in \{1, \ldots, K\}$, $l_2 \in \{1, \ldots, m\}$ and $i \in \{2, \ldots, k_l\}$. Hence, by Lemma E.1, the solution of (14) is given by

$$\operatorname{vec}(\mathbf{x})(t) = \sum_{l_1=1}^{K}\sum_{l_2=1}^{m}\exp\left(\lambda_{l_2}(\mathbf{M})\lambda_{l_1}(\mathbf{W})t\right)\sum_{i=1}^{k_{l_2}}c_{l_1,l_2}^{i}\sum_{j=1}^{i}\frac{t^{i-j}}{(i-j)!}\left(\lambda_{l_1}(\mathbf{W})\right)^{1-j}(\mathbf{e}_{l_1}\otimes\psi_{l_2}^{j}(\mathbf{L}))\,,$$

where the coefficients $c_{l_1,l_2}^{i}$ are given by

$$\operatorname{vec}(\mathbf{x}_0) = \sum_{l_1=1}^{K}\sum_{l_2=1}^{m}\sum_{i=1}^{k_{l_2}}c_{l_1,l_2}^{i}(\mathbf{I}\otimes\mathbf{P})\tilde{\mathbf{P}}(\mathbf{e}_{l_1}\otimes\mathbf{e}_{l_2}^{i}(\mathbf{M}))\,.$$

$\qquad\square$

### E.1 Proof of Theorem 5.6

In the following, we reformulate and prove Theorem 5.6.

**Corollary E.4.** *Let $\mathcal{G}$ be a strongly connected directed graph with SNA $\mathbf{L} \in \mathbb{R}^{N \times N}$. Consider the initial value problem in (2) with diagonal channel mixing matrix $\mathbf{W} \in \mathbb{R}^{K \times K}$ and $\alpha = 1$. Then, for almost all initial values $\mathbf{x}_0 \in \mathbb{R}^{N \times K}$, the solution to (2) is HFD if*

$$\lambda_K(\mathbf{W})\Re\lambda_1(\mathbf{L}) < \lambda_1(\mathbf{W})\lambda_N(\mathbf{L})$$

*and $\lambda_1(\mathbf{L})$ is the unique eigenvalue that minimizes the real part among all eigenvalues of $\mathbf{L}$. Otherwise, the solution is LFD.*

*Proof.* Using the notation from Proposition E.3 and its proof, we can write the solution of the vectorized form of (2) as

$$\text{vec}(\mathbf{x})(t) = \sum_{l_1=1}^{K} \sum_{l_2=1}^{m} \exp\left(-\lambda_{l_1}(\mathbf{W})\lambda_{l_2}(\mathbf{L})t\right) \sum_{i=1}^{k_{l_2}} c_{l_1,l_2}^i \sum_{j=1}^{i} \frac{t^{i-j}}{(i-j)!}(\lambda_{l_1}(\mathbf{W}))^{1-j}(\mathbf{e}_{l_1} \otimes \psi_{l_2}^j(\mathbf{L})).$$

As done extensively, we separate the terms corresponding to the frequency with minimal real part. This frequency dominates as the exponential converges faster than polynomials for $t$ going to infinity. Consider the case $\lambda_K(\mathbf{W})\Re(\lambda_1(\mathbf{L})) < \lambda_1(\mathbf{W})\Re(\lambda_N(\mathbf{L}))$. As $\lambda_1(\mathbf{L})$ is unique, the product $\lambda_K(\mathbf{W})\Re(\lambda_1(\mathbf{L}))$ is the unique most negative frequency. Assume without loss of generality that $\lambda_K(\mathbf{W})$ has multiplicity one. The argument does not change for higher multiplicities as the corresponding eigenvectors are orthogonal since $\mathbf{W}$ is diagonal. Then, $\lambda_K(\mathbf{W})\lambda_1(\mathbf{L})$ has multiplicity one, and we calculate $\text{vec}(\mathbf{x})(t)$ as

$$\sum_{l_1=1}^{K} \sum_{l_2=1}^{m} \exp\left(-\lambda_{l_1}(\mathbf{W})\lambda_{l_2}(\mathbf{L})t\right) \sum_{i=1}^{k_{l_2}} c_{l_1,l_2}^i \sum_{j=1}^{i} \frac{t^{i-j}}{(i-j)!}(\lambda_{l_1}(\mathbf{W}))^{1-j}(\mathbf{e}_{l_1} \otimes \psi_{l_2}^j(\mathbf{L}))$$

$$= c_{K,1}^{k_1} \exp\left(-t\lambda_K(\mathbf{W})\lambda_1(\mathbf{L})\right) \frac{t^{k_1-1}}{(k_1-1)!}(\mathbf{e}_K \otimes \psi_1^1(\mathbf{L}))$$

$$+ c_{K,1}^{k_1} \exp\left(-t\lambda_K(\mathbf{W})\lambda_1(\mathbf{L})\right) \sum_{j=2}^{k_1} \frac{t^{k_1-j}}{(k_1-j)!}(\lambda_K(\mathbf{W}))^{1-j}(\mathbf{e}_K \otimes \psi_1^j(\mathbf{L}))$$

$$+ \exp\left(-t\lambda_K(\mathbf{W})\lambda_1(\mathbf{L})\right) \sum_{i=1}^{k_1-1} c_{K,1}^i \sum_{j=1}^{i} \frac{t^{i-j}}{(i-j)!}(\lambda_K(\mathbf{W}))^{1-j}(\mathbf{e}_K \otimes \psi_1^j(\mathbf{L}))$$

$$+ \sum_{l_1=1}^{K} \sum_{l_2=2}^{m} \exp\left(-\lambda_{l_1}(\mathbf{W})\lambda_{l_2}(\mathbf{L})t\right) \sum_{i=1}^{k_{l_2}} c_{l_1,l_2}^i \sum_{j=1}^{i} \frac{t^{i-j}}{(i-j)!}(\lambda_{l_1}(\mathbf{W}))^{1-j}(\mathbf{e}_{l_1} \otimes \psi_{l_2}^j(\mathbf{L}))$$

$$= \exp\left(-t\lambda_K(\mathbf{W})\lambda_1(\mathbf{L})\right) t^{k_1-1} \left( c_{K,1}^{k_1} \frac{1}{(k_1-1)!}(\mathbf{e}_K \otimes \psi_1^1(\mathbf{L})) \right.$$

$$+ c_{K,1}^{k_1} \sum_{j=2}^{k_1} \frac{t^{1-j}}{(k_1-j)!}(\lambda_K(\mathbf{W}))^{1-j}(\mathbf{e}_K \otimes \psi_1^j(\mathbf{L}))$$

$$+ \sum_{i=1}^{k_1-1} c_{K,1}^i \sum_{j=1}^{i} \frac{1}{(i-j)!} t^{i-j-k_1+1}(\lambda_K(\mathbf{W}))^{1-j}(\mathbf{e}_K \otimes \psi_1^j(\mathbf{L}))$$

$$+ \sum_{l_1=1}^{K} \sum_{l_2=2}^{m} \exp\left(-t(\lambda_{l_1}(\mathbf{W})\lambda_{l_2}(\mathbf{L}) - \lambda_K(\mathbf{W})\lambda_1(\mathbf{L}))\right) \sum_{i=1}^{k_{l_2}} c_{l_1,l_2}^i$$

$$\left. \cdot \sum_{j=1}^{i} \frac{t^{i-j-k_1+1}}{(i-j)!}(\lambda_{l_1}(\mathbf{W}))^{1-j}(\mathbf{e}_{l_1} \otimes \psi_{l_2}^j(\mathbf{L})) \right).$$

We can then write the normalized solution as

$$
\Bigg(\frac{c_{K,1}^{k_1}}{(k_1-1)!}(\mathbf{e}_K \otimes \psi_1^1(\mathbf{L})) + c_{K,1}^{k_1}\sum_{j=2}^{k_1}\frac{t^{1-j}}{(k_1-j)!}(\lambda_K(\mathbf{W}))^{1-j}(\mathbf{e}_K \otimes \psi_1^j(\mathbf{L}))
$$

$$
+\sum_{i=1}^{k_1-1} c_{K,1}^i \sum_{j=1}^{i}\frac{t^{i-j-k_1+1}}{(i-j)!}(\lambda_K(\mathbf{W}))^{1-j}(\mathbf{e}_K \otimes \psi_1^j(\mathbf{L}))
$$

$$
+\sum_{l_1=1}^{K}\sum_{l_2=2}^{m} e^{-t(\lambda_K(\mathbf{W})\lambda_{l_2}(\mathbf{L})-\lambda_K(\mathbf{W})\lambda_1(\mathbf{L}))}\sum_{i=1}^{k_{l_2}} c_{l_1,l_2}^i \sum_{j=1}^{i}\frac{t^{i-j-k_1}}{(i-j)!}(\lambda_{l_1}(\mathbf{W}))^{1-j}(\mathbf{e}_{l_1}\otimes \psi_{l_2}^j(\mathbf{L}))\Bigg)
$$

$$
\cdot\Bigg\|\frac{c_{K,1}^{k_1}}{(k_1-1)!}\left(\mathbf{e}_K \otimes \psi_1^1(\mathbf{L})\right) + c_{K,1}^{k_1}\sum_{j=2}^{k_1}\frac{t^{1-j}}{(k_1-j)!}(\lambda_K(\mathbf{W}))^{1-j}(\mathbf{e}_K \otimes \psi_1^j(\mathbf{L}))
$$

$$
+\sum_{i=1}^{k_1-1} c_{K,1}^i \sum_{j=1}^{i}\frac{t^{i-j-k_1+1}}{(i-j)!}(\lambda_{l_1}(\mathbf{W}))^{1-j}(\mathbf{e}_K \otimes \psi_1^j(\mathbf{L}))
$$

$$
+\sum_{l_1=1}^{K}\sum_{l_2=2}^{m}\exp\left(-t(\lambda_{l_1}(\mathbf{W})\lambda_{l_2}(\mathbf{L})-\lambda_K(\mathbf{W})\lambda_1(\mathbf{L}))\right)
$$

$$
\cdot\sum_{i=1}^{k_{l_2}} c_{l_1,l_2}^i \sum_{j=1}^{i}\frac{t^{i-j-k_1}}{(i-j)!}(\lambda_{l_1}(\mathbf{W}))^{1-j}(\mathbf{e}_{l_1}\otimes \psi_{l_2}^j(\mathbf{L}))\Bigg\|_2^{-1}.
$$

All summands, except the first, converge to zero for $t$ going to infinity. Hence,

$$
\frac{\mathrm{vec}(\mathbf{x})(t)}{\|\mathrm{vec}(\mathbf{x})(t)\|_2}\xrightarrow{t\to\infty}\left\|\frac{c_{K,1}^{k_1}}{(k_1-1)!}(\mathbf{e}_K \otimes \psi_1^1(\mathbf{L}))\right\|_2^{-1}\left(\frac{c_{K,1}^{k_1}}{(k_1-1)!}(\mathbf{e}_K \otimes \psi_1^1(\mathbf{L}))\right).
$$

We apply Lemma D.2 to finish the proof for the HFD case. Note that $\psi_1^1(\mathbf{L})$ is an eigenvector corresponding to $\lambda_1(\mathbf{L})$. The LFD case is equivalent. By Perron-Frobenius for irreducible non-negative matrices, there is no other eigenvalue with the same real part as $1-\lambda_N(\mathbf{L})=\lambda_1(\mathbf{I}-\mathbf{L})$. $\qquad\square$

**Remark E.5.** *If the hypotheses are met, the convergence result also holds for $\mathbf{L}^\alpha$. With the same reasoning, we can prove that the normalized solution converges to the eigenvector corresponding to the eigenvalue of $\mathbf{L}^\alpha$ with minimal real part. It suffices to consider the eigenvalues and generalized eigenvectors of $\mathbf{L}^\alpha$. However, we do not know the relationship between the singular values of $\mathbf{L}^\alpha$, where we defined the fractional Laplacian, and the eigenvalues of $\mathbf{L}$. Hence, it is much more challenging to draw conclusions on the Dirichlet energy.*

## E.2 Explicit Euler

In this subsection, we show that the convergence properties of the Dirichlet energy from Theorem 5.6 are also satisfied when (2) is approximated via an explicit Euler scheme.

As noted in (11), the vectorized solution to (2) can be written as

$$
\mathrm{vec}(\mathbf{x})(n\,h) = (\mathbf{I}-h\,(\mathbf{W}\otimes\mathbf{L}))^n\,\mathrm{vec}(\mathbf{x}_0),
$$

when $\alpha=1$. We thus aim to analyze the Jordan decomposition of $\mathbf{L}^n$ for $\mathbf{L}\in\mathbb{C}^{n\times n}$ and $n\in\mathbb{N}$. Let $\mathbf{L}=\mathbf{P}\mathbf{J}\mathbf{P}^{-1}$, where $\mathbf{J}$ is the Jordan form, and $\mathbf{P}$ is a invertible matrix of generalized eigenvectors.

Consider a Jordan block $\mathbf{J}_i$ associated with the eigenvalue $\lambda_i(\mathbf{M})$. For a positive integer $n$, the $n$-th power of the Jordan block can be computed as:

$$\mathbf{J}_l^n = \lambda_l(\mathbf{L})^n \begin{bmatrix} 1 & \binom{n}{1}\lambda_l(\mathbf{L})^{-1} & \binom{n}{2}\lambda_l(\mathbf{L})^{-2} & \cdots & \binom{n}{k_l-1}\lambda_l(\mathbf{L})^{-k_l+1} \\ & 1 & \binom{n}{1}\lambda_l(\mathbf{L})^{-1} & & \binom{n}{k_l-2}\lambda_l(\mathbf{L})^{-k_l+2} \\ & & 1 & & \vdots \\ & & & \ddots & \binom{n}{1}\lambda_l(\mathbf{L})^{-1} \\ & & & & 1 \end{bmatrix}$$

We compute the $n$-th power of $\mathbf{L}$ as $\mathbf{L}^n = (\mathbf{PJP}^{-1})^n = \mathbf{PJ}^n\mathbf{P}^{-1}$, and we expand $\mathbf{x}_0$ as

$$\mathbf{x}_0 = \sum_{l=1}^m \sum_{i=1}^{k_l} c_l^i \mathbf{Pe}_l^i,$$

where $\left\{ \mathbf{e}_l^i : i \in \{1, \ldots k_l\}, l \in \{1, \ldots, m\} \right\}$ is the standard basis and $\mathbf{Pe}_l^i = \psi_l^i(\mathbf{L})$ are the generalized eigenvectors of $\mathbf{L}$. It is easy to see that

$$\mathbf{L}^n\mathbf{x}_0 = \mathbf{PJ}^n\mathbf{P}^{-1}\left( \sum_{l=1}^m \sum_{i=1}^{k_l} c_l^i \mathbf{Pe}_l^i \right) = \mathbf{PJ}^n \left( \sum_{l=1}^m \sum_{i=1}^{k_l} c_l^i \mathbf{e}_l^i \right).$$

As $\mathbf{J}^n = \bigoplus_{l=1}^m \mathbf{J}_l^n$, we can focus on a single Jordan block. Fix $l \in \{1, \ldots, m\}$, and compute

$$\mathbf{PJ}_l^n \left( \sum_{i=1}^{k_l} c_l^i \mathbf{e}_l^i \right) = \mathbf{P}\left( \lambda_l(\mathbf{M})^n c_l^1 \mathbf{e}_l^1 \right) + \mathbf{P}\left( \binom{n}{1}\lambda_l(\mathbf{M})^{n-1} c_l^1 \mathbf{e}_l^1 + \lambda_l(\mathbf{M})^n c_l^2 \mathbf{e}_l^2 \right)$$

$$+ \mathbf{P}\left( \binom{n}{2}\lambda_l(\mathbf{M})^{n-2} c_l^1 \mathbf{e}_l^1 + \binom{n}{1}\lambda_l(\mathbf{L})^{n-1} c_l^2 \mathbf{e}_l^2 + \lambda_l(\mathbf{M})^n c_l^3 \mathbf{e}_l^3 \right)$$

$$+ \ldots.$$

We can summarize our findings in the following lemma.

**Lemma E.6.** *For any* $\mathbf{L} = \mathbf{PJP}^{-1} \in \mathbb{R}^{N \times N}$ *and* $\mathbf{x}_0 = \sum_{l=1}^m \sum_{i=1}^{k_l} c_l^i \psi_l^i(\mathbf{L})$, *we have*

$$\mathbf{L}^n\mathbf{x}_0 = \sum_{l=1}^m \sum_{i=1}^{\min\{k_l, n-1\}} \sum_{j=1}^i \binom{n}{i-j} \lambda_l(\mathbf{L})^{n-i+j} c_l^j \psi_l^j(\mathbf{L}).$$

We proceed with the main result of this subsection.

**Proposition E.7.** *Let* $\mathcal{G}$ *be a strongly connected directed graph with* SNA $\mathbf{L} \in \mathbb{R}^{N \times N}$. *Consider the initial value problem in* (2) *with diagonal channel mixing matrix* $\mathbf{W} \in \mathbb{R}^{K \times K}$ *and* $\alpha = 1$. *Approximate the solution to* (2) *with an explicit Euler scheme with a sufficiently small step size* $h$. *Then, for almost all initial values* $\mathbf{x}_0 \in \mathbb{C}^{N \times K}$ *the following holds. If* $\lambda_1(\mathbf{L})$ *is unique and*

$$\lambda_K(\mathbf{W})\Re\lambda_1(\mathbf{L}) < \lambda_1(\mathbf{W})\Re\lambda_N(\mathbf{L}), \tag{15}$$

*the approximated solution is* HFD. *Otherwise, the solution is* LFD.

*Proof.* As noted in (11), the vectorized solution to (2) with $\alpha = 1$, can be written as

$$\mathrm{vec}(\mathbf{x})(n\,h) = (\mathbf{I} - h\,(\mathbf{W} \otimes \mathbf{L}))^n \, \mathrm{vec}(\mathbf{x}_0).$$

Consider the Jordan decomposition of $\mathbf{L} = \mathbf{PJP}^{-1}$ and the Jordan decomposition of $\mathbf{W} \otimes \mathbf{J} = \tilde{\mathbf{P}}\tilde{\mathbf{J}}\tilde{\mathbf{P}}^{-1}$, where $\tilde{\mathbf{J}}$ and $\tilde{\mathbf{P}}$ are specified in Lemma E.2. Then,

$$\mathrm{vec}(\mathbf{x})(n\,h) = \left( \mathbf{I} + h\mathbf{W} \otimes (\mathbf{PJP}^{-1}) \right)^n \mathrm{vec}(\mathbf{x}_0)$$

$$= (\mathbf{I} \otimes \mathbf{P})(\mathbf{I} - h\mathbf{W} \otimes \mathbf{J})^n (\mathbf{I} \otimes \mathbf{P})^{-1}\mathrm{vec}(\mathbf{x}_0)$$

$$= (\mathbf{I} \otimes \mathbf{P})(\mathbf{I} - h\tilde{\mathbf{P}}\tilde{\mathbf{J}}\tilde{\mathbf{P}}^{-1})^n(\mathbf{I} \otimes \mathbf{P})^{-1}\mathrm{vec}(\mathbf{x}_0)$$

$$= (\mathbf{I} \otimes \mathbf{P})\tilde{\mathbf{P}}(\mathbf{I} - h\tilde{\mathbf{J}})^n\tilde{\mathbf{P}}^{-1}(\mathbf{I} \otimes \mathbf{P})^{-1}\mathrm{vec}(\mathbf{x}_0)$$

$$= (\mathbf{I} \otimes \mathbf{P})\tilde{\mathbf{P}}(\mathbf{I} - h\tilde{\mathbf{J}})^n((\mathbf{I} \otimes \mathbf{P})\tilde{\mathbf{P}})^{-1}\mathrm{vec}(\mathbf{x}_0).$$

Now, decompose $\mathbf{x}_0$ into the basis of generalized eigenvectors, i.e.,

$$\mathrm{vec}(\mathbf{x}_0) = \sum_{l_1=1}^{K} \sum_{l_2=1}^{m} \sum_{i=1}^{k_{l_2}} c_l^i ((\mathbf{I} \otimes \mathbf{P})\tilde{\mathbf{P}})(\mathbf{e}_{l_1} \otimes \mathbf{e}_{l_2}^i(\mathbf{L})).$$

Then, by Lemma E.6, we have

$$\mathrm{vec}(\mathbf{x})(n\,h) = \sum_{l_1=1}^{K} \sum_{l_2=1}^{m} \sum_{i=1}^{\min\{k_{l_2},t-1\}} \sum_{j=1}^{i} \binom{n}{i-j} (1 - h\lambda_{l_1}(\mathbf{W})\lambda_{l_2}(\mathbf{L}))^{n-i+j} c_{l_1,l_2}^j$$
$$(\lambda_{l_1}(\mathbf{W}))^{1-j}\,\psi_{l_1}(\mathbf{W}) \otimes \psi_{l_2}^j(\mathbf{L}).$$

Now, consider the maximal frequency, i.e.,

$$L_1, L_2 = \arg\max_{l_1,l_2} \{|1 - h\lambda_{l_1}(\mathbf{W})\lambda_{l_2}(\mathbf{L})|\}.$$

Then, the solution $\mathrm{vec}(\mathbf{x})(n\,h)$ can be written as

$$\sum_{l_1=1}^{K} \sum_{l_2=1}^{m} \sum_{i=1}^{\min\{k_{l_2},n-1\}} \sum_{j=1}^{i} \binom{n}{i-j} (1 - h\lambda_{l_1}(\mathbf{W})\lambda_{l_2}(\mathbf{L}))^{n-i+j} c_{l_1,l_2}^j \psi_{l_1}(\mathbf{W}) \otimes \psi_{l_2}^j(\mathbf{L})$$
$$= (1 - h\lambda_{L_1}(\mathbf{W})\lambda_{L_2}(\mathbf{L}))^n \tag{16}$$
$$\cdot \sum_{l_1=1}^{K} \sum_{l_2=1}^{m} \sum_{i=1}^{\min\{k_{l_2},t-1\}} \sum_{j=1}^{i} \binom{n}{i-j} \frac{(1 - h\lambda_{l_1}(\mathbf{W})\lambda_{l_2}(\mathbf{L}))^{n-i+j}}{(1 - h\lambda_{L_1}(\mathbf{W})\lambda_{L_2}(\mathbf{L}))^n} c_{l_1,l_2}^j \psi_{l_1}(\mathbf{W}) \otimes \psi_{l_2}^j(\mathbf{L}).$$

With a similar argument as in the proof of Theorem 5.6, we can then see that

$$\frac{\mathrm{vec}(\mathbf{x})(n\,h)}{\|\mathrm{vec}(\mathbf{x})(n\,h)\|_2} \xrightarrow{t\to\infty} \frac{c_{L_1,L_2}^1\,\psi_{L_1}(\mathbf{W}) \otimes \psi_{L_2}^1(\mathbf{L})}{\|c_{L_1,L_2}^1\,\psi_{L_1}(\mathbf{W}) \otimes \psi_{L_2}^1(\mathbf{L})\|_2},$$

where $\psi_{L_2}^1(\mathbf{L})$ is the eigenvector corresponding to $\lambda_{L_2}(\mathbf{L})$. Note that for almost all $\mathbf{x}$, we have $c_{L_1,L_2}^1 \neq 0$. Then $\psi_{L_2}^1(\mathbf{L})$ is also an eigenvector of $\mathbf{I} - \mathbf{L}$ corresponding to the eigenvalue $1 - \lambda_{L_2}(\mathbf{L})$. By Lemma D.2, we have that the approximated solution is $(1 - \lambda_{L_2}(\mathbf{L}))$-FD.

We finish the proof by showing that $L_2 = 1$ if (15) is satisfied, and $L_2 = N$ otherwise. First, note that either $\lambda_K(\mathbf{W})\Re\lambda_1(\mathbf{L})$ or $\lambda_1(\mathbf{W})\Re\lambda_N(\mathbf{L})$ are the most negative real parts among all $\{\lambda_l(\mathbf{W})\Re\lambda_r(\mathbf{L})\}_{l\in\{1,\ldots,K\},r\in\{1\ldots,N\}}$. Assume first that $\lambda_K(\mathbf{W})\Re\lambda_1(\mathbf{L})$ has the most negative real part, i.e., (15) holds. Then, define

$$\varepsilon := \max_{l,r} |\lambda_K(\mathbf{W})\Re\lambda_1(\mathbf{L}) - \lambda_l(\mathbf{W})\Re\lambda_r(\mathbf{L})|,$$

and assume $h < \varepsilon\|\mathbf{W}\|^2$. Now it is easy to see that

$$2\lambda_K(\mathbf{W})\Re\lambda_1(\mathbf{L}) - h\lambda_K(\mathbf{W})^2|\lambda_1(\mathbf{L})|^2 < 2\lambda_l(\mathbf{W})\Re\lambda_r(\mathbf{L}) - h\lambda_l(\mathbf{W})^2|\lambda_r(\mathbf{L})|^2,$$

which is equivalent to $(K, 1) = (L_1, L_2)$. Hence, the dynamics are $(1 - \lambda_1(\mathbf{L}))$-FD. As $(1 - \lambda_1(\mathbf{L}))$ is highest frequency of $\mathbf{I} - \mathbf{L}$, we get HFD dynamics. Similary, we can show that if $\lambda_1(\mathbf{W})\Re\lambda_N(\mathbf{L})$ is the most negative frequency, we get LFD dynamics. Note that for the HFD argument, we must assume that $\lambda_1(\mathbf{L})$ is the unique eigenvalue with the smallest real part. For the LFD argument, it is already given that $\lambda_N(\mathbf{L})$ has multiplicity one by Perron-Frobenius Theorem. $\qquad\square$

### E.3 GCN oversmooths

**Proposition E.8.** *Let $\mathcal{G}$ be a strongly connected and aperiodic directed graph with SNA $\mathbf{L} \in \mathbb{R}^{N \times N}$. A GCN with the update rule*

$$\mathbf{x}_{t+1} = \mathbf{L}\mathbf{x}_t \mathbf{W},$$

*where $\mathbf{x}_0 \in \mathbb{R}^{N \times K}$ are the input node features, always oversmooths.*

*Proof.* The proof follows similarly to the proof of Proposition E.7. The difference is that instead of (16), we can write the node features after $t$ layers as

$$\text{vec}(\mathbf{x}_t) = \sum_{l_1=1}^{K} \sum_{l_2=1}^{m} \sum_{i=1}^{\min\{k_{l_2}, t-1\}} \sum_{j=1}^{i} \binom{t}{i-j} (\lambda_{l_1}(\mathbf{W})\lambda_{l_2}(\mathbf{L}))^{t-i+j} c_{l_1,l_2}^{j} \psi_{l_1}(\mathbf{W}) \otimes \psi_{l_2}^{j}(\mathbf{L}).$$

Now note that by Perron-Frobenius the eigenvalue $\lambda_N(\mathbf{L})$ with the largest absolute value is real and has multiplicity one. Then, $\max_{l_1,l_2} |\lambda_{l_1}(\mathbf{W})\lambda_{l_2}(\mathbf{L})|$ is attained at either $\lambda_1(\mathbf{W})\lambda_N(\mathbf{L})$ or $\lambda_K(\mathbf{W})\lambda_N(\mathbf{L})$. Equivalently to the proof of Proposition E.7, we can show that the corresponding GCN is $1 - \lambda_N(\mathbf{L})$-FD. Now $1 - \lambda_N(\mathbf{L}) = \lambda_1(\mathbf{I} - \mathbf{L})$, and $\lambda_1(\mathbf{I} - \mathbf{L})$-FD corresponds to LFD, hence the GCN oversmooths. $\qquad\square$

## F  Appendix for the Cycle Graph Example

Consider the cycle graph with $N$ nodes numbered from $0$ to $N - 1$. Since each node has degree 2, the SNA $\mathbf{L} = \mathbf{A}/2$ is a circulant matrix produced by the vector $\mathbf{v} = (\mathbf{e}_1 + \mathbf{e}_{N-1})/2$. Denote $\omega = \exp(2\pi i/N)$, the eigenvectors can be computed as

$$\mathbf{v}_j = \frac{1}{\sqrt{N}} \left(1, \omega^j, \omega^{2j}, \dots, \omega^{(N-1)j}\right)$$

asociated to the eigenvalue $\lambda_j = \cos(2\pi j/N)$. First, we can note that $\lambda_j = \lambda_{N-j}$ for all $j \in \{1, \dots, N/2\}$; therefore, the multiplicity of each eigenvalue is 2 except $\lambda_0$ and, if $N$ is even, $\lambda_{N/2}$. Since the original matrix is symmetric, there exists a basis of real eigenvectors. A simple calculation

$$\mathbf{L}\Re\mathbf{v}_j + i\mathbf{L}\Im\mathbf{v}_j = \mathbf{L}\mathbf{v}_j = \lambda_j \mathbf{v}_j = \lambda_j \Re\mathbf{v}_j + i\lambda_j \Im\mathbf{v}_j$$

shows that $\Re\mathbf{v}_j$ and $\Im\mathbf{v}_j$, defined as

$$\Re\mathbf{v}_j = \frac{1}{\sqrt{N}} \left(\cos\left(\frac{2\pi j\, n}{N}\right)\right)_{n=0}^{N-1}, \quad \Im\mathbf{v}_j = \frac{1}{\sqrt{N}} \left(\sin\left(\frac{2\pi j\, n}{N}\right)\right)_{n=0}^{N-1}$$

are two eigenvectors of the same eigenvalue $\lambda_j$. To show that they are linearly independent, we compute under which conditions

$$0 = a\Re\mathbf{v}_j + b\Im\mathbf{v}_j\,.$$

We note that the previous condition implies that for all $n \notin \{0, N/2\}$

$$0 = a\cos\left(\frac{2\pi j\, n}{N}\right) + b\sin\left(\frac{2\pi j\, n}{N}\right)$$

$$= \sqrt{a^2 + b^2} \sin\left(\frac{2\pi j\, n}{N} + \arctan\left(\frac{b}{a}\right)\right)$$

Suppose $a, b \neq 0$, then it must be

$$\frac{2\pi j\, n}{N} + \arctan\left(\frac{b}{a}\right) = k\pi\,, \ k \in \mathbb{Z}$$

which is equivalent to

$$2j\, n = \left(k - \frac{\arctan\left(\frac{b}{a}\right)}{\pi}\right) N\,, \ k \in \mathbb{Z}$$

The left-hand side is always an integer, while the right-hand side is an integer if and only if $b = 0$. This reduces the conditions to

$$\begin{cases} a \cos\left(\dfrac{2\pi j\, n}{N}\right) = 0 \\ |a| \sin\left(\dfrac{2\pi j\, n}{N}\right) = 0 \end{cases}$$

which is true if and only if $a = 0$. Consider now an even number of nodes $N$; the eigenspace of $\lambda_{N/2} = -1$ is

$$\mathbf{v}_{N/2} = \frac{1}{\sqrt{N}} \left((-1)^n\right)_{n=0}^{N-1}$$

hence, the maximal eigenvector of $\mathbf{I} - \mathbf{L}$ guarantees homophily $0$. Consider now a number of nodes $N$ divisible by $4$; the eigenspace of $\lambda_{N/4} = 0$ has basis

$$\Re \mathbf{v}_{N/4} = \frac{1}{\sqrt{N}} \left(\cos\left(\frac{\pi n}{2}\right)\right)_{n=0}^{N-1} \,,\quad \Im \mathbf{v}_{N/4} = \frac{1}{\sqrt{N}} \left(\sin\left(\frac{\pi n}{2}\right)\right)_{n=0}^{N-1}$$

Their sum is then equivalent to

$$\begin{aligned}
\Re \mathbf{v}_{N/4} + \Im \mathbf{v}_{N/4} &= \frac{1}{\sqrt{N}} \left(\cos\left(\frac{\pi n}{2}\right) + \sin\left(\frac{\pi n}{2}\right)\right)_{n=0}^{N-1} \\
&= \frac{\sqrt{2}}{\sqrt{N}} \left(\sin\left(\frac{\pi n}{2} + \frac{\pi}{4}\right)\right)_{n=0}^{N-1} \\
&= \sqrt{\frac{2}{N}} \left(\sin\left(\frac{\pi}{4}(2n+1)\right)\right)_{n=0}^{N-1} \\
&= \frac{1}{\sqrt{N}} \left(1, 1, -1, -1, \ldots\right)
\end{aligned}$$

hence, the mid eigenvector of $\mathbf{L}$ guarantees homophily $1/2$. A visual explanation is shown in Figure 4.

