# Acronyms

AUROC    Area under the ROC curve

DSBM    Directed Stochastic Block Model

FD    Frequency Dominant
FGL    Fractional Graph Laplacian

GAT    Graph Attention Network
GCN    Graph Convolutional Network
GNN    Graph Neural Network

HFD    Highest Frequency Dominant

LCC    Largest Connected Components
LFD    Lowest Frequency Dominant

MLP    Multi-Layer Perceptron

ODE    Ordinary Differential Equation

SNA    Symmetrically Normalized Adjacency
SVD    Singular Value Decomposition

# Notation

| | |
|---|---|
| $i$ | Imaginary unit |
| $\Re(z)$ | Real part of $z \in \mathbb{C}$ |
| $\Im(z)$ | Imaginary part of $z \in \mathbb{C}$ |
| $\mathrm{diag}(\mathbf{x})$ | Diagonal matrix with $\mathbf{x}$ on the diagonal. |
| $\mathbf{1}$ | Constant vector of all $1$s. |
| $\mathbf{M}^{\mathsf{T}}$ | Transpose of $\mathbf{M}$ |
| $\mathbf{M}^{*}$ | Conjugate of $\mathbf{M}$ |
| $\mathbf{M}^{\mathsf{H}}$ | Conjugate transpose of $\mathbf{M}$ |
| $\|\mathbf{M}\|$ | Spectral norm of $\mathbf{M}$ |
| $\|\mathbf{M}\|_{2}$ | Frobenius norm of $\mathbf{M}$ |
| $\lambda(\mathbf{M})$ | Spectrum of $\mathbf{M}$ |
| $\sigma(\mathbf{M})$ | Singular values of $\mathbf{M}$ |
| $\mathscr{E}(\mathbf{x})$ | Dirichlet energy computed on $\mathbf{x}$ |
| $\mathscr{H}(\mathcal{G})$ | Homophily coefficient of the graph $\mathcal{G}$ |
| $\mathbf{A} \otimes \mathbf{B}$ | Kronecker product between $\mathbf{A}$ and $\mathbf{B}$ |
| $\mathrm{vec}(\mathbf{M})$ | Vector obtained stacking columns of $\mathbf{M}$. |

# A   Implementation Details

In this section, we give the details on the numerical results in Section 6. We begin by describing the exact model.

**Model architecture.**   Let $\mathcal{G}$ be a directed graphs and $\mathbf{x}_0 \in \mathbb{R}^{N \times K}$ the node features. Our architecture first embeds the input node features $\mathbf{x}$ via a multi-layer perceptron (MLP). We then evolve the features $\mathbf{x}_0$ according to a slightly modified version of (3), i.e, $\mathbf{x}'(t) = -i\,\mathbf{L}^{\alpha}\mathbf{x}(t)\mathbf{W}$ for some time $t \in [0, T]$. In our experiments, we approximate the solution with an Explicit Euler scheme with step size $h > 0$. This leads to the following update rule

$$\mathbf{x}_{t+1} = \mathbf{x}_t - ih\mathbf{L}^{\alpha}\mathbf{x}_t\mathbf{W}\,.$$

The channel mixing matrix is a diagonal learnable matrix $\mathbf{W} \in \mathbb{C}^{K \times K}$, and $\alpha \in \mathbb{R}$, $h \in \mathbb{C}$ are also learnable parameters. The features at the last time step $\mathbf{x}_T$ are then fed into a second MLP, whose output is used as the final output. Both MLPs use LeakyReLU as non-linearity and dropout (Srivastava et al., 2014). On the contrary, the graph layers do not use any dropout nor non-linearity. A sketch of the algorithm is reported in fLode.

---

**Algorithm 1:** fLode

```
  % A, x₀ are given.
  % Preprocessing
1 D_in  = diag (A1)
2 D_out = diag (Aᵀ1)
3 L  = D_in^(-1/2) A D_out^(-1/2)
4 U, Σ, Vᴴ  = svd(L)
  % The core of the algorithm is very simple
5 def training_step(x₀):
6     x₀ = input_MLP(x₀)
7     for t ∈ {1,...,T} do
8         x_t  = x_{t-1} - i h UΣᵅVᴴx_{t-1}W
9     x_T = output_MLP(x_T)

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

$$\operatorname*{arg\,max}_{(k,n) \in [K] \times [N]} \{\Re\left(i\, \lambda_{k,n}\right)\} = \operatorname*{arg\,min}_{(k,n) \in \in [K] \times [N]} \{\Im\left(\lambda_{k,n}\right)\} .$$

$\qquad\square$

### D.1.1 Proof of Theorem 5.3

We denote the eigenvalues of $\mathbf{L}$ closest to $0$ from above and below as

$$\lambda_+(\mathbf{L}) := \arg\min_l \left\{ \lambda_l(\mathbf{L}) \ : \ \lambda_l(\mathbf{L}) > 0 \right\},$$
$$\lambda_-(\mathbf{L}) := \arg\max_l \left\{ \lambda_l(\mathbf{L}) \ : \ \lambda_l(\mathbf{L}) < 0 \right\}. \tag{9}$$

We assume that the channel mixing $\mathbf{W} \in \mathbb{R}^{K \times K}$ and the graph Laplacians $\mathbf{L}, \mathbf{I} - \mathbf{L} \in \mathbb{R}^{N \times N}$ are real matrices. Finally, we suppose the eigenvalues of a generic matrix $\mathbf{M}$ are sorted in ascending order, i.e., $\lambda_i(\mathbf{M}) \leq \lambda_j(\mathbf{M})$, $i < j$.

We now reformulate Theorem 5.3 for the fractional heat equation (2) and provide its full proof, which follows a similar frequency analysis to the one in (Di Giovanni et al., 2023, Theorem B.3)

**Theorem D.5.** *Let $\mathcal{G}$ be an undirected graph with SNA $\mathbf{L}$. Consider the initial value problem in (2) with channel mixing matrix $\mathbf{W} \in \mathbb{R}^{K \times K}$ and $\alpha \in \mathbb{R}$. Then, for almost all initial conditions $\mathbf{x}_0 \in \mathbb{R}^{N \times K}$ the following is satisfied.*

$(\alpha > 0)$ *The solution to (2) is HFD if*

$$\lambda_K(\mathbf{W}) \, \mathrm{f}_\alpha \left( \lambda_1(\mathbf{L}) \right) < \lambda_1(\mathbf{W}),$$

*and LFD otherwise.*

$(\alpha < 0)$ *The solution to (2) is $(1 - \lambda_-(\mathbf{L}))$-FD if*

$$\lambda_K(\mathbf{W}) \, \mathrm{f}_\alpha \left( \lambda_-(\mathbf{L}) \right) < \lambda_1(\mathbf{W}) \, \mathrm{f}_\alpha \left( \lambda_+(\mathbf{L}) \right),$$

*and $(1 - \lambda_+(\mathbf{L}))$-FD otherwise.*

*Proof of $(\alpha > 0)$.* As derived in (8), the solution of (2) with initial condition $\mathbf{x}_0$ can be written in a vectorized form as

$$\mathrm{vec}(\mathbf{x})(t) = \exp\left( -t \, \mathbf{W}^\mathsf{T} \otimes \mathbf{L}^\alpha \right) \mathrm{vec}(\mathbf{x}_0)$$
$$= \sum_{r=1}^{K} \sum_{l=1}^{N} c_{r,l} \, \exp\left( -t \, \lambda_r(\mathbf{W}) \, \mathrm{f}_\alpha \left( \lambda_l(\mathbf{L}) \right) \right) \psi_r(\mathbf{W}) \otimes \psi_l(\mathbf{L}),$$

where $\lambda_r(\mathbf{W})$ are the eigenvalues of $\mathbf{W}$ with corresponding eigenvectors $\psi_r(\mathbf{W})$, and $\lambda_l(\mathbf{L})$ are the eigenvalues of $\mathbf{L}$ with corresponding eigenvectors $\psi_l(\mathbf{L})$. The coefficients $c_{r,l}$ are the Fourier coefficients of $\mathbf{x}_0$, i.e.,

$$c_{r,l} := \langle \mathrm{vec}(\mathbf{x}_0), \ \psi_r(\mathbf{W}) \otimes \psi_l(\mathbf{L}) \rangle.$$

The key insight is to separate the eigenprojection corresponding to the most negative frequency. By Lemma D.3, this frequency component dominates for $t$ going to infinity.

Suppose

$$\lambda_K(\mathbf{W}) \, \mathrm{f}_\alpha \left( \lambda_1(\mathbf{L}) \right) < \lambda_1(\mathbf{W}) \, \mathrm{f}_\alpha \left( \lambda_N(\mathbf{L}) \right) = \lambda_1(\mathbf{W}).$$

In this case, $\lambda_K(\mathbf{W}) \, \mathrm{f}_\alpha \left( \lambda_1(\mathbf{L}) \right)$ is the most negative frequency. Assume for simplicity that $\lambda_K(\mathbf{W})$ has multiplicity one; the argument can be applied even if this is not the case, since the corresponding eigenvectors are orthogonal for higher multiplicities.

For almost all initial conditions $\mathbf{x}_0$, the coefficient $c_{K,1}$ is not null; hence

$$\frac{\mathrm{vec}(\mathbf{x})(t)}{\|\mathrm{vec}(\mathbf{x})(t)\|_2} \xrightarrow{t \to \infty} \frac{c_{K,1} \, \psi_K(\mathbf{W}) \otimes \psi_1(\mathbf{L})}{\|c_{K,1} \, \psi_K(\mathbf{W}) \otimes \psi_1(\mathbf{L})\|_2}.$$

By standard properties of the Kronecker product, we have

$$(\mathbf{I} \otimes \mathbf{L}) \left( \psi_K(\mathbf{W}) \otimes \psi_1(\mathbf{L}) \right) = \left( \mathbf{I} \, \psi_K(\mathbf{W}) \right) \otimes \left( \mathbf{L} \, \psi_1(\mathbf{L}) \right) = \