# OpenReview forum: "A Fractional Graph Laplacian Approach to Oversmoothing"
_NeurIPS.cc/2023/Conference — NeurIPS 2023 poster_

### Official Review · Reviewer_PWaE · 2023-06-26

**Soundness:** 2 fair
**Presentation:** 2 fair
**Contribution:** 2 fair
**Rating:** 3
**Confidence:** 5

**Summary:**

The authors propose two novel Fractional Graph Laplacian (FGL)-based neural Ordinary Differential Equations (ODEs): the fractional heat equation and the fractional Schrödinger equation. These solutions provide enhanced flexibility in the convergence of the Dirichlet energy and make the exponent of the fractional graph Laplacian a learnable parameter. This allows the network to adaptively decide the optimal exponent for a specific task and graph. The fractional graph Laplacian operator generalizes the Laplacian operator. The experimental results highlight the improvements of using fractional graph Laplacians, but the benefit is limited.

**Strengths:**

1. The proposed FGL Neural ODE can be $\lambda-FD$ which extends the Neural ODE-based GNNs that are limited to being either LFD or HFD.
2. This paper generalizes the concepts related oversmoothing from undirected graphs to directed graphs.


**Weaknesses:**

1. The application of the fractional graph Laplacian operator results in the loss of the graph's sparse property, transforming it into a dense even complete graph.
2. The paper's primary contribution appears to target oversmoothing issues, as suggested by the title. However, the evidence supporting the claim that the fractional graph Laplacian can mitigate oversmoothing in GNN is insufficient.The paper posits that the theory and experimental results presented counteract the issue of oversmoothness. However, based on [1], oversmoothness can still occur even when node features exist within an invariant space that has a rank greater than one. In this context, the paper's conclusions about oversmoothness mitigation defined using Dirichlet energy appear quite constrained. How would the proposed model fare with 256 layers stacked? How does it measure against the ODE-based methods (some are also designed to address oversmoothing), such as the recent work like GRAND[2], GraphCON [3], GRAFF [4], GRAND++[5] and GREAD [6]?
3. While the paper introduces concepts pertaining to directed graphs, the significant Theorem 5.3 only applies to undirected graphs, and Theorem 5.5 is solely restricted to the standard Laplacian where $\alpha=1$. Although it seems that the Fractional Graph Laplacian (FGL) may bridge the gap between homophilic and heterophilic graphs, the benefits of using FGL for directed graphs aren't clearly established.
4. The authors introduce several definitions, notably the Dirichlet energy for a directed graph (equation 1), but fail to provide solid justification for their choice. For instance, one could question why the energy is not defined in the following way:
$$\sum_{i, j=1}^N a_{i, j}\left(\left\|\frac{\mathbf{x}_i}{\sqrt{d_i^{\text {in }}}}-\frac{\mathbf{x}_j}{\sqrt{d_j^{\text {in }}}}\right\|_2^2 +\left\|\frac{\mathbf{x}_i}{\sqrt{d_i^{\text {out }}}}-\frac{\mathbf{x}_j}{\sqrt{d_j^{\text {out }}}}\right\|_2^2\right).$$
The experimental results provided do not sufficiently substantiate these definitions or the theoretical assertions made in the paper.
5. Very poor experimental results: While the paper focuses on directed graph Laplacian, the enhancements it introduces seem to be quite narrow, especially when considering Table 2, the only table for directed graphs. The inclusion of several definitions and theories seems to add complexity to the paper without providing a substantial contribution to the GNN community. The authors also need to include the SOTA results on those datasets.
6. The usage of complex numbers, as in equation (5), should be justified convincingly. While this may be commonplace in pure mathematics or physics, it needs to be shown why such a model is necessary for tackling oversmoothness in GNNs. Recent developments, such as Pytorch's support for complex numbers, do not automatically warrant their use in all contexts. If equation (5) is utilized, it only increases the model's complexity and memory consumption. Therefore, a robust justification is needed.

[1] Oono, Kenta, and Taiji Suzuki. "Graph neural networks exponentially lose expressive power for node classification." arXiv preprint arXiv:1905.10947 (2019).



[2] Chamberlain, Ben, et al. "Grand: Graph neural diffusion." International Conference on Machine Learning. PMLR, 2021.



[3] Rusch, T. Konstantin, et al. "Graph-coupled oscillator networks." International Conference on Machine Learning. PMLR, 2022.



[4] Di Giovanni, F., Rowbottom, J., Chamberlain, B. P., Markovich, T., and Bronstein, M. M. (2022). Graph Neural Networks as Gradient Flows: Understanding Graph Convolutions via Energy. arXiv: 2206.10991 [cs, stat].



[5] Thorpe, Matthew, et al. "GRAND++: Graph neural diffusion with a source term." International Conference on Learning Representations. 2021.



[6] Choi, Jeongwhan, et al. "GREAD: Graph Neural Reaction-Diffusion Networks." (2023).


**Questions:**

1. In Table 1, could you provide the learned values for $\lambda_{K}(W)$ and $\lambda_{1}(W)$ for each dataset? Moreover, could you offer a detailed analysis of how these results relate to Theorem 5.3? Also, all the learned $\alpha$ values in Table 4 are greater than 0, so according to part(i) of Theorem 5.3, the FGL-ODE is still either L-FD or H-FD. Could you clarify under which circumstances the proposed $\lambda$-FD could be applied?
2. The paper doesn't clearly illustrate how the fractional Laplacian operator mitigates the oversmoothing issue, both from theoretical and experimental perspectives. More in-depth explanations and experiments are needed.
3. Is the FGL-based Neural ODE applicable to large-scale graph datasets, such as those in references [1] and [2]?
4. The results for the Cora dataset presented in Table 1 are inferior to MLP. Could you also display the results for the homophilic dataset using the data split method from reference [5]?
5. Table 1 seems to miss important baselines for heterophilic graphs, such as ACM-GNN[3] and ACMP[4]. It would be beneficial to include these.
6. The paper lacks ablation studies. Could you provide the node classification results for $\alpha=1$ in equations (4) and (5) of FLODE? As the main contribution appears to be the FGL, are there any ablation studies that use layer-wise FGL-based GNNs instead of the FGL Neural ODE?
7. Other questions please refer to weakness part.







[1].Hu W, Fey M, Zitnik M, et al. Open graph benchmark: Datasets for machine learning on graphs[J]. Advances in neural information processing systems, 2020, 33: 22118-22133.
[2].Lim D, Hohne F, Li X, et al. Large scale learning on non-homophilous graphs: New benchmarks and strong simple methods[J]. Advances in Neural Information Processing Systems, 2021, 34: 20887-20902.
[3].Luan S, Hua C, Lu Q, et al. Revisiting heterophily for graph neural networks[J]. arXiv preprint arXiv:2210.07606, 2022.
[4].Wang Y, Yi K, Liu X, et al. ACMP: Allen-Cahn Message Passing for Graph Neural Networks with Particle Phase Transition[J]. arXiv preprint arXiv:2206.05437, 2022.
[5]. Shchur O, Mumme M, Bojchevski A, et al. Pitfalls of graph neural network evaluation[J]. arXiv preprint arXiv:1811.05868, 2018.

**Limitations:**

Yes,  the authors addressed the limitations.

---

> ### Author Rebuttal · Authors · 2023-08-07
>
> We are very grateful to the reviewer for the time taken to carefully assess our work and for the valuable feedback. We address each point individually. “W/Q” numbers the weakness or question, followed by our response.
>
> ---
>
> **W1**: Using FGLs does not lead to a loss of sparsity across all graphs: the graph's density incrementally increases as $\alpha$ goes to 0, see Appendix A, Fig. 6. Moreover, we've demonstrated in Thm. 4.1 that the FGL still respects the topology of the underlying graph. We also argue that adding some “virtual” edges can be advantageous for heterophilic graphs as nodes far apart in the input graph can directly interact.
>
> ---
>
> **W2**: The notion of oversmoothing (OS) in [1] and the Dirichlet energy are intimately connected, see [Cai et al., 2020]. When the invariant subspace is the kernel of the SNL this corresponds to OS (wrt the decay of the Dirichlet energy). While the invariant subspace is always the kernel in standard GNNs, we show that this is not true for our method! This mitigates *provably* OS. To more directly address your query, Fig. 1 in the supplementary pdf illustrates that the performance of our model does not deteriorate with 256 layers. Please also note that Dirichlet energy is a standard notion to measure OS, see, e.g., [3,4] that you mention.
>
> Regarding the comparisons, [2,4] are already included in Tab. 1, and we add [3,5,6] to the already discussed models designed to mitigate OS, see also Point 4 in the general response.
>
> ---
>
> **W3**: We extend Thm. 5.5, see Point 3 in our general response.
> The efficacy of the FGL for directed graphs is substantiated in two key ways. 1) FGLs exhibit the ability to directly capture long-range dependencies by adding “virtual” edges, see also our answer to **W1**, 2) we present ablation studies on the effect of the FGL, see Point 1 in the general response. All ablation studies show that the combination of the FGL and the ODE framework improves the performance on directed graphs.
>
> We hope that this is enough evidence for the benefits of using FGL in directed graphs.
>
> ---
>
> **W4**: The justification for our def. of Dirichlet energy is actually elaborated on lines 103 ff., immediately after its introduction. Additionally, our defs. of the Dirichlet energy (Def. 3.1), SNA, and SNL (Def. 3.2) and their relationship to the Dirichlet energy (Prop 3.6) in directed graphs recovers the standard defs. in the undirected case, see, e.g., [Cai et al., 2020].
>
> ---
>
> **W5**: We would like to respectfully clarify a few points. In addition to Tab. 2, Fig. 3 also presents results for 11 directed graphs.
> Our model demonstrates superior performance on 4 out of 9 real-world datasets, and is on pair with the SOTA on the 11 directed graphs in Fig. 3 (see Tab. 5 in Appendix F.2). Moreover, our approach does not rely heavily on hyperparam. tuning and avoids unnecessary complexity in the GNN layers (jumping knowledge, non-linearities, dropout, or positional encodings), maintaining the comprehensibility and relevance of our theoretical results. This is in contrast to the models we compare with, e.g., GREAD, GraphCON, GRAND.
>
> We believe our contribution to the GNN community lies in our novel approach to understanding directed graphs and fractional Laplacians, an area that has been less explored in the past. Our work can provide a foundation for further exploration and advancements in this area.
>
> Regarding SOTA results, we welcome suggestions on methods to include in our revision.
>
> ---
>
> **W6**: Our justification is straightforward: complex numbers are natural when dealing with directed graphs, as eigenvalues and eigenvectors can be complex. Also one the ODEs we consider, the Schrödinger equation, is inherently complex. Moreover, the complexity of the GNN layers scales linearly to the hidden dimension, i.e., complex numbers are not problematic at all. Thus, the use of complex numbers in our work is rooted in necessity and applicability, rather than the availability of software support. This was already used by related works on directed graphs, e.g., MagNet [Zhang et al., 2021].
>
> ---
>
> **Q1**: Please see Point 2 in the general response for the first two questions. Regarding $\lambda$-FD: as noted in Fig. 2, graphs with medium homophily could benefit from this. One could for example initialze the exponent with a negative value when the homophily level is known in prior. Our theory offers the backbone for future work to analyze these use-cases.
>
> ---
>
> **Q2**: Please see our response to **W2** and Point 2 in the general response. Also, note that we actually *prove* that our approach mitigates OS, see Section 5.
>
> ---
>
> **Q3**: Our approach can be scaled using methods such as randomized or truncated SVD. We noted this in Fig. 8  in Appendix A, where 20% of the singular values are already sufficient for SOTA performance.
> However,  given our present resource constraints we are currently unable to run experiments on such large-scale datasets. We acknowledge this as a significant future direction for our work, though it is beyond the immediate scope of this study.
>
> ---
>
> **Q4**: Cora is relatively simple as both, the nodes’ features and their 1-hop neighbourhood, are expressive. However, as you rightly suggested, employing fewer training nodes (as in [5]) does indeed showcase the utility of our method, even without any hyperparameter tuning:
>
> | model | Cora | Citeseer | Pubmed |
> |:-:|:-:|:-:|:-:|
> | MLP |  58.2±2.1  |  59.1±2.3  |  70.0±2.1  |
> | Ours | 77.8±1.1 | 69.1±1.6 | 74.2±2.2 |
>
> We happily incorporate these findings into the paper's appendix.
>
> ---
>
> **Q5**: Please see Point 4 in the general response.
>
> ---
>
> **Q6**: Please see Point 1 in the general response.
>
> ---
>
> We've thoroughly addressed your concerns. Considering the positive feedback from other reviewers and our comprehensive revisions, we respectfully request a reevaluation of your score. Specifically, if any sections still appear unsound or lacking in presentation, please guide us to them.

---

> > ### Comment · Reviewer_PWaE · 2023-08-14
> >
> > Thank you for the authors' response. Here is my feedback:
> >
> > Regarding Q1: However, Figure 2 showcases a synthetic cycle graph. Are there any realistic graphs where applying the  $\lambda-FD$ would be necessary? **The gap between your theoretical framework and empirical results is evident**.
> >
> > For Q3: I find the response insufficiently persuasive. In my view, **the limitations of your method when applied to large-scale graphs are apparent**.
> >
> > For Q4: The results of your methods are **worse than GCN/GAT** on all Cora/Citeseer/Pubmed Datasets.
> >
> > For W2/Q2: Referring to Figure 1 in the provided PDF, it's essential to compare your results with other state-of-the-art deep GNNs. Moreover, as highlighted in [1], the squirrel and chameleon datasets contain a significant number of duplicate nodes. Evaluating your model's efficacy primarily based on these two datasets significantly does not provide a comprehensive assessment. You need to refer to the experiment settings in other deep GNN papers.
> >
> > [1]. Platonov, Oleg, et al. "A critical look at the evaluation of GNNs under heterophily: are we really making progress?." arXiv preprint arXiv:2302.11640 (2023).

---

> > > ### Author Response · Authors · 2023-08-16
> > >
> > > We appreciate the feedback from the Reviewer. To address the raised points, we would like to emphasize two critical aspects:
> > >
> > > 1. We did not expect to achieve state-of-the-art performance on Cora or other homophilic graphs where node features with low Dirichlet Energy show promising results.
> > > 2. The essence of our work lies in introducing the fractional Laplacian, its theoretical advantages over standard Laplacians, and its demonstrable improvement over the standard Laplacian and flexibility across various graph datasets. Therefore, our emphasis is not solely on our performance metrics on Cora but instead on the broader contributions we present.
> > >
> > > ---
> > > Regarding Q1: We kindly but strongly disagree with the Reviewer's statement: "The gap between your theoretical framework and empirical results is evident." Our theoretical analysis is verified by the experiments, and vice versa, the experiments are fully backed up by our theoretical analysis, as shown in our general response (Point 2) and the supplementary pdf (Tab. 2, Fig. 1). We also back this up with Figs. 6 and 7 in our appendix and our ablation studies (see Point 1 in our general response).
> > >
> > > We have conducted a complete analysis w.r.t. the exponent since, in our model, $\alpha$ is an unconstrained learnable parameter. Hence, it could also be negative.
> > >
> > > The synthetic cycle graph serves as one illustrative example. Another example is Squirrel (e.g., Fig. 1 in the supplementary pdf), where the learned exponent is negative at depth 2. **The fact that the model only sometimes learns negative exponents does not change our alignment of theory and practice.**
> > >
> > > ---
> > > Regarding Q3: The limitations of our method are indeed clear, as stated in the "Limitation" section of the main paper.
> > > However, our model already comfortably accommodates medium size graphs of 20,000 nodes. Handling large-scale graphs is currently beyond the scope of our work. We would like to highlight that **before our work, the application of fractional Laplacians was restricted to tiny graphs (a few hundred nodes) due to the computation of Jordan decompositions.**
> > >
> > > ---
> > > Regarding Q4: Please note that in the main paper, our model performs better than GCN/GAT on both Citeseer and Pubmed. As of Cora, we perform slightly better than GAT but slightly worse than GCN.
> > >
> > > As previously clarified, we did not hyperparameter tune for that particular experiment. We rather followed the request of the Reviewer and demonstrated that in a low-labeling regime, our model substantially outperforms MLPs. The experiment was not meant to chase SOTA in this new split scenario, as it would have required more time for hyperparameter tuning. It is crucial to understand that **we do not assert superiority over smoothing models like GAT or GCN on Cora, where node features with low Dirichlet Energy show promising results.** At the same time, our model breaks down to GCN (with skip connection and shared weights) if $\alpha=1$. We thus expect that, given the correct hyperparameters, FLODE does not perform significantly worse.
> > >
> > > ---
> > > For W2/Q2: Fig. 1 is an **ablation study** on the effect of different update rules on the test accuracy. More specifically, we wanted to show how the choice of learnable exponent and skip connections influence the performance. Therefore, the comparison with SOTA GNNs is not pertinent. Please note that we compare against SOTA GNNs in Tab. 1 and 2 in the main paper, and in the general response.
> > >
> > > Chameleon, Squirrel, and Actor are widely accepted as standard heterophilic datasets in the GNN field, often used in studies at top-tier conferences like [1] and [2]. Even if there are duplicate nodes, these datasets are still challenging due to their heterophilic nature. It is worth noting that none of the papers the Reviewer suggested for comparison consider the alternative datasets that the Reviewer proposes in their most recent answer.
> > >
> > > ---
> > > [1] Di Giovanni, F., Rowbottom, J., Chamberlain, B. P., Markovich, T., and Bronstein, M. M. (2022). Graph Neural Networks as Gradient Flows: Understanding Graph Convolutions via Energy. arXiv: 2206.10991 [cs, stat].
> > >
> > > [2] Choi, Jeongwhan, et al. "GREAD: Graph Neural Reaction-Diffusion Networks." (2023).

---

> > > > ### Comment · Reviewer_PWaE · 2023-08-17
> > > > **Part 1/2**
> > > >
> > > > Thank you for the response.
> > > >
> > > > Regarding Q4 in your response, it is noted that the accuracy of your model is lower than that of GCN/GAT across all three small datasets in your response table. While you mentioned that "The experiment was not meant to chase SOTA in this new split scenario," it should be clarified that this is not a new split scenario. In fact, it is a common method for evaluating GNN models, as evidenced by its use in models such as GCN, GAT, and SGC.
> > > >
> > > > For W2/Q2: The paper [4] I cited was published at top-tier conferences and has been cited by other papers such as [5] from top-tier conferences. My intention was to suggest that adding more datasets  to support your claims, rather than relying solely on the two datasets currently included in the study.
> > > >
> > > > For  "The gap between your theoretical framework and empirical results is evident." :
> > > >
> > > >
> > > >
> > > >
> > > > # ``I suspect that this paper is solely playing a definition game about oversmoothness.``
> > > >
> > > >
> > > > Besides the above strong practical concerns about whether it fits the large datasets, etc., upon careful scrutiny of the manuscript, I am compelled to express significant concerns regarding its core contentions and proposed methodologies. I am deeply skeptical about the veracity of the paper's primary claims and urge the authors to provide a thorough clarification.
> > > >
> > > >
> > > > # Let's first be clear about the definition of oversmooth in GNN: oversmoothing means that after sufficient time or layers, the initial node features states have been forgotten.
> > > >
> > > >
> > > > ## Clarify about oversmoothness in GCN and oversmoothness in other GNNs
> > > >
> > > >
> > > >
> > > > I acknowledge that exponential decay of Dirichlet Energy towards 0 signifies oversmoothness in **``GCN``** as highlighted in [3].
> > > >
> > > >
> > > >
> > > > However, a pertinent question that arises is whether this universally holds true for all GNNs, ``especially across the spectrum of graph ODE models``. ``Is it accurate to assert that the only metric for oversmoothness is the Dirichlet Energy and its``**``exponential decay to 0``** in Definition 5.1?

---

> > > > > ### Comment · Reviewer_PWaE · 2023-08-17
> > > > > **part 2/2**
> > > > >
> > > > > ### I highly suspect that the authors are playing a definition game here:
> > > > >
> > > > >
> > > > >
> > > > > In Definition 5.1., they define the oversmoothness as the Dirichlet Energy decays exponentially fast to $\lambda_{min}$.
> > > > >
> > > > >
> > > > >
> > > > > **This definition itself is ridiculous if we think carefully here. If Dirichlet Energy decays exponentially fast to $\lambda_k$ that is different to $\lambda_{min}$, can we claim there is no oversmoothness or the oversmoothness has been mitigated?**
> > > > >
> > > > >
> > > > >
> > > > >
> > > > >
> > > > > A counter-example to ponder upon: If all node features converge to the eigenspace associated with $\lambda_k$ (the $k$-th eigenvalue of the graph Laplacian), can we claim that this is no oversmoothness or the oversmoothness has been mitigated? **``But please note that the features are still converging to something that only depends on the graph topology!``**
> > > > >
> > > > >
> > > > >
> > > > >
> > > > >
> > > > > Definition 5.1's claim, wherein oversmoothness is quantified as the Dirichlet Energy's exponential decay to $\lambda\_{\text{min}}$, appears problematic. **``If the Dirichlet Energy exhibits an exponential decay to ``** $\lambda_k$,**`` distinct from``** $\lambda\_{min}$, **``it is imprudent to assert that oversmoothness is nullified or mitigated.``**
> > > > >
> > > > >
> > > > >
> > > > >
> > > > >
> > > > > I have serious doubts about the definitions in Section 5.1. How can the term "Frequency-Dominant" be an indicator of oversmooth mitigation?
> > > > >
> > > > >
> > > > >
> > > > > The datasets, either homophilic or heterophilic, should depend more on the features of the graph input states besides the graph topology. But your definition only touches on the graph topology finally. **``This is already a potential indicator of oversmoothness.``**
> > > > >
> > > > >
> > > > >
> > > > > Theorem 5.4 in this paper is another perplexing point. It suggests that given some conditions on the graph structure, the Dirichlet energy will converge to a value linked to certain eigenvalues of the graph topology. This doesn't add up. The node features can still converge to states independent of their original features. **``But if the final states depend only on graph topology, it is still oversmoothing!``**
> > > > >
> > > > >
> > > > >
> > > > > Moreover, the ODE model introduced here doesn't convincingly back up the authors' claims. The skip connections in ODE solvers might indeed help maintain node classification accuracy across many layers [1][2]. But there's no comparison with other oversmoothness ODE models. This glaring omission makes it extremely hard to judge the paper's true contribution.
> > > > >
> > > > >
> > > > >
> > > > > One glaring weakness: after reading the rebuttal and the further explanations by the authors, I'm left thinking the authors might be biting off more than they can chew. They dabble in directed graphs, but this is **only weakly addressed in relation to oversmoothing in Section 5.2.**
> > > > >
> > > > > The entire paper feels like it's based on a **toy model with just small datasets.** The reliance on singular value decomposition restricts the model from tackling large graph datasets. More seriously, I highly suspect that the authors are just playing a math definition game here instead of truly trying to mitigate oversmoothness. More experiments need to be done to demonstrate the over-smoothness mitigation as I suggested early.
> > > > >
> > > > > I do not have any idea about how this paper offers any real value to the GNN community. Between the shaky theoretical discussion on oversmoothness, the limited scope of experiments on just small datasets, and the lackluster dive into directed graphs, the paper is in a very weak form, and it is fundamentally not ready for publication on NeurIPS.
> > > > >
> > > > > [1] Li, G., Xiong, C., Thabet, A. and Ghanem, B., 2020. Deepergcn: All you need to train deeper gcns.
> > > > >
> > > > > [2] Li, G., Muller, M., Thabet, A. and Ghanem, B., 2019. Deepgcns: Can gcns go as deep as cnns?.
> > > > >
> > > > > [3] Cai, Chen, and Yusu Wang. "A note on over-smoothing for graph neural networks."
> > > > >
> > > > > [4]. Platonov O, Kuznedelev D, Diskin M, et al. A critical look at the evaluation of GNNs under heterophily: are we really making progress?
> > > > >
> > > > > [5]. Zhao K, Kang Q, Song Y, et al. Graph neural convection-diffusion with heterophily[J].

---

> > > > > > ### Author Response · Authors · 2023-08-19
> > > > > > **Oversmoothing and Missunderstandings of Reviewer Clarified**
> > > > > >
> > > > > > The phenomenon of oversmoothing, as its name suggests, describes a scenario where node features become overly smooth, too quickly. When this occurs, nodes features of nodes connected by an edge converge to the same feature if we let the depth of the underlying GNN go to infinity. This phenomenon is not unique to GNNs; in other pre-machine learning methods, like Laplacian smoothing [1], the same phenomenon occurs. The concept of oversmoothing was then observed in GNN architectures like GCN, GAT and GRAND. This is a serious limitation, as oversmoothing hinders GNNs to perform well on graphs where long-range dependencies exist, e.g., heterophilic graphs. Oversmoothing's formal definition has evolved over time. One well-accepted definition relates to the convergence of node features towards the null space of the graph's SNL [2]. Other definitions link to the exponential decay of Dirichlet Energy towards zero [3,4] or the convergence of random walks to its stationary distribution [5]. These definitions, while varying in phrasing, capture the same essence: node features converge rapidly to similar values for connected nodes.
> > > > > >
> > > > > > In our study, we analyze oversmoothing within the mathematical framework of the Dirichlet Energy. In undirected graphs, the smoothest possible node features of a connected graph correspond to zero Dirichlet Energy. A fact supported by spectral graph theory results that confirm the SNL will always have a zero eigenvalue when the graph is connected. Directed graphs do not necessarily follow this pattern, leading us to select the smoothest possible node features: the energy corresponding to the minimum eigenvalue $\lambda_{\mathrm{min}}$. Our claim that oversmoothing is mitigated if the output node features do not converge to $\lambda_{\mathrm{min}}$ is firmly grounded in both theory and intuition: when increasing the depth the underlying GNN will not output overly smooth node features.
> > > > > >
> > > > > > Our work goes a step further by demonstrating that the limit of our model is not strictly $\lambda_{\mathrm{min}}$  both theoretically and empirically, see, e.g. Sec. 5 and Point 1 in the general response. This fact, by definition, shows mitigation of oversmoothing. It’s also worth highlighting that eigenspaces of other eigenvalues than $\lambda_{\mathrm{min}}$ might convey non-trivial information, both for undirected and directed graphs. Yet, this implies our model has the capacity to retain high Dirichlet Energy, making it apt for heterophilic graphs, be they directed or not. We want to highlight once again, while we chose the mathematical framework of the Dirichlet energy, it is equivalent to the other (less used) definitions in [2] or [5]. For a survey, we refer to [7], which defines oversmoothing in their first sentence: „Node features of graph neural networks (GNNs) tend to become more similar with the increase of the network depth“, and links it to the Dirichlet energy as we do.
> > > > > >
> > > > > > **To conclude, the standard definition of oversmoothing and its underlying intuition differs from the reviewer's perspective.** We hope this clarification assures that we are not „playing a definition game.“
> > > > > >
> > > > > > [1] Zhou, D., & Schölkopf, B. (2005, August). Regularization on discrete spaces. In Joint Pattern Recognition Symposium (pp. 361-368).
> > > > > >
> > > > > > [2] Oono, K., & Suzuki, T. (2019). Graph neural networks exponentially lose expressive power for node classification. arXiv preprint arXiv:1905.10947.
> > > > > >
> > > > > > [3] Cai, C., & Wang, Y. (2020). A note on over-smoothing for graph neural networks. arXiv preprint arXiv:2006.13318.
> > > > > >
> > > > > > [4] Rusch, T. K., Chamberlain, B., Rowbottom, J., Mishra, S., & Bronstein, M. (2022, June). Graph-coupled oscillator networks. In International Conference on Machine Learning
> > > > > >
> > > > > > [5] Thorpe, M., Nguyen, T. M., Xia, H., Strohmer, T., Bertozzi, A., Osher, S., & Wang, B. (2021, October). GRAND++: Graph neural diffusion with a source term. In International Conference on Learning Representations.
> > > > > >
> > > > > > [6] Di Giovanni, F., Rowbottom, J., Chamberlain, B. P., Markovich, T., & Bronstein, M. M. (2022). Graph neural networks as gradient flows. arXiv preprint arXiv:2206.10991.
> > > > > >
> > > > > > [7] Rusch, T. K., Bronstein, M. M., & Mishra, S. (2023). A survey on oversmoothing in graph neural networks. arXiv preprint arXiv:2303.10993.

---

> > > > > > > ### Author Response · Authors · 2023-08-19
> > > > > > > **Addressing Reviewer PWaE’s Concerns**
> > > > > > >
> > > > > > > We address the concerns.
> > > > > > >
> > > > > > > > „I highly suspect that the authors are playing a definition game here [...]“
> > > > > > >
> > > > > > > Our clarification on the definition and intuition behind oversmoothing should respond to this concern. **Our definition is grounded in established literature** (refer to the previous comment) and aligns with both theoretical and intuitive understandings of oversmoothing. Notably, convergence to certain graph topologies doesn't inherently signify oversmoothing. Oversmoothing specifically pertains to the exponential decay of the Dirichlet Energy to zero — a state where node features become indistinguishable for connected nodes. The primary idea behind advocating model flexibility in Frequency-Dominance is straightforward and intuitive: homophilic graphs benefit from low energy, heterophilic graphs benefit from high energy, and graphs between homophilic and heterophilic extremes may benefit from mid-energy levels.
> > > > > > >
> > > > > > >  >   „Is it accurate to assert that the only metric for oversmoothness is the Dirichlet Energy and its exponential decay to 0 in Definition 5.1?“
> > > > > > >
> > > > > > > Regarding this query, we again refer to our previous comment, showcasing that oversmoothing is not about a "definition game", but is mathematically defined. **We consequently prove mathematically that we mitigate oversmoothing.** Moreover, we also use accuracy as an indicator to demonstrate that oversmoothing is mitigated. As shown in Fig. 1 of our supplementary pdf, our method's performance remains stable for up to 256 layers.
> > > > > > >
> > > > > > > >    „But if the final states depend only on graph topology, it is still oversmoothing!“
> > > > > > >
> > > > > > > Please see our previous comments. Let's summarize: **if we effectively mitigate the energy's convergence to zero, we are provably mitigating oversmoothing**. Moreover, results on oversmoothing, including our findings, often focus on limit behaviors. Thus, while the limit may be solely dependent on the graph topology, **the output from a finite GNN relies on both the initial node features and the graph topology**. Plus, the eigenspace corresponding to other eigenvalues than $0$ may carry non-trivial information. For example, the second eigenvector is a relaxation of the Cheeger vector, and the largest eigenvector gives a highly non-smooth vector, suitable for classification in heterophilic graphs (see Sec. 5.1).
> > > > > > >
> > > > > > >  >   „ But there's no comparison with other oversmoothness ODE models.“
> > > > > > >
> > > > > > > We have compared our method with various models in our experiments, as evident in Tab. 1 and the supplementary pdf. Our method stands out in terms of flexibility, especially concerning directed graphs.
> > > > > > > For instance, our work is the only work that shows a) that oversmoothing is mitigated for fractional Laplacian-based ODEs in undirected graphs, b) possible mid-frequencies dominance, c) that our method mitigates oversmoothing on directed graphs with normal SNA, and d) if $\alpha=1$ for all directed graphs.
> > > > > > > We're open to adding short survey about other ODE-based models that *provably* mitigate oversmoothing in a revised version of the paper, including GRAFF, GraphCON, and GRAND++.
> > > > > > >
> > > > > > > >    „How can the term "Frequency-Dominant" be an indicator of oversmooth mitigation?“
> > > > > > >
> > > > > > > We hope that our previous responses answer this query. The notion of "Frequency-Dominant" as oversmoothing mitigation indicator stems from [6], and our incorporation of it is consistent with its original intent.
> > > > > > >
> > > > > > > We would like to gently remind the Reviewer to consider the broader context rather than getting overly fixated on specific terminology. **The essence lies not in the label "Frequency-Dominant" per se, but in the associated literature, the theoretical insights, and the practical implications we bring forth.**
> > > > > > >
> > > > > > > >    „I'm left thinking the authors might be biting off more than they can chew.“
> > > > > > >
> > > > > > > Please refer to Sec 4, 5, and 6 and our general response, which discusses directed graphs.
> > > > > > >
> > > > > > >  >   „The entire paper feels like it's based on a toy model with just small datasets.“
> > > > > > >
> > > > > > >  We have tested our model on both real-world and synthetic directed graphs, encompassing up to 20K nodes. These datasets are not small or toy datasets; they are standard in the GNN literature.
> > > > > > >
> > > > > > > >  "More experiments need to be done to demonstrate the over-smoothness mitigation as I suggested early.“
> > > > > > >
> > > > > > > **See the general response or Fig. 7 in the appendix for many such experiments. More importantly, we *provably* mitigate oversmoothing**.
> > > > > > >
> > > > > > > >Regarding the inclusion of other splits.
> > > > > > >
> > > > > > > We acknowledge that the suggested split is common. Still, we employed standard splits for the considered graphs. Hyperparameter tuning for different splits is beyond our work's scope. Note that we already consider 10 different splits per dataset, and it's uncommon to analyze various split methods, see .e.g, [5-7].
> > > > > > >
> > > > > > > >Regarding the inclusion of new datasets.
> > > > > > >
> > > > > > > Although we have evaluated over 20 datasets, we value this feedback. Given the time constraints for the rebuttal, we're prepared to include more datasets in a potential final version of the paper.

---

> > > > > > > > ### Author Response · Authors · 2023-08-19
> > > > > > > > **Summary**
> > > > > > > >
> > > > > > > > We thank the reviewer for their comments. We hope that our answers clarify the definition of oversmoothing and our contributions to mitigating it. Furthermore, we will include a short survey (see first reply "Oversmoothing Clarified") on oversmoothing and related works in a revised paper to make our work more accessible. We are also happy to include more datasets for a revised paper as only one working day is left for the discussion phase, and are open to further questions.

---

### Official Review · Reviewer_R3EG · 2023-07-06

**Soundness:** 2 fair
**Presentation:** 3 good
**Contribution:** 2 fair
**Rating:** 5
**Confidence:** 4

**Summary:**

The paper extends the notion of Dirichlet energy to define oversmoothing in directed graphs. It introduces a fractional Laplacian operator that is used to create graph neural ODE architectures. The resulting designs are non-local and can alleviate oversmoothing. The paper provides empirical results (accuracy) on nine real-world benchmarks for node classification.


**Strengths:**

* Overall, the paper is well-written and easy to follow. Also, the proposed framework (FLODE) is simple;
* The proposed method works relatively well for both directed and undirected graphs (flexibility).


**Weaknesses:**

- Novelty: I rank the novelty as moderate. First, directed symmetrically normalized laplacian has been previously used. Moreover, it seems that the advantages come mainly from the choice of diffusion operator. However, it is clear that the choice of diffusion operator impacts oversmoothing, and we can design, for instance, high-pass (or non-local) graph filters to alleviate oversmoothing. Last, there is no novelty in the evaluation setup.
- Theory: Some theoretical results focus on the properties of the fractional Laplacian, which appears only complementary to the paper. As a result, the paper does not discuss empirical results in light of the theoretical ones.
- Experiments: The empirical results are not strong. Except for Squirrel, the performance gains (over the second-best baseline) do not seem to be statistically significant. Also, the proposed model is the best-performing model only in 4 out of 9 datasets. In addition, the paper does not consider large-scale datasets from OGB, for instance.

**Questions:**

**Questions and comments**

1. Why is it called Fractional graph laplacian if it is based on the decomposition of the normalized adjacency (SNA)?
2. Do the authors have any intuition to explain why the proposed method works so well on the Squirrel dataset?
3. Some parts could be improved for precision and clarity, e.g.,
    - In the paragraph 'Homophily and Heterophily', the paper uses $i \in V$ to denote the distribution used in the expectation. What does it mean (e.g., uniform dist. over the nodes)?
    - In the introduction, the paper says: "a GNN that minimizes the Dirichlet energy is expected to perform well on homophilic graphs...". However, there is no dependency on the GNN --- x is defined as initial node features.
5. Running the ablation study on Appendix F3 to more datasets would be interesting and helpful to validate some intuitions behind the proposal.
6. There is a vast literature on methods for tackling oversmoothing in GNNs (DropEdge, etc.), enabling such approaches for heterophilic datasets. Shouldn't these approaches also be considered as baselines for comparison?
7. Have you tried employing an NN instead of using just a channel mixing matrix?

**Limitations:**

The authors discuss the limitations of the proposal, mainly mentioning the computation cost associated with SVD.

---

> ### Author Rebuttal · Authors · 2023-08-07
>
> We are very grateful to the reviewer for the time taken to carefully assess our work and for the valuable feedback. We address each point individually. “W/Q” numbers the weakness or question, followed by our response.
>
> ---
> **W1**: Please note that our work is the first in-depth theoretical analysis of the directed SNA, tying it to Dirichlet energy. We introduce a new def. for the fractional graph Laplacian (FGL) specific to directed graphs and present novel theoretical results.
>
> Regarding your comment on the choice of diffusion operator: we agree that the operator selection can have an impact on OS. However, it's not a foregone conclusion that just any diffusion operator can alleviate this issue. Through our work, we've demonstrated, both theoretically and empirically, that the FGL *combined* with the ODE framework is capable of mitigating OS. However, this does not necessarily hold for the FGL without the ODE framework. To further clarify this point, we have included new additional ablation studies (see Point 1 in our general response, Figure 1 and Table 1 in the supplementary pdf), complementing the one we conducted on the Chameleon dataset (refer to Appendix F.3).
>
> ---
> **W2**: Our theoretical insights are not mere supplements but rather fundamental components of our work. They underline how the FGL, used in conjunction with graph ODEs, is adept at capturing long-range dependencies in both directed and undirected graphs. This conclusion stems from Thms. 4.1 and 5.3-5.5; the theoretical results from Section 3 are fundamental for the proofs of the above-mentioned thms.
>
> To further emphasize this point, we have made efforts to closely associate our empirical results with our theoretical ones. See Points 1 and 2 in our general response.
>
> ---
> **W3**: We value your feedback but have a different perspective on our empirical results. Our method excels on 4 out of 9 datasets and performs well especially on heterophilic graphs, ranking best on 3/6 and top three on 5/6. Additionally, our performance on the directed datasets (Fig. 3) is competitive with the SOTA.
>
> What distinguishes our approach is its simplicity. Unlike many models, ours doesn't rely on layer normalization, positional encodings, jumping knowledge, rewiring, dropout, non-linearity in the GNN, or extensive hyperparameter searches. Our ablation study underscores how our model's performance stems from our simple design choices.
>
> On large-scale datasets, while our method holds potential for scalability through truncated SVD, our hardware constrains us from effectively handling datasets with 100K+ nodes. Nevertheless, our current results and theory pave the way for future advancements, especially around scaling.
>
> ---
> **Q1**:  The term "Fractional Graph Laplacian" refers to our approach to applying the fractional power of a Laplacian. The term "Laplacian" is often used in graph theory to denote a matrix representation of a graph. Our approach of taking fractional powers in the singular value domain generalizes to any graph laplacian, not only the SNA or the SNL, hence the name. We hope this clarifies our choice, and we remain open to alternative suggestions!
>
> ---
> **Q2**:  The Squirrel dataset is heterophilic, where nodes of different classes connect. Standard GNNs often underperform on such datasets due to OS. Our method mitigates this, explaining our enhanced performance over baselines on Squirrel. Yet, OS isn't the sole challenge in graph learning; "oversquashing" is another, preventing long-range node information transfer. [Song et al., 2023] suggest Squirrel experiences oversquashing. Our model's success might also stem from the virtual edges introduced by the FGL, possibly addressing oversquashing. However, this remains speculative without formal proof.
>
> Additionally, we present ablation studies on “Squirrel”, see Point 1 in the general response.
>
> [Song et al., 2023] Song, Yunchong, et al. "Ordered GNN: Ordering Message Passing to Deal with Heterophily and Over-smoothing." (2023).
>
> ---
> **Q3**:  Thank you for your feedback. To address your concerns:
> a) Yes, we mean a uniform distribution over nodes. We've now opted for a simple sum over the nodes to avoid confusion.
>
> b) We clarify the GNN's dependency by stating: "A GNN whose output node features minimize the Dirichlet energy is expected to perform well on homophilic graphs."
>
> If there are other areas needing clarity, please let us know. Your assistance is vital in ensuring that our work is both comprehensive and accessible to all readers.
>
> ---
> **Q4**:  We appreciate your suggestion. In response, we've included a more comprehensive ablation study for the Squirrel dataset, as detailed in the supplementary pdf (see Tab. 1). This extended analysis should provide additional validation for our method.
>
> Additionally, we added ablation studies that show that our method can scale to large depths and validate our theoretical results (see Points 1-2 in the general response and Fig. 1 in the supplementary pdf).
>
> ---
> **Q5**:  See Point 3 in our general response.
>
> ---
> **Q6**:  We have explored other learnable parameters beyond a diagonal channel mixing matrix, including full matrices and time-dependent matrices. These did not have the same theoretical results or empirical performances as our method, which is why we discarded them. However, we ran the experiments, see also our response to Reviewer dNLj.
>
> While it would be possible to apply a neural network, this approach can not be analyzed in our current framework. We believe this could be a direction for future work and we'd be glad to mention this possibility in our conclusion.
>
> ---
> We have addressed each of your comments and feel our paper stands on solid theoretical and empirical grounds. We would appreciate if you could specify the parts you consider “unsound”, given your score of 2, so we can further refine our work. If our responses have solved your concerns and you believe our paper has improved, we kindly ask you to improve the score.

---

> > ### Comment · Reviewer_R3EG · 2023-08-17
> >
> > Thanks for taking the time to answer my questions. I have no further questions/comments. I have read the other reviews and authors' responses and would like to keep my initial score.

---

### Official Review · Reviewer_dNLj · 2023-07-06

**Soundness:** 3 good
**Presentation:** 4 excellent
**Contribution:** 2 fair
**Rating:** 6
**Confidence:** 2

**Summary:**

The authors consider the problem of classification on attributed graphs with a geometrical approach. They introduce a fractional graph Laplacian for undirected and directed graphs. They generalize the notion of Dirichlet energy to directed graphs. They study the ODEs based on this Laplacian and prove that their solutions converge to high-, low- or middle-frequency patterns, for undirected graphs and for directed graphs in a particular case. Finally they implement these ODEs as GNNs and assess their performances.

**Strengths:**

I do not know well this topic and can hardly evaluate how significant are the improvements this article brings.

**Weaknesses:**

To my understanding a main point of this article is fractional Laplacian for directed graphs, but theorem 5.5 is restricted to the SNL.

**Questions:**

On formatting: in the current state of the manuscript many intra-links are broken. Some figures of interest for understanding are not in the main text (I did not look at them). Maybe for table 1 the authors could use font formatting (bold, italic, ...) instead of colors.

L. 245 « In accordance with the results in Section 5, we select W as a diagonal matrix. » could the authors develop?

Part 6, for the directed SBM, what are the features?

Out of curiosity: what happens if W is untied, i.e. taken time-dependent or different at each layer? Does one have the learned alpha small (or negative) for heterophilic datasets?

**Limitations:**

The authors address a main limitation of their model, its computational complexity. They may conduct an experiment showing that truncating the SVD is correct.

---

> ### Author Rebuttal · Authors · 2023-08-07
>
> We are very grateful to the reviewer for the time taken to carefully assess our work and for the valuable feedback. We address each point individually. “W/Q” numbers the weakness or question, followed by our response.
>
> ---
> **W1**: We agree that Thm. 5.5 seemed to be restricted, and we have since made improvements to address this concern. We have now generalized Thm. 5.5 to encompass not only $\alpha=1$, but any $\alpha$ as long as the underlying graph has a normal SNA. This was feasible without altering our proofs since any normal SNA is unitary diagonalizable. See also Point 3 in our general response.
> Moreover, we would like to clarify some points:
>
> 1. Our results not only hold for directed graphs but also for undirected. Moreover, the improvement in performance can also be seen in undirected graphs, see ablation studies or Point 2 in general response.
>
> 2. The ability to capture long-range dependencies via the FGL isn't tied solely to Thm. 5.5. The FGL does also add “virtual” edges between long-distant nodes, as described in Section 4. Thus, our approach exhibits its effectiveness on tasks with (directed) heterophilic graphs. We further substantiate this claim with an additional ablation study reported in the supplementary pdf.
>
> ---
> **Q1**: We apologize for the inconvenience caused by the broken intra-links. They currently refer to the appendix, and this issue will be resolved in the camera-ready version, as the main manuscript and appendix can be combined into one document.
>
> Regarding the Figs. you've mentioned, we would greatly appreciate if you could specify which ones you are referring to, so we could potentially move them to the main text, considering the availability of an extra page for the camera-ready version. Based on the suggestions of the other Reviewers, we move our ablation studies to the main paper, however, we are open for discussion if you believe other Figs. are even more important for understanding our points.  Lastly, we thank you for your suggestion about the color usage in Tab. 1. We will underline the first place, bold for the second, and italic for the third, to ensure better readability.
>
> ---
> **Q2**: In Thm. 5.3 and Thm. 5.5, our theoretical findings are valid for symmetric and diagonal channel mixing matrices respectively. To align with these theoretical insights and ensure their applicability to our model, we selected $\mathbf{W}$ to be a diagonal matrix in all our experiments. It's worth noting that all diagonal matrices inherently possess symmetric properties, which makes them suitable for satisfying the conditions of both Thms. We will clarify this in our revised paper.
>
> ---
> **Q3**: For the directed SBM, we followed the original setup used in the MagNet paper, where the features are one-dimensional and are sampled from the standard Gaussian distribution.
>
> Making $\mathbf{W}$ time-dependent or distinct at each layer is certainly intriguing. We conducted a set of experiments with this modification to our model and found that it yielded comparable performance.  Moreover, the exponent does display similar tendencies. Specifically, for heterophilic graphs (Chameleon and Squirrel), the learned exponent is small. Meanwhile, for homophilic graphs, the learned exponent remains close to $1$. Notably, Film presents an anomaly — there's a drop in accuracy, and its learned exponent doesn't diminish, despite it being a heterophilic graph.
>
>
> | | | Film | Squirrel | Chameleon |Citeseer | Pubmed | Cora |
> |:-:|:-:|:-:|:-:|:-:|:-:|:-:|:-:|
> |Undirected| Accuracy |34.72±1.21|63.49±1.66|73.42±2.0|78.19±2.55|88.96±0.35|86.66±1.09|
> | | $\alpha$ |0.92±0.01|0.30±0.04|0.38±0.06|0.97±0.007|0.98±0.05|0.95±0.02|
> |Directed|Accuracy|35.27±1.33|73.29±1.62|78.79±1.86|-|-|-|
> | | $\alpha$ |0.94±0.004|0.25±0.08|0.25±0.08|-|-|-|
>
> ---
> However, our theoretical analysis regarding OS would not directly extend to this adjusted model. Analyzing solutions for time-dependent ODEs is more challenging. Since one of our goals in this work was to maintain a tight relationship between theory and practice, we chose to share $\mathbf{W}$ among all layers. Nevertheless, we are happy to include our experimental results using an untied $\mathbf{W}$ in the appendix and identify the investigation of such time-dependent neural ODEs in both theory and practice as a promising avenue for future work. Finally, to facilitate further research in this direction, we will update our code to include an ``--no_sharing`` option. This will allow other researchers to easily explore these questions using our open-source code.
>
> ---
> **Limitations**: By employing a truncated SVD and retaining $k$ singular values, we achieve the optimal $k$-rank approximation of the fractional Laplacian, as supported by the Eckart–Young–Mirsky theorem. As shown in Appendix A, Figure 8, even when using a truncated SVD that captures only 24% of the singular values, we still attain the same SOTA performance on Chameleon as with the full SVD. In the revised version of our paper, we will clarify this point and provide a reference for the corresponding theoretical result.
>
> ---
> We appreciate your thorough feedback and have incorporated all suggested changes. Given our revisions and the excellent rating you provided for the presentation, we kindly ask you to reconsider the scores for soundness, contribution, and overall score. We emphasize our novel approach in introducing a FGL, generalizing Dirichlet energy to directed graphs, ensuring that our models can mitigate OS and that our strong experimental results are fully backed up by the theoretical ones. We hope that this convinces you of our strengths.

---

> > ### Comment · Reviewer_dNLj · 2023-08-18
> >
> > I thank the authors for their detailed answers and additional experiments.
> >
> > > Regarding the Figs. you've mentioned, we would greatly appreciate if you could specify which ones you are referring to...
> >
> > I had in mind fig. 4 which is mentionned twice in the main text.

---

### Official Review · Reviewer_ZW5p · 2023-07-07

**Soundness:** 4 excellent
**Presentation:** 4 excellent
**Contribution:** 3 good
**Rating:** 7
**Confidence:** 3

**Summary:**

The authors propose a neural ODE (FLODE) that uses a fractional graph Laplacian as an alternative to GNNs, which famously suffers from oversmoothing after a few layers. The heat equation $x'(t) = -L^{\alpha}x(t)W$ is shown to possess nice qualities wrt Dirichlet energy such that it does not always end up with a low frequency solution (i.e. oversmoothing as $t \to \infty$). Several experiments are done with synthetic, real, directed, and undirected graphs, and the proposed method are shown to perform comparably with existing method.

**Strengths:**

Even though I am not an expert in neural-ODE based GNNs, I found the paper clearly written and easy to follow. The strength of this paper is in its theoretical analysis of the spectral values of their solutions. Based on a cursory reading of Di Giovanni et al (2022), and given that the learned $\alpha$s are often not 1 (Appendix Table 4), this does not seem to be a trivial extension of existing work.

**Weaknesses:**

Please see questions below.

**Questions:**

1. If Film is a heterophilic graph, I expected the learned exponent to be closer to 0, which is the case with Squirrel and Chameleon. Do the authors have a hypothesis about why this might be the case?

2. Similarly, do the authors have a hypothesis about why FLODE was not in the top three performing models for Film, Pubmed and Cora?

3. Have the authors actually inspected the eigenvalues and singular values of their models and compared them to the theoretical results?

4. I recommend using boldface and italics in addition to color in Table 1. In Figure 3, the line for FLODE should be thicker or have a different line style.

**Limitations:**

Yes.

---

> ### Author Rebuttal · Authors · 2023-08-07
>
> We are very grateful to the reviewer for the time taken to carefully assess our work and for the valuable feedback. We address each point individually. “W/Q” numbers the weakness or question, followed by our response.
>
> ---
> > **Q1**: “If Film is a heterophilic graph, I expected the learned exponent to be closer to 0, which is the case with Squirrel and Chameleon. Do the authors have a hypothesis about why this might be the case?”
>
> Film is a notably challenging dataset, where GNN-based models often struggle to significantly outperform models that do not incorporate the graph structure, such as MLPs. We tried to initialize the exponent $\alpha$ to smaller values, however, the learning algorithm did not adapt $\alpha$, and also the performance did not improve. There could be several reasons for this.
> 1. Film is by far the sparsest graph among the datasets we used, which could potentially impede the propagation of information.
> 2. The features in this dataset may be less expressive (with 932 dimensions compared to 2325 and 2089 for Chameleon and Squirrel, respectively).
> 3. It is a class-imbalanced graph, which makes predictions more challenging.
> 4. The Film dataset represents a subgraph of a larger real-world Wikipedia graph, which may result in a loss of critical information.
>
> Nevertheless, these are mere hypotheses and the definitive cause remains unclear.
>
> ---
> > **Q2**: “Similarly, do the authors have a hypothesis about why FLODE was not in the top three performing models for Film, Pubmed and Cora?”
>
> Please note that FLODE is among the best models on the directed version of the Film dataset, see Tab. 2. For Pubmed and Cora, we suspect that our model's performance may have been affected by our limited computational resources and a consequently restricted hyperparameter search. Importantly, our model essentially reduces to GRAFF when we set $\alpha=1$. Therefore, given the same hyperparameters, we should always be able to match or outperform GRAFF. We are optimistic that with more extensive tuning, our model would improve its standings on these datasets.
>
> Finally, we want to mention that Pubmed and Cora are highly homophilic datasets, where standard GNNs already perform very well as the 1-hop neighborhood is already very expressive. Hence, we do not expect improvements by using FLODE as long-range dependencies are probably not that important.
>
> ---
> > **Q3**: “Have the authors actually inspected the eigenvalues and singular values of their models and compared them to the theoretical results?”
>
> We have examined the eigenvalues of our models and compared them to our theoretical predictions, see Point 2 in our general response. These detailed results are included in the supplementary pdf, see Tab. 1 and Fig 1. The observed parameters align perfectly with our theoretical results, indicating that our method can adaptively learn to exhibit HFD behavior, particularly in heterophilic settings. This supports our claim that our proposed model dynamically adjusts its behavior according to the data.
>
> ---
> > **Q4**: “I recommend using boldface and italics in addition to color in Tab. 1. In Fig. 3, the line for FLODE should be thicker or have a different line style.”
>
> We appreciate your suggestions. We will use underline, boldface, and italics in addition to colors in Tab. 1 to ensure its accessibility. Also, in Fig. 3, we will make the line for FLODE thicker for better visibility.
>
> ---
> We value your thorough review and positive feedback on our work, especially given the depth of our theoretical analysis, which you highlighted. In light of our clarifications and the strengths you've identified, we believe that our work represents a significant advancement in addressing the oversmoothing issue in GNNs, backed by a solid theoretical foundation. We hope that our explanations and additional ablations further underscore the connection between our strong experimental results and our theoretical framework, convincing you even more.

---

> > ### Comment · Reviewer_ZW5p · 2023-08-17
> >
> > Thank you for the additional experiments and details addressing my and other reviewers' comments. I don't have any further questions.

---

### Author Rebuttal · Authors · 2023-08-07

We thank the reviewers for their thorough and insightful remarks. We fully implemented all remarks in the revised version of the paper, which we believe improved the paper significantly.

**1. More extensive ablation studies**

To augment our ablation studies on Chameleon and Citeseer (see Appendix F.3), we perform several new experiments, the results of which can be found in the new supplementary pdf. First, we carried out a comprehensive ablation study on “Squirrel”, see Tab. 2. Here, we systematically remove various parts of our model (the learnable exponent, the ODE, both, and the complex weight matrix). The results show a significant drop in performance when any of these components are removed, reinforcing the fact that our model's convincing performance is indeed a result of the combination of the fractional Laplacian and the neural ODE framework.

Upon request of the Reviewers, we conduct another ablation study to investigate the role of depth on Chameleon, Citeseer, Cora, and Squirrel datasets. The results are depicted in Fig. 1 of the supplementary pdf. The results demonstrate that the neural ODE framework enables GNNs to scale to large depths (256 layers). Finally, we see that the fractional Laplacian improves over the standard Laplacian in the heterophilic graphs which is supported by our claims in the main paper. We highlight that using only the fractional Laplacian without the neural ODE framework oftentimes outperforms the standard Laplacian with the neural ODE framework, see ablations in Tab. 2 in the supplementary pdf and Tab. 7 in Appendix F.3. This indicates the importance of the long-range connections built by the fractional Laplacian.

Please also be aware that Cora, Citeseer, and Pubmed are graphs with high homophily levels. While we do not anticipate improved performance on these graphs (as capturing long-range dependencies is less important on the graphs), we also do not expect a performance drop. For instance, we could simply set $\alpha=1$, which should ensure that we achieve at least the same performance as GRAFF, provided the hyperparameters are equal.

**2. Highlighting the intimate connection between our theoretical and experimental findings**

We further demonstrate the close alignment of our theoretical and experimental results in Fig. 1 of the supplementary pdf, which enables us to precisely anticipate when the models will exhibit HFD or LFD behaviors. In this context, we calculated parameters (according to Thm. 5.3) and illustrated at each depth the expected and observed behaviors. For Squirrel and Chameleon, which are heterophilic graphs, we observe that both their theoretical and empirical behaviors are HFD. Additionally, the learned exponent is small. In contrast, for Cora and Citeseer, we see the opposite.

We have also incorporated the calculated parameters (according to Thm. 5.3) for Tab. 1 in the main (see Tab. 1 in the supplementary pdf). Here, we employ the best hyperparams to solve both fractional heat and Schrödinger graph ODEs., further substantiating the intimate link between our theoretical advancements and practical applications. This tight connection provides strong validation for our theoretical framework, indicating its utility in effectively guiding and predicting the behavior of GNNs in practice.

Please also note that we could enforce LFD/HFD behavior by using diagonal channel mixing matrices with only positive/negative entries. However, this doesn’t lead to the same empirical results which may be due to the limited expressivity of the resulting network.

**3. Generalization of Thm. 5.3**

We appreciate the insights provided by two of the reviewers and generalize Thm. 5.3 to any directed graph with normal SNA. The updated version of the Thm. will be included in the revised paper, and the existing proof remains valid as any normal matrix is unitarily diagonalizable, allowing the same proof technique used in Thm. 5.3. We regard further generalizations to arbitrary directed graphs as an intriguing open question for future exploration.

We thank the reviewers for inspiring this straightforward but crucial extension of Thm. 5.3.

**4. Inclusion of baseline models for experiments**

In response to the insightful suggestions from several reviewers, we have expanded our set of compared models (see Tab. below) to include recent neural ODE-based GNNs and other methods designed to mitigate oversmoothing (OS). These additions encompass GREAD, GraphCon, ACMP, and GCN/GAT with DropEdge. Please note that for models not evaluated using the same (standard) splits, we have run their code on these standard splits, a detail we will highlight with a footnote in our revised paper.

The results will be added in our paper:

| Model | Film | Squirrel | Chameleon | Citeseer | Pubmed | Cora |
| - | - | - | - | - | - | - |
| GREAD [Choi et al., 2023] | **37.90±1.17** | 59.22±1.44 | 71.38±1.3 | 77.60±1.81 | **90.23±0.55** | **88.57±0.66** |
| GraphCon [Rusch et al., 2022] | 35.58±1.24 | 35.51±1.40 | 49.63±1.89 | 76.36±2.67 | 88.01±0.47 | 87.22±1.48 |
| ACMP [Wang et al., 2023] | 34.93±1.26 | 40.05±1.53 | 57.59±2.09 | 76.71±1.77 | 87.79±0.47 | 87.71±0.95 |
| GCN+DropEdge [Rong et al., 2020] | 29.93±0.80 | 41.30±1.77 | 59.06±2.04 | 76.57±2.68 | 86.97±0.42 | 83.54±1.06 |
| GAT+DropEdge [Rong et al., 2020] | 28.95 ± 0.76 | 41.27±1.76 | 58.95±2.13 | 76.13±2.20 | 86.91±0.45 | 83.54±1.06 |
| Ours (This work) | 37.16±1.42 | **64.23±1.84** | **73.60±1.55** | **78.07±1.62** | 89.02±0.38 | 86.44±1.17 |

We also move the paragraph detailing our baseline models from Appendix F.1 (lines 704 ff.) to the main text, improving clarity on our selection of robust baseline models for heterophilic graphs. Given the reviewers' emphasis on baseline models addressing oversmoothing — a concern already addressed in our initial submission — this shift seemed important.


---
We address the remaining concerns as part of the response to each Reviewer.

---

### Decision · Program_Chairs · 2023-09-21

**Decision:**

Accept (poster)

**Comment:**

Most reviewers recommended the paper to be accepted.  Reviewer PWaE engaged in a detailed and lengthy discussion with the reviewers and maintained firmly that the paper should be rejected.
I found the discussion between the authors and the reviewers interesting, and the authors' rebuttal sufficient for the acceptance of the paper. Nevertheless, I strongly recommend that the authors add the additional experiments on larger datasets as reviewer PWaE suggested.